# IN-CONTEXT CONVERGENCE OF TRANSFORMERS

## ABSTRACT

Transformers have recently revolutionized many domains in modern machine learning and one salient discovery is their remarkable in-context learning capability, where models can solve an unseen task by utilizing task-specific prompts without further parameters fine-tuning. This also inspired recent theoretical studies aiming to understand the in-context learning mechanism of transformers, which however focused only on *linear* transformers. In this work, we take the first step toward studying the learning dynamics of a one-layer transformer with *softmax* attention trained via gradient descent in order to in-context learn linear function classes. We consider a structured data model, where each token is randomly sampled from a set of feature vectors in either balanced or imbalanced fashion. For data with balanced features, we establish the finite-time convergence guarantee with near-zero prediction error by navigating our analysis over two phases of the training dynamics of the attention map. More notably, for data with imbalanced features, we show that the learning dynamics take a stage-wise convergence process, where the transformer first converges to a near-zero prediction error for the query tokens of dominant features, and then converges later to a near-zero prediction error for the query tokens of under-represented features, respectively via one and four training phases. Our proof features new techniques for analyzing the competing strengths of two types of attention weights, the change of which determines different training phases.

## 1 INTRODUCTION

Transformers (Vaswani et al., 2017) have emerged as the foundational architectures in various domains, including natural language processing (Devlin et al., 2018; OpenAI, 2023), computer vision (Dosovitskiy et al., 2020; He et al., 2022), reinforcement learning (Chen et al., 2021; Janner et al., 2021), and so on. Recently, large language models (LLMs) based on transformers have exhibited remarkable in-context learning capabilities, where the model can solve a new task solely through inference based on prompts of the task without further fine-tuning (Brown et al., 2020).

Such striking abilities have inspired a recent line of research to understand the underlying mechanisms of in-context learning from various aspects (Garg et al., 2022; Min et al., 2022; Wei et al., 2023; Von Oswald et al., 2023; Xie et al., 2021). Among these studies, the pioneering work of Garg et al. (2022) empirically studied in-context learning via an interpretable framework, highlighting the capacity of transformers to acquire in-context knowledge of linear and some more complex function classes. Specifically, they showed that an in-context trained model over a function class $\mathcal{F}$ can accurately predict the function value $f(x_{\text{query}})$ of a new query token $x_{\text{query}}$ for most $f \in \mathcal{F}$ by using a prompt sequence including in-context input-label pairs along with the query token $(x_1, f(x_1), \ldots, x_N, f(x_N), x_{\text{query}})$.

Built on this theoretically amenable setting, many follow-up works explored theoretical properties of in-context learning of transformers from different perspectives such as expressive power (Akyürek et al., 2022; Giannou et al., 2023), generalization (Li et al., 2023b), internal mechanisms (Von Oswald et al., 2023; Bai et al., 2023), etc. Specially, a few recent studies (Zhang et al., 2023a; Mahankali et al., 2023; Ahn et al., 2023) made interesting progress towards understanding the training dynamics of transformers for in-context learning[1]. However, those studies focused only on 'linear' transformers, and does not capture the crucial role of the 'softmax' mapping, which lies in the core design of transformers to be advantageous over other network architectures. Therefore, the following fundamental problem still remains largely open:

---

[1]More detailed discussions for related work can be found in Appendix B.

*How do **softmax**-based transformers trained via gradient descent learn in-context?*

This paper takes the first step toward addressing this problem by investigating the learning dynamics of a single-layer transformer with *softmax* attention trained by gradient descent (GD) for in-context learning. We focus on the setting with training prompts generated from linear regression models as in Garg et al. (2022), and with structured input data, where each token is randomly selected from a set of feature vectors $\{v_k\}_{k=1}^K$ with probability $\{p_k\}_{k=1}^K$, respectively. We then train the transformer over the squared loss of prediction error using GD. We study the training dynamics under both balanced and imbalanced feature distributions, and characterize the in-context learning ability for both settings. We highlight our contributions as follows.

**Our Contributions.**

- We first establish the convergence guarantee for the setting with balanced features, where $p_k = \Theta(\frac{1}{K})$ for each $k \in [K]$, and characterize the training evolution of the attention map into a two-phase dynamic process. In the first phase, for each $k \in [K]$, the parameters of the self-attention module undergo fast growth, aligning the query token featuring $v_k$ with input tokens featuring $v_k$ rapidly disregarding other feature directions. In the second phase, the loss of prediction error converges to a near-minimum value.

- We then prove the convergence for the setting with imbalanced features, where one feature dominates, say $v_1$ with $p_1 = \Theta(1)$, while others are under-represented with $p_k = \Theta(\frac{1}{K})$ for $k > 1$, which serves as a remarkable showcase of the in-context learning capabilities of transformers. We demonstrate that the learning dynamics display a *stage-wise* convergence process. Initially, the transformer quickly attains near-zero prediction error for the query tokens of dominant features, and then converges to near-zero prediction error for the query tokens of under-represented features, irrespective of their infrequent occurrence, through one and four phases, respectively.

- Our analysis hinges on a novel proof technique that characterizes the *softmax* attention dynamics via the interplay between two types of bilinear attention weights: 'weight of query token and its target feature' and 'weight of query token and off-target features'. Which weight plays a dominant role in the attention dynamics can change over the learning process, resulting in different training phases. Our analysis tools may be of independent interest and hold the potential to study various other problems involving transformer architectures.

**Notations.** We let $[K] := \{1, 2, \ldots, K\}$. We use capital letters for matrices (e.g., $A$), and lowercase letters for vectors and scalars (e.g., $a$). For a general matrix $A$, we use $A_i$ to represent the $i$-th column of $A$ and $A_{i:j}$ to indicate a collection of columns spanning from $i$ to $j$. We use $\mathbf{1}\{\cdot\}$ to denote the indicator function. We use $O(K)$, $\Omega(K)$, and $\Theta(K)$ to omit universal constants concerning the variable $K$. We use $\text{poly}(K)$ and $\text{polylog}(K)$ to denote large constant-degree polynomials of $K$ and $\log(K)$, respectively. Given $h(x) \leq 0$ and $g(x) > 0$, we denote $h(x) = -\Omega(g(x))$ if there exists some constant $C_1 > 0$ and $a_1$, s.t. $|h(x)| \geq C_1 g(x)$ for all $x \geq a_1$; $h(x) = -O(g(x))$ if there exist some constant $C_2 > 0$ and $a_2$, s.t. $|h(x)| \leq C_2 g(x)$ for all $x \geq a_2$; $h(x) = \Theta(g(x))$ if there exists some constant $C_3, C_4 > 0$ and $a_3$, s.t. $C_3 g(x) \leq |h(x)| \leq C_4 g(x)$ for all $x \geq a_3$.

## 2 PROBLEM SETUP

In this section, we present our problem formulations, including the in-context learning framework, one-layer transformer architecture, and the training settings we consider in this paper.

### 2.1 IN-CONTEXT LEARNING FRAMEWORK

We adopt the well-established in-context learning framework as given in Garg et al. (2022). The objective is to enable the training of models capable of in-context learning within a specified function class $\mathcal{F}$, where the functions and input data are sampled respectively by the distributions $D_{\mathcal{F}}$ and $D_{\mathcal{X}}$. Specifically, the process is initiated by generating random training prompts as follows. For each prompt, we first sample a random function $f$ from the class according to the distribution $D_{\mathcal{F}}$. We then create a set of random inputs $x_1, \ldots, x_N$ and query $x_{\text{query}}$, all drawn independently by $D_{\mathcal{X}}$. Finally, we compute the value of function $f$ on these inputs to construct the prompt $P = (x_1, y_1, \ldots, x_N, y_N, x_{\text{query}})$, where $y_i = f(x_i)$. The goal for an in-context learner is to use the prompt to form a prediction $\widehat{y}(x_{\text{query}})$ for the query such that $\widehat{y}(x_{\text{query}}) \approx f(x_{\text{query}})$.

**Task Distribution.** In this work, our focus is on the task of linear functions defined as $\mathcal{F} = \left\{ f : \mathcal{X} \to \mathbb{R} \mid f(x) = \langle w, x \rangle \text{ with } w \in \mathbb{R}^d, \mathcal{X} \subset \mathbb{R}^d \right\}$, which is widely adopted in recent studies for in-context learning (Ahn et al., 2023; Zhang et al., 2023a; Mahankali et al., 2023). For each prompt, the task-specific weight $w$ is independently drawn from a task distribution $\mathcal{D}_\Omega$ with zero mean and identity covariance matrix $\mathbf{I}_{d \times d}$.

**Data Distribution.** To specify the data distribution $\mathcal{D}_\mathcal{X}$, we consider a set of distinct features $\{v_k \in \mathbb{R}^d, k = 1, \ldots, K\}$, where all features are orthonormal vectors. Each data point $x$ is sampled from the feature set with the probability $p_k$ for sampling $v_k$, where $p_k \in (0, 1)$ for $k \in [K]$ and $\sum_{k \in [K]} p_k = 1$. Such a data model has been widely employed in the theoretical studies of deep learning, including ensemble methods (Allen-Zhu & Li, 2020), multi-modal learning (Huang et al., 2022), vision transformers (Li et al., 2023a), etc.

## 2.2 ONE-LAYER TRANSFORMER ARCHITECTURE

To present the one-layer transformer model we consider in this work, we first introduce the self-attention mechanism (Bahdanau et al., 2014; Vaswani et al., 2017) for the transformer model.

**Definition 2.1** (Self-Attention (SA) Mechanism). A self-attention layer (Bahdanau et al., 2014; Vaswani et al., 2017) in the single-head case with width $d_e$ consists of the following components: a key matrix $W^{\text{Key}} \in \mathbb{R}^{d_e \times d_e}$, a query matrix $W^Q \in \mathbb{R}^{d_e \times d_e}$, and a value matrix $W^V \in \mathbb{R}^{d_e \times d_e}$. Given a prompt $P$ of length $N$, let $E \in \mathbb{R}^{d_e \times d_N}$ be an embedding matrix of the prompt $P$, and the self-attention mechanism will output:

$$F_{\text{SA}} \left( E; W^{\text{Key}}, W^Q, W^V \right) = W^V E \cdot \text{softmax} \left( \left( W^{\text{Key}} E \right)^\top W^Q E \right), \tag{1}$$

where the $\text{softmax}(\cdot)$ function is applied column-wisely, i.e., for a vector input $z$, the $i$-th entry of $\text{softmax}(z)$ is given by $e^{z_i} / \sum_s e^{z_s}$.

**Embeddings.** For in-context learning, given a prompt $P = (x_1, y_1, \ldots, x_N, y_N, x_{\text{query}})$, a natural token embedding is to stack $x_i \in \mathbb{R}^d$ and $y_i$ into the first $N$ columns. The final column consists of $x_{\text{query}} \in \mathbb{R}^d$ and 0. Formally,

$$E = E(P) = \left( \begin{array}{ccccc} x_1 & x_2 & \cdots & x_N & x_{\text{query}} \\ y_1 & y_2 & \cdots & y_N & 0 \end{array} \right) \in \mathbb{R}^{(d+1) \times (N+1)}.$$

Therefore, $d_N = N + 1$ and $d_e = d + 1$ in the above embedding. Let us further denote the first $d$ rows of $E$ as $E^x(P) \in \mathbb{R}^{d \times (N+1)}$ and the last row of $E$ as $E^y(P) \in \mathbb{R}^{1 \times (N+1)}$. Then we write $E(P) = \{E^x(P), E^y(P)\}$. We omit the dependency on $P$ for $E(P)$, $E^x(P)$ and $E^x(P)$ when there is no ambiguity.

We next instantiate additional operations and certain parameter settings based on the general SA mechanism (1) for our one-layer transformer model to mitigate unnecessary complications in theoretical analysis while keeping the most critical component of the SA mechanism.

**Masking.** Let $M(\cdot)$ denote the masking operation, which masks (removes) the last column of the entry matrix. In other words, for a given matrix $A \in \mathbb{R}^{(d+1) \times (N+1)}$, $M(A)$ yields $A_{1:N} \in \mathbb{R}^{(d+1) \times N}$. We will first mask the embedding matrix $E$ before its multiplication with the key matrix $W^{\text{Key}}$ and the value matrix $W^V$, which results in $W^{\text{Key}} M(E)$ and $W^V M(E)$, in order to prevent the query token from attending to itself. This approach has been commonly taken in previous works (Tian et al., 2023; Mahankali et al., 2023; Von Oswald et al., 2023; Kitaev et al., 2020).

**Reparameterization.** We consolidate the query and key matrices into one matrix denoted as $W^{KQ} \in \mathbb{R}^{(d+1) \times (d+1)}$, often taken in recent theoretical frameworks (Zhang et al., 2023a; Jelassi et al., 2022; Tian et al., 2023). Furthermore, we consider $W^V$ and $W^{KQ}$ in the following specific forms:

$$W^V = \left( \begin{array}{cc} 0_{d \times d} & 0_d \\ 0_d^\top & \nu \end{array} \right), \quad W^{KQ} = \left( \begin{array}{cc} Q & 0_d \\ 0_d^\top & 0 \end{array} \right), \tag{2}$$

where $\nu \in \mathbb{R}$ and $Q \in \mathbb{R}^{d \times d}$. The above structures of $W^V$ and $W^{KQ}$ are inspired by the recent study (Zhang et al., 2023a), which showed that such structured matrices achieve the global optimum in the *linear* SA model. Furthermore, we set $\nu = 1$ (where $\nu$ is the only parameter in $W^V$) and do

not update it during the training. The reason is twofold: 1) this aligns with the common practice in theoretical studies of deep learning, where the last linear layer is often kept fixed to focus on the analysis of hidden layers. Our objective remains highly nonconvex and challenging even with a fixed $\nu$; and 2) the form of the global optimum outlined in recent work (Zhang et al., 2023a) suggests that for *linear* SA, the optimal solution for $\nu$ serves as a scaling factor to normalize the output of linear attention. In our case, the output of *softmax* attention is already inherently normalized.

**Remark 1** (Nealy no loss of optimality). Despite the specific form of $\{W^V, W^{KQ}\}$ that we take, the minimum of the loss function $L^* = \Theta(e^{-\text{poly}(K)})$ (as shown in Theorem 3.1) implies that such a specific form at most incurs an error of $\Theta(e^{-\text{poly}(K)})$ that vanishes exponentially with $K$, compared to the minimum loss over the general parameter space $\{W^V, W^{\text{Key}}, W^Q\}$. Therefore, for our nonlinear *softmax* SA, such specific parameterization does not lose optimality.

With the aforementioned masking operations and reparameterization, the overall transformer model consisting of a single SA layer can be recast in the parameterization of $\theta = \{1, Q\}$ as follows:

$$F_{\text{SA}}(E; \theta) = M(E^y) \cdot \text{softmax}\left(M(E^x)^\top Q E^x\right). \tag{3}$$

Such a reparameterization separates the label $E^y$ from the softmax operator while maintaining simultaneous processing of both input $E^x$ and label $E^y$ information. The prediction for the token $x_{\text{query}}$ will be the last entry of $F_{\text{SA}}$, namely,

$$\widehat{y}_{\text{query}} = \widehat{y}_{\text{query}}(E; \theta) = [F_{\text{SA}}(E; \theta)]_{(N+1)}.$$

Henceforth, we may omit the reference to $E$ and $\theta$, and use $\widehat{y}_{\text{query}}$ if it is not ambiguous.

## 2.3 TRAINING SETTINGS

**Loss Function.** To train the transformer model $F_{\text{SA}}$ over linear regression tasks, we minimize the following squared loss of the prediction error, which has also been taken by (Zhang et al., 2023a; Ahn et al., 2023):

$$L(\theta) = \frac{1}{2}\mathbb{E}_{w\sim\mathcal{D}_\Omega, \{x_i\}_{i=1}^N\cup\{x_{\text{query}}\}\sim\mathcal{D}_\mathcal{X}^{N+1}}\left[\left(\widehat{y}_{\text{query}} - \langle w, x_{\text{query}}\rangle\right)^2\right] \tag{4}$$

where the expectation is taken with respect to the prompt $P$ including input and query tokens $\{x_i\}_{i=1}^N \cup \{x_{\text{query}}\}$ and the weight vector $w$. In the following, we omit subscripts of the expectation to simplify the notation.

**Training Algorithm.** The above learning objective in eq. (4) is minimized via GD with the learning rate $\eta$. At $t = 0$, we initialize $Q^{(0)}$ as zero matrix $\mathbf{0}_{d\times d}$. The parameter is updated as follows:

$$\theta^{(t+1)} = \theta^{(t)} - \eta\nabla_\theta L(\theta^{(t)}).$$

## 3 MAIN RESULTS

In this section, we characterize the convergence of in-context learning by GD for the settings with balanced and imbalanced features, respectively.

To measure the degree to which the query token $x_{\text{query}}$ attends to the specific input token and to a certain class of features, we define the following notions of the attention scores.

**Definition 3.1** (Attention Score). Given a prompt $P = (x_1, y_1, \cdots, x_N, y_N, x_{\text{query}})$ and its corresponding embedding $E$, where $\{x_i \in \mathbb{R}^d\}_{i=1}^N, x_{\text{query}}$ is drawn independently from $\mathcal{D}_\mathcal{X}$, then at time $t$, for $F_{\text{SA}}$ with parameter $\theta^{(t)}$, we define the attention score as follows.

1. Given $i \in [N]$, the attention score for the $i$-th token $x_i$ is

$$\mathbf{attn}_i(\theta^{(t)}; E) := \left[\text{softmax}(M(E^x)^\top Q^{(t)} E^x)\right]_i = \frac{e^{E_i^{x\top} Q^{(t)} E_{N+1}^x}}{\sum_{j\in[N]} e^{E_j^{x\top} Q^{(t)} E_{N+1}^x}}.$$

2. For $k \in [K]$, denote $\mathcal{V}_k(P) \subset [N]$ as the index set for input tokens, such that $x_i = v_k$ for $i \in \mathcal{V}_k(P)$. Then the attention score for the $k$-th feature is given by

$$\mathbf{Attn}_k(\theta^{(t)}; E) := \sum_{i\in\mathcal{V}_k(P)} \mathbf{attn}_i(\theta^{(t)}; E).$$

For simplicity, we represent $\mathbf{attn}_i(\theta^{(t)}; E)$ and $\mathbf{Attn}_k(\theta^{(t)}; E)$ as $\mathbf{attn}_i^{(t)}$ and $\mathbf{Attn}_k^{(t)}$, respectively, and denote $\mathcal{V}_k(P)$ as $\mathcal{V}_k$. We also rewrite the prediction output at time $t$ as follows:

$$\widehat{y}_{\text{query}}^{(t)} = \sum_{i \in [N]} \mathbf{attn}_i^{(t)} y_i = \sum_{k \in [K]} \mathbf{Attn}_k^{(t)} \langle w, v_k \rangle. \tag{5}$$

### 3.1 IN-CONTEXT LEARNING WITH BALANCED FEATURES

In this subsection, we study in-context learning with *balanced* features, where the probabilities of sampling all $K$ features are in the same order, i.e., $p_k = \Theta(\frac{1}{K})$ for each $k \in [K]$. In such a setting, each feature appears equally likely in the prompt, ensuring their equal recognition. The following theorem characterizes the convergence of GD.

**Theorem 3.1** (In-context Learning with Balanced Features). *Suppose $p_k = \Theta(\frac{1}{K})$ for $k \in [K]$. For any $0 < \epsilon < 1$, suppose $N \geq \text{poly}(K)$ and $\text{polylog}(K) \gg \log(\frac{1}{\epsilon})$. We apply GD to train the loss function given in eq. (4). Then with at most $T^* = O(\frac{\log(K)K^2}{\eta} + \frac{K \log\left(K\epsilon^{-\frac{1}{2}}\right)}{\epsilon\eta})$ iterations, we have*

1. *The loss converges: $L(\theta^{(T^*)}) - L^* \leq \epsilon$, where $L^* = \Theta(e^{-\text{poly}(K)})$ is the global minimum of the population loss in eq. (4).*

2. *Attention score concentrates: if $x_{query} = v_k$, then with probability at least $1 - e^{-\Omega(\text{poly}(K))}$[2], the one-layer transformer nearly "pays all attention" to input tokens featuring $v_k$, i.e., $(1 - \mathbf{Attn}_k^{(T^*)})^2 \leq O(\epsilon)$.*

Theorem 3.1 shows that training a one-layer transformer with softmax attention can converge to the minimum of the objective loss in the reparameterization space via GD, with polynomial time efficiency with respect to $K$ and $\frac{1}{\epsilon}$. The learning dynamics for such a case with balanced features exhibit a **two-phase behavior**. (i) The first term of $T^*$ captures the duration of phase I, where the network actively aligns the query token (suppose $x_{\text{query}} = v_k$) with those tokens featuring $v_k$ itself, thus substantially increasing $\mathbf{Attn}_k^{(t)}$ to a constant level. (ii) The second term captures the duration of phase II, where the loss converges to the near-zero prediction error.

**In-context Learning Ability.** For the obtained model with $\theta^{(T^*)}$, let us evaluate a test prompt associated with a linear task $w$, which might not be drawn from the support of $\mathcal{D}_\Omega$ (i.e., $w$ may not be present in the training process), but has its data drawn by $\mathcal{D}_\mathcal{X}$. Suppose the query token is $x_{\text{query}} = v_k$. Following from the attention score concentration principle in Theorem 3.1, eq. (5) yields that with high probability the query prediction is given by

$$\widehat{y}_{\text{query}}^{(T^*)} = \mathbf{Attn}_k^{(T^*)} \langle w, v_k \rangle + \sum_{m \neq k} \mathbf{Attn}_m^{(T^*)} \langle w, v_m \rangle \approx \langle w, v_k \rangle.$$

This implies that the in-context learned model can still well approximate the test prompt even if the task model $w$ does not lie in the support of the training task distribution $\mathcal{D}_\Omega$ and was *unseen* during training. This showcases the remarkable in-context learning capability of trained transformers.

### 3.2 IN-CONTEXT LEARNING WITH IMBALANCED FEATURES

In real-world datasets, skewed distributions are common, where a few classes or features dominate in data while others are under-represented. It is typically difficult to train models to perform well on features that have limited representation in those datasets (Cui et al., 2019; Chou et al., 2020). In this subsection, we investigate the setting with imbalanced features, where the dominant feature $v_1$ is sampled with the probability $p_1 = \Theta(1)$, and all other features are sampled with $p_k = \Theta(\frac{1}{K})$ for $2 \leq k \leq K$. We will show that somewhat remarkably, in-context learning is less sensitive to imbalanced features and can achieve a near-zero error even when the query token takes an under-represented feature.

To investigate the performance for the imbalanced scenario, we focus on the following prediction error for each feature $v_k$:

$$\mathcal{L}_k(\theta) = \frac{1}{2} \mathbb{E}\left[ (\widehat{y}_{\text{query}} - \langle w, x_{\text{query}} \rangle)^2 \,\big|\, x_{\text{query}} = v_k \right]. \tag{6}$$

The following theorem characterizes the convergence of GD.

---

[2] The randomness originates from the first $N$ input tokens in the test prompt.

**Theorem 3.2** (In-context Learning with Imbalanced Features). *Suppose $p_1 = \Theta(1)$ and $p_k = \Theta(\frac{1}{K})$ for $2 \leq k \leq K$. For any $0 < \epsilon < 1$, suppose $N \geq \mathrm{poly}(K)$, and $\mathrm{polylog}(K) \gg \log(\frac{1}{\epsilon})$. We apply GD to train the loss function given in eq. (4). Then the following results hold.*

1. *The prediction error for the **dominant** feature converges: for $v_1$, with at most $T_1 = O(\frac{\log(\epsilon^{-\frac{1}{2}})}{\eta\epsilon})$ GD iterations, $\mathcal{L}_1(\theta^{(T_1)}) \leq \mathcal{L}_1^* + \epsilon$, where $\mathcal{L}_1^* = \Theta(e^{-\mathrm{poly}(K)})$ is the global minimum of eq. (6) for $k = 1$;*

2. *The prediction error for the **under-represented** features converges: for $v_k$ with $2 \leq k \leq K$, with at most $T_k = O(\frac{\log(K)K^2}{\eta} + \frac{K \log\left(K\epsilon^{-\frac{1}{2}}\right)}{\epsilon\eta})$ GD iterations, $\mathcal{L}_k(\theta^{(T_k)}) \leq \mathcal{L}_k^* + \epsilon$, where $\mathcal{L}_k^* = \Theta(e^{-\mathrm{poly}(K)})$ is the global minimum of eq. (6);*

3. *Attention score concentrates: for each $k \in [K]$, if the query token is $v_k$, then after $T_k$ iterations, with probability at least $1 - e^{-\Omega(\mathrm{poly}(K))}$, the one-layer transformer nearly "pays all attention" to input tokens featuring $v_k$: $(1 - \mathbf{Attn}_k^{(T_k)})^2 \leq O(\epsilon)$.*

Theorem 3.2 shows that the GD dynamics of the in-context training exhibit *'stage-wise'* convergence. The trained transformer rapidly (within $T_1$) converges to a model that achieves a near-zero prediction error $\mathcal{L}_1$ for the dominant feature; and then takes a much longer time (up to $T_k \gg T_1$) to converge to a model that attains a near-zero prediction error $\mathcal{L}_k$ for the under-represented features. Our analysis captures the later learning dynamics associated with the under-represented features into a four-phase behavior as further described in the subsequent section. Despite the longer convergence time it takes, in-context learning still achieves the same accurate prediction for under-represented features as that for the dominant feature.

## 4 OVERVIEW OF TRAINING PHASES

In this section, we explain our key ideas for analyzing the in-context learning capabilities of transformers. We will first characterize the training process of the setting with imbalanced features for under-represented features in Section 4.1, which comprehensively exhibits four phases. Other scenarios take only one or two of those phases, which we will briefly describe in Section 4.2. The complete proofs of all the results are provided in the appendix.

We will first provide the general training dynamics for the *bilinear attention weights* (defined in Definition 4.1 below), which is useful for analyzing all learning phases. These quantities are the key elements in the attention scores $\mathbf{attn}_i^{(t)}$ for $1 \leq i \leq N$, which play an important role in determining the prediction $\widehat{y}_{\mathrm{query}}^{(t)}$. Hence, our analysis mainly tracks the training dynamics of those bilinear attention weights.

**Definition 4.1.** (Bilinear Attention Weights) Given $k, n \in [K]$, where $k \neq n$, for $t \geq 0$, we define the bilinear attention weights as follows:

$$A_k^{(t)} := v_k^\top Q^{(t)} v_k, \quad B_{k,n}^{(t)} := v_n^\top Q^{(t)} v_k.$$

By our initialization, we have $A_k^{(0)} = B_{k,n}^{(0)} = 0$.

To further interpret these weights, suppose the query token corresponds to the feature $v_k$. Then $e^{A_k^{(t)}}$ serves as the (un-normalized) weight for the input token featuring $v_k$, while $e^{B_{k,n}^{(t)}}$ captures the weight for the input token featuring a different vector $v_n$ with $n \neq k$. Having a larger $A_k^{(t)}$ compared to other $B_{k,n}^{(t)}$ indicates a better capture of the target feature $v_k$. As shown in eq. (5), this condition implies a higher 'attention' towards input tokens featuring $v_k$, resulting in $\widehat{y}_{\mathrm{query}}^{(t)} \approx \sum_{i \in \mathcal{V}_k} \mathbf{attn}_i^{(t)} y_i \approx \langle w, v_k \rangle$, where the prediction well approximates the ground truth.

The following lemma provides the GD updates of the bilinear attention weights $A_k^{(t)}$ and $B_{k,n}^{(t)}$.

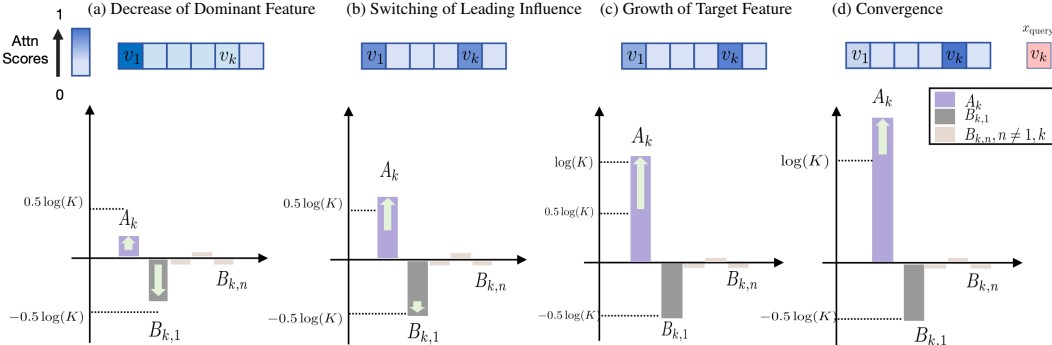

Figure 1: Overview of the dynamics of attention scores and bilinear attention weights for under-represented features. Assume the query token is $v_k$ with $2 \leq k \leq K$. The *top* row depicts the trend of the attention score $\mathbf{Attn}_m^{(t)}$ for each feature $v_m$, where a darker color corresponds to a higher score. The *bottom* row shows the interplay and leading effect among bilinear attention weights $A_k^{(t)}, B_{k,1}^{(t)}$, and $B_{k,n}^{(t)}$ (where $n \neq 1, k$) in different training phases. **(a)** Phase I: $B_{k,1}^{(t)}$ significantly decreases and the attention on tokens with the dominant feature $v_1$ is suppressed (Section 4.1.1); **(b)** Phase II: With the suppression of $\mathbf{Attn}_1^{(t)}$, the decreasing rate for $B_{k,1}^{(t)}$ drops and the growth of $A_k^{(t)}$ becomes the leading influence (Section 4.1.2); **(c)** Phase III: $A_k^{(t)}$ rapidly grows and $\mathbf{Attn}_k^{(t)}$ reaches $\Omega(1)$ (Section 4.1.3); **(d)** Phase IV: $\mathbf{Attn}_k^{(t)}$ nearly grows to 1 and the prediction error converges to a global minimum (Section 4.1.4).

**Lemma 4.1.** *Let $t \geq 0$. For $k, n \in [K]$, where $k \neq n$, $A_k^{(t)}$ and $B_{k,n}^{(t)}$ satisfy:*

$$A_k^{(t+1)} = A_k^{(t)} + \eta \alpha_k^{(t)}, \qquad B_{k,n}^{(t+1)} = B_{k,n}^{(t)} + \eta \beta_{k,n}^{(t)},$$

$$\alpha_k^{(t)} = \mathbb{E}\left[ \mathbf{1}\{x_{query} = v_k\} \, \mathbf{Attn}_k^{(t)} \cdot \left( \sum_{m \neq k} \mathbf{Attn}_m^{(t)2} + (1 - \mathbf{Attn}_k^{(t)})^2 \right) \right],$$

$$\beta_{k,n}^{(t)} = \mathbb{E}\left[ \mathbf{1}\{x_{query} = v_k\} \, \mathbf{Attn}_n^{(t)} \cdot \left( \sum_{m \neq k} \mathbf{Attn}_m^{(t)2} - \mathbf{Attn}_n^{(t)} - \mathbf{Attn}_k^{(t)}(1 - \mathbf{Attn}_k^{(t)}) \right) \right].$$

Lemma 4.1 shows that $A_k^{(t)}$ is monotonically increasing at any time since $\alpha_k^{(t)} \geq 0$, whereas the monotonicity does not always hold for $B_{k,n}^{(t)}$. Therefore, we need to analyze whether $B_{k,n}^{(t)}$ decreases and determine its rate of change compared to $A_k^{(t)}$. Such a comparison between $B_{k,n}^{(t)}$ and $A_k^{(t)}$ determines which bilinear weight plays a dominant role in the attention dynamics, and the change of the leading weight over the learning process results in different training phases.

## 4.1 LEARNING PROCESS FOR UNDER-REPRESENTED FEATURES

We consider the setting with imbalanced features and focus on the under-represented features.

Given a prompt $P = (x_1, y_1, \cdots, x_N, y_N, x_{query})$, denote $P_{input}$ to be the collection of input tokes, i.e., $\{x_i\}_{i=1}^N$. Recall that $|\mathcal{V}_k|$ is the number of input tokens featuring $v_k$. Based on our data generation setup, we can show that for imbalanced data, with high probability, $P_{input}$ belongs to

$$\mathcal{E}_{imbal}^* := \left\{ P_{input} : |\mathcal{V}_1| = \Theta(N), |\mathcal{V}_k| = \Theta\left(\frac{N}{K}\right) \text{ for } 2 \leq k \leq K \right\}.$$

In the following, we focus on the event that $P_{input} \in \mathcal{E}_{imbal}^*$ unless otherwise specified. We next characterize the learning process for under-represented features $v_k$ with $k > 1$ by four phases. An illustration of these four phases is provided in Figure 1.

### 4.1.1 PHASE I: DECREASE OF DOMINANT FEATURE.

Consider the query token featuring $v_k$ for some $k > 1$. At $t = 0$, $A_k^{(0)} = B_{k,n}^{(0)} = 0$, and hence $\mathbf{attn}_i^{(0)} = \frac{1}{N}$ for $i \in [N]$ which implies that the transformer equally attends each input token. However, due to the imbalanced occurrence of features in $\mathcal{E}_{imbal}^*$, the number of tokens featuring $v_1$

is much larger than others. Hence, $\mathbf{Attn}_1^{(0)} = \frac{|\mathcal{V}_1|}{N} \geq \Omega(1)$ while $\mathbf{Attn}_m^{(0)} = \Theta(\frac{1}{K})$ for $m > 1$. Therefore, by Lemma 4.1, we obtain

$$\beta_{k,1}^{(0)} = \mathbb{E}\left[\mathbf{1}\{x_{\text{query}} = v_k\} \mathbf{Attn}_1^{(0)}\right.$$
$$\left. \cdot \left(\sum_{m \neq k,1} \mathbf{Attn}_m^{(0)\,2} - \mathbf{Attn}_1^{(0)}(1 - \mathbf{Attn}_1^{(0)}) - \mathbf{Attn}_k^{(0)}(1 - \mathbf{Attn}_k^{(0)})\right)\right] \leq -\Omega(\tfrac{1}{K}),$$

whereas $\alpha_k^{(0)}, |\beta_{k,n}^{(0)}| \approx \Theta(\frac{1}{K^2})$ for $n \neq k, 1$. Therefore, $B_{k,1}^{(t)}$ enjoys a much larger decreasing rate initially. It can be shown that the decrease of $B_{k,1}^{(t)}$ will dominate for a certain time period that defines phase I. The following lemma summarizes our main result in this phase.

**Lemma 4.2** (Informal). *Under the same conditions as Theorem 3.2, given $k > 1$, there exists $T_{1,k} = O(\frac{\log(K)K^{1.98}}{\eta})$, such that for all $0 \leq t \leq T_{1,k}$*

$$\beta_{k,1}^{(t)} \leq -\Omega\left(\tfrac{1}{K^{1.98}}\right), \quad \alpha_k^{(t)} = \Theta\left(\tfrac{1}{K^2}\right), \quad |\beta_{k,n}^{(t)}| \leq O\left(\tfrac{\alpha_k^{(t)} + |\beta_{k,1}^{(t)}|}{K}\right) \quad \text{for all } n \neq k, 1.$$

$B_{k,1}^{(T_{1,k}+1)} \leq -0.49\log(K)$, *while $A_k^{(T_{1,k}+1)}$ and $B_{k,n}^{(T_{1,k}+1)}$ for $n \neq k, 1$ remain close to zero.*

During phase I, $B_{k,1}^{(t)}$ significantly decreases, leading to a reduction in $\mathbf{Attn}_1^{(t)}$, whereas other $\mathbf{Attn}_n^{(t)}$ with $n > 1$ remain at the level of $\Theta(\frac{1}{K})$. By the end of this phase, $(\mathbf{Attn}_1^{(t)})^2$ drops to $O(\frac{1}{K^{0.98}})$, resulting in a decrease in $|\beta_{k,1}^{(t)}|$ as it approaches $\alpha_k^{(t)}$. Phase II then begins.

### 4.1.2 PHASE II: SWITCHING OF LEADING INFLUENCE
Soon after entering this phase, the dominance role of $B_{k,1}^{(t)}$ diminishes as $|\beta_{k,1}^{(t)}|$ reaches the same order of magnitude as $\alpha_k^{(t)}$. The following result captures the shift of the leading influence, where the growth of $A_k^{(t)}$ takes dominance.

**Lemma 4.3** (Informal). *Under the same conditions as Theorem 3.2, given $k > 1$, there exists $T_{2,k} = T_{1,k} + O(\frac{\log(K)K^2}{\eta})$, such that at iteration $t = T_{2,k} + 1$, we have*

$$A_k^{(T_{2,k}+1)} \geq 0.5\log(K), \quad B_{k,1}^{(T_{2,k}+1)} \in [-0.51\log(K), -0.49\log(K)]$$

*and $B_{k,n}^{(T_{2,k}+1)}$ for $n \neq k, 1$ remain close to zero.*

Lemma 4.3 shows that by the end of phase II, $A_k^{(t)}$ matches the magnitude of $B_{k,1}^{(t)}$, and during phase II $B_{k,1}^{(t)}$ changes only slightly from the end of phase I. This suggests that, at certain moments in this phase, $A_k^{(t)}$ significantly increases and its growth becomes the dominant factor. We next provide some insights into the reasons behind this transition. Once $B_{k,1}^{(t)}$ decreases to $-0.5\log(K)$, we observe that $|\beta_{k,1}^{(t)}| \approx \alpha_k^{(t)} = \Theta(\frac{1}{K^2})$. After this point, it becomes challenging for $B_{k,1}^{(t)}$ to decrease significantly compared to the increase in $A_k^{(t)}$. To illustrate, let us suppose a minimal decrease of $B_{k,1}^{(t)}$ by an amount of $0.01\log(K)$. This would yield that $\mathbf{Attn}_1^{(t)} \leq O(\frac{1}{K^{0.501}})$ and $\beta_{k,1}^{(t)} \leq O(\frac{1}{K^{2.01}})$, while $\mathbf{Attn}_k^{(t)} \geq \Omega(\frac{1}{K})$ and $\alpha_k^{(t)} \geq \Omega(\frac{1}{K^2})$, establishing a situation where $\alpha_k^{(t)} \gg \beta_{k,1}^{(t)}$. Such a discrepancy leads to the switching of the dominant effect.

### 4.1.3 PHASE III: GROWTH OF TARGET FEATURE
After a transition phase, we observe that $A_k^{(t)}$ enjoys a larger gradient $\alpha_k^{(t)} \approx \Theta(\frac{1}{K^{1.5}})$ compared to $|\beta_{k,1}^{(t)}| \leq O(\frac{1}{K^{1.98}})$ and $|\beta_{k,n}^{(t)}| \leq O(\frac{1}{K^3})$ with $n \neq k, 1$. This gap between $\alpha_k^{(t)}$ and $\beta_{k,n}^{(t)}$ remains over the period, and the gradient $\alpha_k^{(t)}$ continues to grow, driving the rapid growth of $A_k^{(t)}$ with $B_{k,n}^{(t)}$ being relatively unchanged. The following lemma summarizes our main results in this phase.

**Lemma 4.4** (Informal). *Under the same conditions as Theorem 3.2, given $k > 1$, there exists $T_{3,k} = O(\frac{\log(K)K^{1.5}}{\eta})$, such that for all $T_{2,k} < t \leq T_{3,k}$*

$$\alpha_k^{(t)} \geq \Omega\left(\tfrac{1}{K^{1.5}}\right), \beta_{k,1}^{(t)} \in \left[-O\left(\tfrac{\alpha_k^{(t)}}{K^{0.48}}\right), -\Omega\left(\tfrac{1}{K^{2.01}}\right)\right], |\beta_{k,n}^{(t)}| \leq O\left(\tfrac{\alpha_k^{(t)} + |\beta_{k,1}^{(t)}|}{K}\right) \text{ with } n \neq k, 1.$$

*At time $t = T_{3,k} + 1$, we have $A_k^{(T_{3,k}+1)} \geq \log(K)$.*

Lemma 4.4 follows because the continuous growth of $\alpha_k^{(t)}$ is mainly driven by $\mathbf{Attn}_k^{(t)}$, where $1 - \mathbf{Attn}_k^{(t)}$ remains at the constant order. However, as $A_k^{(t)}$ reaches $\log(K)$, $\mathbf{Attn}_k^{(t)}$ is above $\Omega(1)$, necessitating a more detailed analysis to control $\alpha_k^{(t)}$, which starts the final phase.

### 4.1.4 PHASE IV: CONVERGENCE

After learning the target feature $v_k$ at a certain level, the prediction error converges. We characterize this in the following lemma, where we establish a connection between $\alpha_k^{(t)}$ and the prediction error via analyzing the change of $1 - \mathbf{Attn}_k^{(t)}$ that diminishes during this phase.

**Lemma 4.5** (Informal). *Under the same conditions as Theorem 3.2, given $0 < \epsilon < 1$, for each $k > 1$, there exists $T_{4,k} = T_{3,k} + O(\frac{K \log(K\epsilon^{-\frac{1}{2}})}{\eta \epsilon})$, such that for all $T_{3,k} < t \leq T_{4,k}$*

$$\alpha_k^{(t)} \geq \Omega(\frac{\epsilon}{K}), \quad \beta_{k,n}^{(t)} \in [-O(\frac{\alpha_k^{(t)}}{K^{0.49}}), 0], \quad \beta_{k,n}^{(t)} \in [-O(\frac{\alpha_k^{(t)}}{K}), 0] \text{ with } n \neq k, 1.$$

*At time $t = T_{4,k} + 1$, we have $\mathcal{L}_k(\theta^{(T_{4,k}+1)}) - \mathcal{L}_k^* < \epsilon$ and $(1 - \mathbf{Attn}_k^{(t)})^2 \leq O(\epsilon)$, if $x_{query} = v_k$ and $P_{input} \in \mathcal{E}_{imbal}^*$.*

The convergence result for $k > 1$ stated in Theorem 3.2 directly follows by choosing $T_k^* = T_{4,k} + 1$.

### 4.2 TRAINING DYNAMICS OF OTHER SETTINGS

We next describe the training dynamics of other settings, which take the phases similar to those discussed in Section 4.1.

**Imbalanced Setting for the Dominant Feature.** For the dominant feature $v_1$ in the imbalanced setting, since the overall attention $\mathbf{Attn}_1^{(0)}$ to the target feature already reaches $\Omega(1)$ due to the abundance of tokens featuring $v_1$ in $\mathcal{E}_{imbal}^*$, the training directly enters the convergence stage, as summarized in the following lemma.

**Lemma 4.6** (Informal). *Under the same conditions as Theorem 3.2, given $k > 1$, there exists $T_1 = O(\frac{\log(\epsilon^{-\frac{1}{2}})}{\eta \epsilon})$, such that for all $t \leq T_1$*

$$\alpha_1^{(t)} \geq \Omega(\epsilon), \quad \beta_{1,n}^{(t)} \in [-O(\frac{\alpha_n^{(t)}}{K}), 0] \text{ with } n > 1.$$

*Further $\mathcal{L}_1(\theta^{(T_1+1)}) - \mathcal{L}_1^* < \epsilon$, and $(1 - \mathbf{Attn}_1^{(T_1+1)})^2 \leq O(\epsilon)$ if $x_{query} = v_1$ and $P_{input} \in \mathcal{E}_{imbal}^*$.*

**Balanced Scenarios.** Similarly to imbalanced settings, we can show that for balanced data, with high probability, $P_{input}$ belongs to $\mathcal{E}_{bal}^* := \{P_{input} : |\mathcal{V}_k| = \Theta(\frac{N}{K}) \text{ for all } k \in [K]\}$. At initialization, the transformer uniformly assigns attention to each token, i.e., $\mathbf{attn}_i^{(0)} = \frac{1}{N}$ for $i \in [N]$. Unlike the imbalanced case, here, due to $P_{input} \in \mathcal{E}_{bal}^*$, we have that $\mathbf{Attn}_m^{(0)} = \Theta(\frac{1}{K})$ for $m \in [K]$, indicating nearly equal attention to each feature. Consequently, as Lemma 4.1, we observe a significantly larger gradient in $A_k^{(t)}$ at the outset, with $\alpha_k^{(0)} \approx \Theta(\frac{1}{K^2})$, compared to $|\beta_{k,n}^{(0)}| \approx \Theta(\frac{1}{K^3})$ for $n \neq k$. This behavior mirrors the observations from phase III for under-represented features, allowing us to directly generalize the analysis.

## 5 CONCLUSIONS

In this work, we investigated the training dynamics of a one-layer transformer with softmax attention trained by GD for in-context learning. We analyzed two settings respectively with balanced and imbalanced features, and proved the guaranteed convergence to a vanishing in-context prediction error by detailing the evolution of attention dynamics for both settings. Interestingly, we characterized a four-phase behavior for the imbalanced settings that sheds light on the intricate attention dynamics between dominant and target under-represented features during training. To our knowledge, this is the first work that rigorously analyzed the *softmax* attention dynamics for in-context learning. Our approach features novel ideas for phase decomposition based on the changes of the dominant role between two types of bilinear attention weights in the learning process, and has the potential to facilitate further theoretical understanding of how transformers perform in other algorithms and learning paradigms.

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

# Supplementary Materials

## A  EXPERIMENTS

In this section, we present numerical results to demonstrate that our theoretical results are consistent with the actual dynamics during the in-context training of transformers. In particular, our results verify: (1) stage-wise convergence we characterize for the training process; (2) the concentration of the attention scores; and (3) the multi-phase transition during the training process, which we characterize in our proof of convergence.

### A.1  EXPERIMENTAL SETTINGS

**Task and Data Generations.**  We follow the task and data distributions introduced in Section 2.1. For each task, we sample the task weight $w$ from $\mathcal{N}(0, \mathbf{I}_{d \times d})$. Each data point is drawn from the given feature set $\{v_k \in \mathbb{R}^d, k = 1, \cdots, K\}$ with probability $p_k$ for sampling $v_k$, where all features are orthonormal vectors, and $p_k \in (0,1)$ satisfies $\sum_{k=1}^K p_k = 1$. The prompt consists of $N$ random inputs $\{x_i\}_{i=1}^N$ with their task values given by $\{y_i\}_{i=1}^N = \{w^\top x_i\}_{i=1}^N$, and a query $x_{\text{query}}$. We consider the setting with $d = 16$, $N = 60$ and $K = 3$.

We consider the following two types of data distributions with $(p_1, \cdots, p_K)$ given by

- Balanced feature: $p_i = \frac{1}{3}$ for $i \in [3]$;

- Imbalanced feature: $v_1$ is the dominant feature with $p_1 = 0.8$ and $\{v_2, v_3\}$ are under-represented features with $p_2 = p_3 = 0.1$.

**Transformer Architecture.**  We consider a simplified transformer network. The model consists of one block with a single-head self-attention layer, followed by a feedforward neural network, which incorporates layer normalization and ReLU activation, and finally concludes with a linear layer for output processing.

**Training Setup.**  We collect $M = 300$ randomly generated prompts and then train the model based on the empirical version of the training objective eq. (4) for 400 epochs using Adam (Kingma & Ba, 2014) with full batch and the learning rate of 0.002. Notice that Adam is a preferred choice for its stability in training transformers, which is also consistent with recent studies (Garg et al., 2022; Zhang et al., 2023a) to tackle the in-context learning ability of transformers over linear function classes.

**Evaluations.**  We focus on two performance metrics. 1). Prediction error: As defined in eq. (6), the prediction error $\mathcal{L}_k$ measures the loss conditioned on the event that the query token is $v_k$. We evaluate $\mathcal{L}_k$ by averaging the squared loss on the prompts whose query token is $v_k$. 2). Attention score: We also evaluate the attention score $\mathbf{Attn}_k$ for each feature, where $\mathbf{Attn}_k$ is defined in Definition 3.1 as the average attention score for the $k$-th feature over the prompts with query token featuring $v_k$.

### A.2  NUMERICAL RESULTS

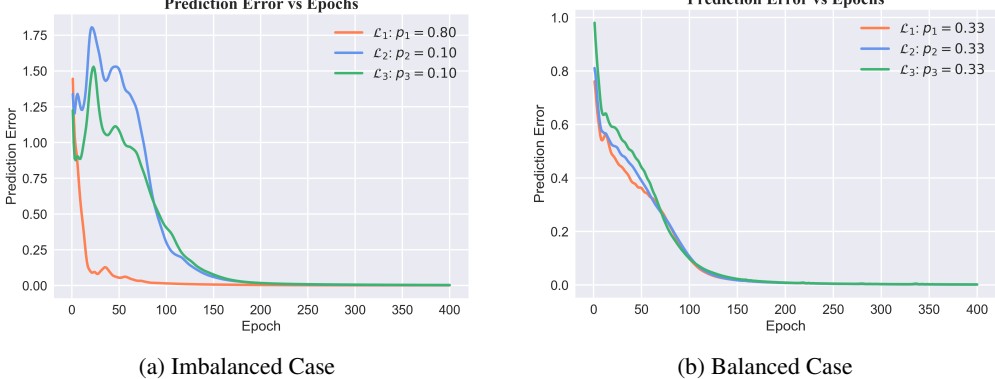

(a) Imbalanced Case  (b) Balanced Case

Figure 2: The prediction error for each feature versus the training epochs

**Stage-Wise Convergence.** In Figure 2, we plot the evolution of prediction errors for each feature throughout the training process. In the imbalanced case (Figure 2a), the transformer quickly converges to a model with nearly vanishing prediction error $\mathcal{L}_1$ for the dominant feature. However, the errors $\mathcal{L}_2$ and $\mathcal{L}_3$ for under-represented features initially fluctuate and then converge to zero after a considerably longer period. This behavior aligns with the stage-wise convergence process characterized in our Theorem 3.2. On the other hand, in the balanced scenario (Figure 2b), the prediction errors for all features decrease in a similar manner throughout the training, which validates our theory on convergence in the balanced case in Theorem 3.1.

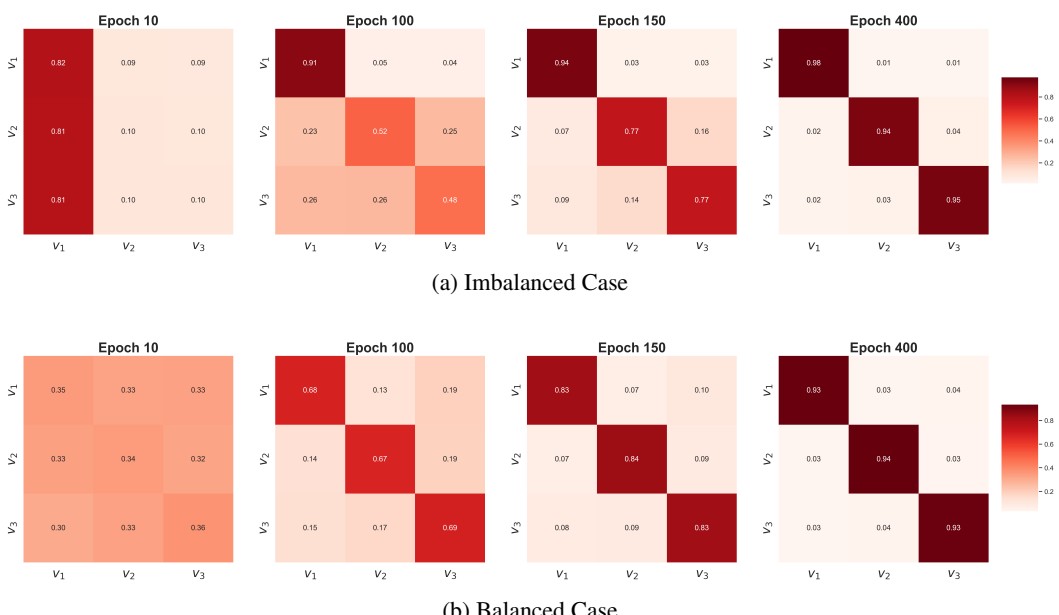

Figure 3: The attention heatmap during the training. For each heatmap, the $i$-th row represents the average attention scores of the query token attending to each feature when $x_{\text{query}} = v_i$.

**Attention Score Concentration.** In Figure 3, we present the dynamic evolution of attention scores throughout the training process for both balanced and imbalanced scenarios. For each $k \in [3]$, and when $x_{\text{query}} = v_k$, it is observed that $\mathbf{Attn}_k$ progressively increases to be close to 1 while other $\mathbf{Attn}_{k'}$ diminishes at the end of training. These results support the principle of attention score concentration as elaborated in Theorems 3.1 and 3.2, and demonstrate that the attention are allocated towards those tokens with the same feature as the query token.

**Multi-Phase Transition during Training Process.** Figure 3 further illustrates the *multi-phase* convergence process of under-represented features, which verifies those multiple phases we characterize in our proof of convergence as discussed in Section 4. We elaborate this by taking the case with $x_{\text{query}} = v_2$ as an example. In Figure 3a and focusing on the row of $x_{\text{query}} = v_2$, from epoch 10 to 100, $\mathbf{Attn}_1$ decreases and $\mathbf{Attn}_3$ increases. This change suggests that the decrease in $B_{2,1}$ is the main factor in the initial phase (phase I)[3]. Then moving to epoch 150, the simultaneous increase in $\mathbf{Attn}_2$ and decreases in $\mathbf{Attn}_1$ and $\mathbf{Attn}_3$ indicate a shift in the dominant effect, with the rise of $A_2$ taking over (phases II and III). Finally, the concentration of attention scores at epoch 400 corresponds to the last phase of convergence.

---

[3]If the increase in $A_2$ were the driving factor, we would expect a decrease in all off-diagonal attention scores including $\mathbf{Attn}_3$ similar to Figure 3b for the balanced case, which contradicts our observation.

## B   ADDITIONAL RELATED WORK

**In-Context Learning.**   Recent studies explored theoretical properties of transformers for in-context learning from various perspectives. Focusing on expressive capacity, Akyürek et al. (2022) studied linear regression tasks and showed that trained in-context learners can represent GD of ridge regression and exact least-squares regression. Giannou et al. (2023) proved the existence of a looped transformer that can emulate in-context learning algorithms. Von Oswald et al. (2023); Dai et al. (2023) also showed that transformer trained in-context implements the GD. Bai et al. (2023) further provided comprehensive results of transformers including the expressive power, in-context prediction power, and sample complexity of pretraining, and then constructed two general mechanisms for algorithm selection.Li et al. (2023b) analyzed the generalization error of trained in-context learning transformers. Another line of work considered in-context learning from a different perspective within the Bayesian framework (Xie et al., 2021; Zhang et al., 2023b; Wang et al., 2023; Jiang, 2023; Han et al., 2023; Wies et al., 2023; Ahuja et al., 2023).

Closely related to our work is the line of research by Zhang et al. (2023a); Mahankali et al. (2023); Ahn et al. (2023), which investigated the training dynamics of in-context learning. Specifically, Mahankali et al. (2023) considered linear regression tasks and showed that the one-layer transformer that minimizes the pre-training loss implements one step of GD. Zhang et al. (2023a) investigated in-context learning of transformers with a single linear self-attention layer trained by gradient flow on linear regression tasks, and showed that gradient flow finds a global minimum. Ahn et al. (2023) investigated the landscape of the loss function for linear transformers trained over random instances of linear regression. However, all those works considered only transformers with *linear* self-attention layers and do not capture the crucial role of the *softmax* mapping, which lies in the core design of transformers to be advantageous over other network architectures. Our work focuses on nonlinear transformers with *softmax attention* and characterizes their training dynamics for in-context learning.

**Training Dynamics of Transformers.**   Jelassi et al. (2022) proposed a simplified Vision Transformers (ViT) model in which the attention matrix solely depends on the positional embeddings and showed that the trained model by GD can learn spatial structure. Li et al. (2023a) studied the training of shallow ViT for a classification task and characterized the sample complexity to achieve a desirable generalization performance. However, their analysis relied on a good initialization near the target pattern, which may not be feasible in practice. Tian et al. (2023) analyzed the SGD training dynamics for a one-layer transformer with one self-attention plus one decoder layer and showed how the self-attention layer combines input tokens during the training, but this work did not provide the convergence guarantee for SGD. Recently, Tarzanagh et al. (2023) established an equivalence between the optimization geometry of self-attention and a hard-margin SVM problem that separates optimal input tokens from non-optimal tokens using linear constraints on the outer-products of token pairs. While the mathematical setup of these problems is different from in-context learning, some of our analysis techniques may be useful for studying the training dynamics of these problems.

## C Preliminaries

In this section, we will introduce warm-up gradient computations and probabilistic lemmas that establish essential properties of the data and loss, which are pivotal for the technical proofs in the upcoming sections. Towards the conclusion of this section, we will also provide a summary of the key notations introduced in both the main content and these preliminary sections. These notations will be frequently utilized in our subsequent analyses.

### C.1 Gradient Computations

We first calculate the gradient for $Q$ (note that we do not update the parameter $\nu$ during the training). We omit the superscript $(t)$ and write $L(\theta)$ as $L$ here for simplicity.

**Lemma C.1.** *The gradient of $Q$ for the population risk is given by*

$$\nabla_Q L = \mathbb{E}\left[ (\widehat{y}_{query} - \langle w, x_{query}\rangle) \sum_{i,j\in[N]} \mathbf{attn}_i\, \mathbf{attn}_j (E_i^x - E_j^x) E_{N+1}^x{}^\top y_i \right].$$

*Proof.* We obtain:

$$\nabla_Q L = \mathbb{E}[(\widehat{y}_{\text{query}} - \langle w, x_{\text{query}}\rangle) \frac{\partial \widehat{y}_{\text{query}}}{\partial Q}] = \mathbb{E}\left[ (\widehat{y}_{\text{query}} - \langle w, x_{\text{query}}\rangle) \sum_{i\in[N]} \frac{\partial\, \mathbf{attn}_i}{\partial Q} y_i \right].$$

By product rules for matrix functions, we have

$$\frac{\partial\, \mathbf{attn}_i}{\partial Q} = \mathbf{attn}_i \sum_{j\in[N]} \mathbf{attn}_j (E_i^x - E_j^x) E_{N+1}^x{}^\top.$$

Thus

$$\nabla_Q L = \mathbb{E}\left[ (\widehat{y}_{\text{query}} - \langle w_\tau, x_{\text{query}}\rangle) \sum_{i,j\in[N]} \mathbf{attn}_i\, \mathbf{attn}_j (E_i^x - E_j^x) E_{N+1}^x{}^\top y_i \right].$$

$\square$

Recall that the quantities $A_k$ and $B_{k,n}$ are defined in Definition 4.1. These quantities are associated with the attention weights for each token, and they play a crucial role in our analysis of learning dynamics. We will restate their definitions here for clarity.

**Definition C.1.** For $k, n \in [K]$ and $n \neq k$, define the following quantities for $t \geq 0$:
$$A_k^{(t)} := v_k^\top Q^{(t)} v_k \qquad \alpha_k^{(t)} = -v_k^\top \nabla_Q L(Q^{(t)}) v_k$$
$$B_{k,n}^{(t)} := v_n^\top Q^{(t)} v_k \qquad \beta_{k,n}^{(t)} = -v_n^\top \nabla_Q L(Q^{(t)}) v_k$$

By GD update, we have

$$A_k^{(t+1)} := A_k^{(t)} + \eta \alpha_k^{(t)}$$
$$B_{k,n}^{(t+1)} := B_{k,n}^{(t)} + \eta \beta_{k,n}^{(t)}$$

Moreover, by our initialization, since $Q^{(0)} = \mathbf{0}_{d\times d}$, we have $A_k^{(0)} = B_{k,n}^{(0)} = 0$ for any $k, n \in [K]$ with $n \neq k$.

Next, we utilize the expression in Lemma C.1 to compute the gradient projected onto the feature directions, i.e. $\alpha_k^{(t)}$ and $\beta_{k,n}^{(t)}$.

**Lemma C.2.** *For $k, k' \in [K]$, where $k \neq k'$, we have*

$$\alpha_k^{(t)} = \mathbb{E}\left[ \mathbf{1}\{x_{query} = v_k\} \mathbf{Attn}_k^{(t)} \cdot \left( \sum_{m\neq k} \mathbf{Attn}_m^{(t)^2} + (1 - \mathbf{Attn}_k^{(t)})^2 \right) \right]$$

$$\beta_{k,k'}^{(t)} = \mathbb{E}\left[ \mathbf{1}\{x_{query} = v_k\} \mathbf{Attn}_{k'}^{(t)} \cdot \left( \sum_{m\neq k} \mathbf{Attn}_m^{(t)^2} - \mathbf{Attn}_{k'}^{(t)} - \mathbf{Attn}_k^{(t)}(1 - \mathbf{Attn}_k^{(t)}) \right) \right].$$

*Proof.* For any $k, k' \in [K]$, from the previous gradient computation, and note that only when $E_{N+1}^x = x_{\text{query}} = v_k$, we have ${E_{N+1}^x}^\top v_k \neq 0$. Thus,

$$v_{k'}^\top \nabla_Q L v_k$$

$$= \mathbb{E}\left[ \mathbf{1}\{x_{\text{query}} = v_k\} \left(\widehat{y}_{\text{query}} - \langle w, x_{\text{query}} \rangle\right) \sum_{i,j \in [N]} \mathbf{attn}_i \, \mathbf{attn}_j \, y_i v_{k'}^\top (E_i^x - E_j^x) \right]$$

$$= \mathbb{E}\left[ \mathbf{1}\{x_{\text{query}} = v_k\} \left(\widehat{y}_{\text{query}} - \langle w, x_{\text{query}} \rangle\right) \sum_{m,n \in [K]} \sum_{i \in \mathcal{V}_m} \sum_{j \in \mathcal{V}_n} \mathbf{attn}_i \, \mathbf{attn}_j \, y_i v_{k'}^\top (v_m - v_n) \right]$$

$$= \mathbb{E}\left[ \mathbf{1}\{x_{\text{query}} = v_k\} \left(\widehat{y}_{\text{query}} - \langle w, x_{\text{query}} \rangle\right) \sum_{n \in [K]} \sum_{i \in \mathcal{V}_{k'}} \sum_{j \in \mathcal{V}_n} \mathbf{attn}_i \, \mathbf{attn}_j \, y_i v_{k'}^\top (v_{k'} - v_n) \right]$$

$$+ \mathbb{E}\left[ \mathbf{1}\{x_{\text{query}} = v_k\} \left(\widehat{y}_{\text{query}} - \langle w, x_{\text{query}} \rangle\right) \sum_{m \in [K]} \sum_{i \in \mathcal{V}_m} \sum_{j \in \mathcal{V}_{k'}} \mathbf{attn}_i \, \mathbf{attn}_j \, y_i v_{k'}^\top (v_m - v_{k'}) \right]$$

$$= \mathbb{E}\left[ \mathbf{1}\{x_{\text{query}} = v_k\} \left(\widehat{y}_{\text{query}} - \langle w, x_{\text{query}} \rangle\right) \mathbf{Attn}_{k'} \langle w, v_{k'} \rangle \sum_{n \in [K]} \mathbf{Attn}_n \right]$$

$$- \mathbb{E}\left[ \mathbf{1}\{x_{\text{query}} = v_k\} \left(\widehat{y}_{\text{query}} - \langle w, x_{\text{query}} \rangle\right) \mathbf{Attn}_{k'} \sum_{m \in [K]} \mathbf{Attn}_m \langle w, v_m \rangle \right]$$

$$= \mathbb{E}\left[ \mathbf{1}\{x_{\text{query}} = v_k\} \left(\widehat{y}_{\text{query}} - \langle w, x_{\text{query}} \rangle\right) \mathbf{Attn}_{k'} \sum_{m \in [K]} \mathbf{Attn}_m \langle w, v_{k'} - v_m \rangle \right].$$

Note that $\widehat{y}_{\text{query}} = \sum_{i \in [N]} \mathbf{attn}_i \, y_i = \sum_{m \in [K]} \mathbf{Attn}_m \langle w, v_m \rangle$. Thus when $x_{\text{query}} = v_k$, we have

$$\widehat{y}_{\text{query}} - \langle w, x_{\text{query}} \rangle = - \sum_{m \in [K]} \mathbf{Attn}_m \langle w, v_k - v_m \rangle.$$

Plugging this into the above equation, we have

$$v_{k'}^\top \nabla_Q L v_k$$

$$= -\mathbb{E}\left[ \mathbf{1}\{x_{\text{query}} = v_k\} \mathbf{Attn}_{k'} \left( \sum_{n \in [K]} \mathbf{Attn}_n \langle w, v_k - v_n \rangle \right) \left( \sum_{m \in [K]} \mathbf{Attn}_m \langle w, v_{k'} - v_m \rangle \right) \right]$$

$$= -\mathbb{E}\left[ \mathbf{1}\{x_{\text{query}} = v_k\} \mathbf{Attn}_{k'} \left( \sum_{n \in [K]} \sum_{m \in [K]} \mathbf{Attn}_m \, \mathbf{Attn}_n \langle w, v_k - v_n \rangle \langle w, v_{k'} - v_m \rangle \right) \right]$$

$$= -\mathbb{E}\left[ \mathbf{1}\{x_{\text{query}} = v_k\} \mathbf{Attn}_{k'} \left( \sum_{n \in [K]} \sum_{m \in [K]} \mathbf{Attn}_m \, \mathbf{Attn}_n (v_k - v_n)^\top w w^\top (v_{k'} - v_m) \right) \right]$$

$$= -\mathbb{E}\left[ \mathbf{1}\{x_{\text{query}} = v_k\} \mathbf{Attn}_{k'} \left( \sum_{n \in [K]} \sum_{m \in [K]} \mathbf{Attn}_m \, \mathbf{Attn}_n (v_k - v_n)^\top \mathbb{E}[w w^\top \mid P_{\text{input}} \cup \{x_{\text{query}}\}] (v_{k'} - v_m) \right) \right]$$

$$= -\mathbb{E}\left[ \mathbf{1}\{x_{\text{query}} = v_k\} \mathbf{Attn}_{k'} \left( \sum_{n \in [K]} \sum_{m \in [K]} \mathbf{Attn}_m \, \mathbf{Attn}_n (v_k - v_n)^\top (v_{k'} - v_m) \right) \right]$$

$$= -\mathbb{E}\left[ \mathbf{1}\{x_{\text{query}} = v_k\} \mathbf{Attn}_{k'} \left( \left(v_k - \sum_{n \in [K]} \mathbf{Attn}_n \, v_n\right)^\top \left(v_{k'} - \sum_{m \in [K]} \mathbf{Attn}_m \, v_m\right) \right) \right]$$

When $k' = k$, we can obtain

$$
\alpha_k = -v_k^\top \nabla_Q L v_k = \mathbb{E}\left[\mathbf{1}\{x_{\text{query}} = v_k\}\,\mathbf{Attn}_k\,\|v_k - \sum_n \mathbf{Attn}_n\,v_n\|^2\right]
$$

$$
= \mathbb{E}\left[\mathbf{1}\{x_{\text{query}} = v_k\}\,\mathbf{Attn}_k\left((1 - \mathbf{Attn}_k)^2 + \sum_{m \neq k}\mathbf{Attn}_m^2\right)\right].
$$

When $k' \neq k$, we have

$$
\beta_{k,k'} = -v_{k'}^\top \nabla_Q L v_k
$$

$$
= \mathbb{E}\left[\mathbf{1}\{x_{\text{query}} = v_k\}\,\mathbf{Attn}_{k'}\left(\sum_{m \neq k, k'}\mathbf{Attn}_m^2 - \mathbf{Attn}_k(1 - \mathbf{Attn}_k) - \mathbf{Attn}_{k'}(1 - \mathbf{Attn}_{k'})\right)\right]
$$

$$
= \mathbb{E}\left[\mathbf{1}\{x_{\text{query}} = v_k\}\,\mathbf{Attn}_{k'}\left(\sum_{m \neq k}\mathbf{Attn}_m^2 - \mathbf{Attn}_k(1 - \mathbf{Attn}_k) - \mathbf{Attn}_{k'}\right)\right].
$$

$\square$

### C.2 USEFUL PROBABILISTIC LEMMAS FOR PROMPT

Recall that given a prompt $P = (x_1, y_1, \ldots, x_N, y_N, x_{\text{query}})$, we denote $P_{\text{input}}$ as the collection of input tokens, i.e., $\{x_i\}_{i=1}^N$. It's worth noting that, based on our data distribution, the occurrence count of the $k$-th feature in the first $N$ input tokens from $P_{\text{input}}$, denoted as $|\mathcal{V}_k|$, follows a multinomial distribution. Leveraging the concentration property inherent to multinomial distributions, we can identify a set of high-probability events that $P_{\text{input}}$ aligns with. This set constitutes the crux of our subsequent analysis.

We first introduce the following tail bound for multinomial distributions.

**Lemma C.3** (Tail bound of multinomial distribution Devroye (1983)). *Let $(X_1, \cdots, X_K)$ be a multinomial $(N, p_1, \cdots, p_K)$ random vector. For all $\varepsilon \in (0, 1)$ and all $K$ satisfying $K/N \leq \varepsilon^2/20$, we have*

$$
P\left(\sum_{i=1}^K |X_i - \mathbb{E}(X_i)| > N\varepsilon\right) \leq 3\exp\left(-N\varepsilon^2/25\right).
$$

Now we present our characterization of high-probability events for $P_{\text{input}}$.

**Lemma C.4** (High-probability event for balanced data). *Suppose that $p_k = \Theta(\frac{1}{K})$ for any $k \in [K]$ and $K^3 \ll N$. For some constant $c_{bal} \geq \sqrt{\frac{20K^3}{N}}$, let us define*

$$
\mathcal{E}_{bal}^* := \left\{P_{input} : |\mathcal{V}_k| \in \left[p_k N - \frac{c_{bal}N}{K}, p_k N + \frac{c_{bal}N}{K}\right] \text{ for } k \in [K]\right\}.
$$

*Then , we have*

$$
\mathbb{P}(P_{input} \in \mathcal{E}_{bal}^*) \geq 1 - 3\exp\left(-\frac{c_{bal}^2 N}{25K^2}\right).
$$

*Let us denote $L_k^{bal} = p_k K - c_{bal}$ and $U_k^{bal} = p_k K + c_{bal}$. Note that $L_k^{bal}, U_k^{bal}$ are constant level since $p_k = \Theta(\frac{1}{K})$. Then for any event belonging to $\mathcal{E}_{bal}^*$, $|\mathcal{V}_k| \in [\frac{L_k^{bal}N}{K}, \frac{U_k^{bal}N}{K}] = \Theta(\frac{N}{K})$. Note that we can properly choose $c_{bal}$ to guarantee $L_k^{bal} > 0$ for $k \in [K]$.*

*Proof.* Denote $|\mathcal{V}_k| = X_k$. Then $(X_1, \cdots, X_K) \sim$ multinomial $(N, p_1, \cdots, p_K)$. Noting that $\frac{c_{bal}^2}{20K^2} \geq \frac{K}{N}$ by our choice of $c_{bal}$, then letting $\epsilon = \frac{c_{bal}}{K}$, we have $\epsilon^2/20 \geq \frac{K}{N}$. By multinomial tail bound in Lemma C.3, we obtain

$$
P\left(\sum_{i=1}^K |X_i - \mathbb{E}(X_i)| > c_{\text{bal}}\frac{N}{K}\right) \leq 3\exp\left(-\frac{c_{bal}^2 N}{25K^2}\right).
$$

Then, since $\mathbb{E}(X_i) = p_i N$, we have

$$P\left(\cap_{i=1}^{K}\left\{|X_i - p_i N| > \frac{c_{\text{bal}} N}{K}\right\}\right) \leq P\left(\sum_{i=1}^{K}|X_i - \mathbb{E}(X_i)| > c_{\text{bal}}\frac{N}{K}\right)$$

$$\leq 3\exp\left(-\frac{c_{\text{bal}}^2 N}{25K^2}\right).$$

$\square$

**Lemma C.5** (High-probability event for imbalanced data). *Suppose that $p_1 = \Theta(1)$, $p_k = \Theta(\frac{1}{K})$ for $k \in [K] \setminus \{1\}$, and $K^3 \ll N$. Then for some constant $c_{im} \geq \sqrt{\frac{20K^3}{N}}$, there exist constants $U_k^{im} > L_k^{im} > 0$ for any $k \in [K]$, such that letting*

$$\mathcal{E}_{imbal}^* := \left\{P_{input} : |\mathcal{V}_1| \in [L_1^{im}N, U_1^{im}N] \text{ and } |\mathcal{V}_k| \in \left[\frac{L_k^{im}N}{K}, \frac{U_k^{im}N}{K}\right] \text{ for } k \in [K] \setminus \{1\}\right\}$$

*we have*

$$\mathbb{P}(P_{input} \in \mathcal{E}_{imbal}^*) \geq 1 - 3\exp\left(-\frac{c_{im}^2 N}{25K^2}\right).$$

*Proof.* Similarly to the proof for Lemma C.4, we have

$$P\left(\cap_{i=1}^{K}\left\{|X_i - p_i N| > \frac{c_{\text{im}} N}{K}\right\}\right) \leq 3\exp\left(-\frac{c_{\text{im}}^2 N}{25K^2}\right).$$

For $k > 1$, let us denote $L_k^{im} = p_k K - c_{\text{im}}$ and $U_k^{im} = p_k K + c_{\text{im}}$. Since $p_k = \Theta(\frac{1}{K})$, we can easily conclude that $L_k^{im}, U_k^{im}$ for $k > 1$ are constant level. Furthermore, for $k = 1$, let $L_1^{im} = p_1 - 0.01c_{\text{im}}$ and $U_1^{im} = p_1 + 0.01c_{\text{im}}$. Since $p_1 = \Theta(1)$, we have

$$[p_1 N - \frac{c_{\text{im}} N}{K}, p_1 N + \frac{c_{\text{im}} N}{K}] = [(p_1 - \frac{c_{\text{im}}}{K})N, (p_1 + \frac{c_{\text{im}}}{K})p_1 N] \subset [L_1^{im}N, U_1^{im}N]$$

for sufficiently large $K$.

$\square$

### C.3 Properties of Loss Function and Prediction Error

Recall the population loss we consider:

$$L(\theta) = \frac{1}{2}\mathbb{E}\left[(\widehat{y}_{\text{query}} - \langle w, x_{\text{query}}\rangle)^2\right]. \tag{7}$$

In this part, we will present several important lemmas for such training objectives. We first introduce the following lemma, which connects the loss form with the attention score when the query token is a certain feature.

**Lemma C.6** (Loss calculation). *The population loss $L(\theta)$ can be decomposed in the following form:*

$$L(\theta) = \frac{1}{2}\sum_{k=1}^{K}\mathbb{E}\left[\mathbf{1}\{x_{query} = v_k\}\left(\sum_{m \neq k}\mathbf{Attn}_m^2 + (1 - \mathbf{Attn}_k)^2\right)\right]$$

*Proof.* Following the similar calculations as in Lemma C.2, we have

$$L(\theta) = \frac{1}{2}\sum_{k=1}^{K}\mathbb{E}\left[\mathbf{1}\{x_{\text{query}} = v_k\}(\widehat{y}_{\text{query}} - \langle w, x_{\text{query}}\rangle)^2\right]$$

$$= \frac{1}{2}\sum_{k=1}^{K}\mathbb{E}\left[\mathbf{1}\{x_{\text{query}} = v_k\}\left(\sum_{n \in [K]}\mathbf{Attn}_n\langle w_\tau, v_k - v_n\rangle\right)\left(\sum_{m \in [K]}\mathbf{Attn}_m\langle w_\tau, v_k - v_m\rangle\right)\right]$$

$$= \frac{1}{2} \sum_{k=1}^{K} \mathbb{E} \left[ \mathbf{1}\{x_{\text{query}} = v_k\} \| v_k - \sum_n \mathbf{Attn}_n \, v_n \|^2 \right]$$

$$= \frac{1}{2} \sum_{k=1}^{K} \mathbb{E} \left[ \mathbf{1}\{x_{\text{query}} = v_k\} \left( (1 - \mathbf{Attn}_k)^2 + \sum_{m \neq k} \mathbf{Attn}_m^2 \right) \right].$$

$\square$

Then we introduce some additional crucial notations for the loss objectives.

**Notations for the balanced case.**

$$L^* = \min_\theta L(\theta) = \inf_\theta \frac{1}{2} \mathbb{E} \left[ (\widehat{y}_{\text{query}} - \langle w, x_{\text{query}} \rangle)^2 \right], \tag{8}$$

$$L^{\text{low}} = \frac{1}{2}(1 + \frac{1}{K-1}) \sum_{k=1}^{K} \mathbb{P}\left( x_{\text{query}} = v_k \cap |\mathcal{V}_k| = 0 \right). \tag{9}$$

From now on, we'll denote the minimum as $\theta$ in the form $\{1, Q\}$, with our attention solely on the variation of $Q$. Here, $L^*$ denotes the minimum value of the population loss in eq. (7), and $L^{\text{low}}$ represents the sum of unavoidable errors for each $k \in [K]$, given that the query token is the $k$-th feature but has not been seen in the first $N$ training samples. We will show $L^{\text{low}}$ serves as a lower bound for $L^*$, and demonstrate that the network trained with GD will attain nearly zero error compared to $L^{\text{low}}$, which naturally leads to convergence to $L^*$. We further denote

$$L(\theta) = \sum_{k=1}^{K} L_k(\theta),$$

$$\text{where } L_k(\theta) = \frac{1}{2} \mathbb{E} \left[ \mathbf{1}\{x_{\text{query}} = v_k\} (\widehat{y}_{\text{query}} - \langle w, x_{\text{query}} \rangle)^2 \right].$$

$$L_k^{\text{low}} = \frac{1}{2}(1 + \frac{1}{K-1}) \mathbb{P}\left( x_{\text{query}} = v_k \cap |\mathcal{V}_k| = 0 \right),$$

$$\widetilde{L}_k(\theta) = \frac{1}{2} \mathbb{E} \left[ \mathbf{1}\{x_{\text{query}} = v_k \cap P_{\text{input}} \in \mathcal{E}_{\text{bal}}^*\} (\widehat{y}_{\text{query}} - \langle w, x_{\text{query}} \rangle)^2 \right].$$

**Notations for the imbalanced case.** In the unbalanced case, we should be interested in the prediction error for the query corresponding to the given feature $k \in [K]$. Thus we consider the following conditional prediction error for each $k \in [K]$:

$$\mathcal{L}_k(\theta) = \frac{1}{2} \mathbb{E} \left[ (\widehat{y}_{\text{query}} - \langle w, x_{\text{query}} \rangle)^2 \mid x_{\text{query}} = v_k \right]. \tag{10}$$

Similarly, we define the minimum and the unavoidable values for such conditional prediction error:

$$\mathcal{L}_k^* = \min_\theta \frac{1}{2} \mathbb{E} \left[ (\widehat{y}_{\text{query}} - \langle w, x_{\text{query}} \rangle)^2 \mid x_{\text{query}} = v_k \right], \tag{11}$$

$$\mathcal{L}_k^{\text{low}} = \frac{1}{2}(1 + \frac{1}{K-1}) \mathbb{P}\left( |\mathcal{V}_k| = 0 \right), \tag{12}$$

$$\widetilde{\mathcal{L}}_k(\theta) = \frac{1}{2} \mathbb{E} \left[ \mathbf{1}\{P_{\text{input}} \in \mathcal{E}_{\text{imbal}}^*\} (\widehat{y}_{\text{query}} - \langle w, x_{\text{query}} \rangle)^2 \mid x_{\text{query}} = v_k \right].$$

### C.3.1 Loss Characterization for the Balanced Case

**Lemma C.7.** *For $L^*$ and $L^{\text{low}}$ defined in eq. (8) and eq. (9), respectively, then we have $L^{\text{low}} \leq L^*$ and they are both $\Theta(e^{-\text{poly}(K)})$ for the balanced data.*

*Proof.* We first prove $L^{\text{low}} \leq L^*$:

$$L^* = \min_\theta \frac{1}{2} \sum_{k=1}^{K} \mathbb{E} \left[ \mathbf{1}\{x_{\text{query}} = v_k\} (\widehat{y}_{\text{query}} - \langle w, x_{\text{query}} \rangle)^2 \right]$$

$$\geq \min_{\theta} \frac{1}{2} \sum_{k=1}^{K} \mathbb{E}\left[\mathbf{1}\{x_{\text{query}} = v_k \cap |\mathcal{V}_k| = 0\}\left(\widehat{y}_{\text{query}} - \langle w, x_{\text{query}}\rangle\right)^2\right]$$

$$= \min_{\theta} \frac{1}{2} \sum_{k=1}^{K} \mathbb{E}\left[\mathbf{1}\{x_{\text{query}} = v_k \cap |\mathcal{V}_k| = 0\}\left(\sum_{m \neq k} \mathbf{Attn}_m^2 + (1 - \mathbf{Attn}_k)^2\right)\right]$$

Notice that when the query token is the $k$-th feature but has not been seen in the first $N$ training samples, $\mathbf{Attn}_k = 0$. Moreover, $\frac{1}{K-1} \leq \sum_{m \neq k} \mathbf{Attn}_m^2$ by Cauchy–Schwarz inequality. Thus

$$L^* \geq \frac{1}{2}(1 + \frac{1}{K-1}) \sum_{k=1}^{K} \mathbb{E}\left[\mathbf{1}\{x_{\text{query}} = v_k \cap |\mathcal{V}_k| = 0\}\right] = L^{\text{low}}.$$

Furthermore, since $x_{\text{query}}$ and $P_{\text{input}}$ are independently sampled,

$$L^{\text{low}} = K \cdot \Theta(\frac{1}{K}) \cdot (1 - \Theta(\frac{1}{K}))^N.$$

Since $N \gg K^3$, then $(1 - \Theta(\frac{1}{K}))^N = \Theta\left(e^{-\text{poly}(K)}\right)$.

Now we turn to $L^*$, we only need to show $L^* = O(e^{-\text{poly}(K)})$. We have

$$L^* = \min_{\theta}\left(\frac{1}{2} \sum_{k=1}^{K} \mathbb{E}\left[\mathbf{1}\{x_{\text{query}} = v_k \cap |\mathcal{V}_k| > 0\}\left(\sum_{m \neq k} \mathbf{Attn}_m^2 + (1 - \mathbf{Attn}_k)^2\right)\right]\right.$$

$$\left. + \frac{1}{2} \sum_{k=1}^{K} \mathbb{E}\left[\mathbf{1}\{x_{\text{query}} = v_k \cap |\mathcal{V}_k| = 0\}\left(\sum_{m \neq k} \mathbf{Attn}_m^2 + 1\right)\right]\right)$$

Consider $Q = \sigma \mathbf{I}_{d \times d}$, when $x_{\text{query}} = v_k \cap |\mathcal{V}_k| > 0$ holds,

$$\sum_{m \neq k} \mathbf{Attn}_m^2 + (1 - \mathbf{Attn}_k)^2 \leq (1 - \mathbf{Attn}_k) \max_{m \neq k} \mathbf{Attn}_m + (1 - \mathbf{Attn}_k)^2$$

$$\leq 2(1 - \mathbf{Attn}_k)^2 = 2(\frac{N - |\mathcal{V}_k|}{N - |\mathcal{V}_k| + |V_k|e^\sigma})^2 \leq 2(\frac{N}{N + e^\sigma})^2$$

Taking $\sigma = \text{poly}(N)$, then we have

$$L^* \leq O(e^{-\text{poly}(N)}) + O(e^{-\text{poly}(K)}) = O(e^{-\text{poly}(K)}).$$

$\square$

**Lemma C.8.** *For the balanced data, given $k \in [K]$, for any $\theta$, we have*

$$\widetilde{L}_k(\theta) \leq L_k(\theta) - L_k^{low} \leq \widetilde{L}_k(\theta) + 3p_k \exp\left(-\frac{c_{bal}^2 N}{25K^2}\right).$$

*Proof.*

$$L_k(\theta) - \widetilde{L}_k(\theta) = \frac{1}{2}\mathbb{E}\left[\mathbf{1}\{x_{\text{query}} = v_k \cap P_{\text{input}} \in \mathcal{E}_{\text{bal}}^{*}{}^c\}\left(\widehat{y}_{\text{query}} - \langle w_\tau, x_{\text{query}}\rangle\right)^2\right]$$

$$= \frac{1}{2}\mathbb{E}\left[\mathbf{1}\{x_{\text{query}} = v_k \cap P_{\text{input}} \in \mathcal{E}_{\text{bal}}^{*}{}^c\}\left(\sum_{m \neq k} \mathbf{Attn}_m^2 + (1 - \mathbf{Attn}_k)^2\right)\right]$$

$$\overset{(a)}{\leq} \frac{1}{2} \cdot 2\mathbb{P}\left(x_{\text{query}} = v_k \cap P_{\text{input}} \in \mathcal{E}_{\text{bal}}^{*}{}^c\right)$$

$$\overset{(b)}{\leq} p_k \cdot 3\exp\left(-\frac{c_{\text{bal}}^2 N}{25K^2}\right)$$

$$= 3p_k \exp\left(-\frac{c_{\text{bal}}^2 N}{25K^2}\right).$$

where $(a)$ follows the fact that

$$\sum_{m \neq k} \mathbf{Attn}_m^2 + (1 - \mathbf{Attn}_k)^2 \leq (1 - \mathbf{Attn}_k) \max_{m \neq k} \mathbf{Attn}_m + (1 - \mathbf{Attn}_k)^2 \leq 2,$$

and $(b)$ holds by Lemma C.4.

On the other hand,

$$
\begin{aligned}
L_k(\theta) - \widetilde{L}_k(\theta) &\geq \frac{1}{2}\mathbb{E}\left[\mathbf{1}\{x_{\text{query}} = v_k \cap |\mathcal{V}_k| = 1\}\left(\sum_{m \neq k} \mathbf{Attn}_m^2 + (1 - \mathbf{Attn}_k)^2\right)\right] \\
&\geq \frac{1}{2}\frac{K}{K-1}\mathbb{E}\left[\mathbf{1}\{x_{\text{query}} = v_k \cap |\mathcal{V}_k| = 1\}\right] = L_k^{\text{low}}.
\end{aligned}
$$

$\square$

Consequently, for each $k \in [K]$, $\widetilde{L}_k(\theta)$ closely tracks the deviation between $L_k(\theta)$ and $L_k^{\text{low}}$, which is the aspect we will primarily focus on bounding in the subsequent analysis.

### C.3.2 LOSS CHARACTERIZATION FOR THE IMBALANCED CASE

**Lemma C.9.** *Given $k \in [K]$, for $\mathcal{L}_k^*$ and $\mathcal{L}_k^{low}$ defined in eq. (11) and eq. (12), respectively, then we have $\mathcal{L}_k^{low} \leq \mathcal{L}_k^*$ and they are both $\Theta(e^{-\operatorname{poly}(K)})$ for the imbalanced data.*

*Proof.* The analysis is similar as Lemma C.7, we only show $\mathcal{L}_k^{\text{low}} = \Theta(e^{-\operatorname{poly}(K)})$

$$
\begin{aligned}
\mathcal{L}_k^{\text{low}} &= \frac{1}{2}(1 + \frac{1}{K-1})\mathbb{P}\left(|\mathcal{V}_k| = 0\right) \\
&= \Theta(1)(1 - p_k)^N.
\end{aligned}
$$

For $k = 1$, $(1 - p_1)^N = \Theta(\exp(-N)) = \Theta\left(e^{-\operatorname{poly}(K)}\right)$. For $k > 1$, since $N \gg K^3$, then $(1 - p_k)^N = (1 - \Theta(\frac{1}{K}))^N = \Theta\left(e^{-\operatorname{poly}(K)}\right)$, which completes the proof. $\square$

**Lemma C.10.** *For the imbalanced data, given $k \in [K]$, for any $\theta$, we have*

$$\widetilde{\mathcal{L}}_k(\theta) \leq \mathcal{L}_k(\theta) - \mathcal{L}_k^{low} \leq \widetilde{\mathcal{L}}_k(\theta) + 3\exp\left(-\frac{c_{im}^2 N}{25K^2}\right).$$

*Proof.* The proof of the first inequality is similar to Lemma C.8.

$$
\begin{aligned}
\mathcal{L}_k(\theta) - \mathcal{L}_k^{\text{low}} &\leq \widetilde{\mathcal{L}}_k(\theta) + \frac{1}{2}\mathbb{E}\left[\mathbf{1}\{P_{\text{input}} \in \mathcal{E}_{\text{imbal}}^{*}{}^c\}\left(\widehat{y}_{\text{query}} - \langle w, x_{\text{query}}\rangle\right)^2 \mid x_{\text{query}} = v_k\right] \\
&= \widetilde{\mathcal{L}}_k(\theta) + \frac{1}{2}\mathbb{E}\left[\mathbf{1}\{P_{\text{input}} \in \mathcal{E}_{\text{imbal}}^{*}{}^c\}\left(\sum_{m \neq k}\mathbf{Attn}_m^2 + (1 - \mathbf{Attn}_k)^2\right) \mid x_{\text{query}} = v_k\right] \\
&\leq \widetilde{\mathcal{L}}_k(\theta) + \mathbb{P}\left(P_{\text{input}} \in \mathcal{E}_{\text{imbal}}^{*}{}^c\right) \\
&\leq \widetilde{\mathcal{L}}_k(\theta) + 3\exp\left(-\frac{c_{\text{im}}^2 N}{25K^2}\right).
\end{aligned}
$$

$\square$

### C.4 NOTATIONS AND PARAMETERS

In this part, we consolidate the notations introduced throughout the main content and in the preliminary section, summarized in Table 1. Throughout our proofs, we consider $N = \operatorname{poly}(K) \gg K^3$ and $K$ is sufficiently large.

Table 1: Summary of Notations

| Notations | Descriptions |
|---|---|
| $\mathbf{attn}_i^{(t)}$, $\mathbf{Attn}_k^{(t)}$ | The attention scores for the $i$-token and $k$-th feature, where $i \in [N]$ and $k \in [K]$. |
| $A_k^{(t)}$, $B_{k,n}^{(t)}$ | The feature learning quantities when $x_{\text{query}} = v_k$: $A_k^{(t)} = e^{v_k^\top Q^{(t)} v_k}$, $B_{k,n}^{(t)} = e^{v_n^\top Q^{(t)} v_k}$ for $n \neq k$. |
| $\alpha_k^{(t)}$, $\beta_{k,n}^{(t)}$ | Corresponding gradient updates for $A_k^{(t)}$ and $B_{k,n}^{(t)}$. |
| $P_{\text{input}}$ | The input tokens in the prompt, i.e. $\{x_i\}_{i=1}^N$. |
| $\mathcal{E}_{\text{bal}}^*$, $\mathcal{E}_{\text{imbal}}^*$ | The high-probability sets that $P_{\text{input}}$ belongs to for the balanced and imbalanced data. |
| $L^*$, $L^{\text{low}}$ | The minimum value and lower bound of population loss $L(\theta)$ (7). |
| $L_k(\theta)$, $\widetilde{L}_k(\theta)$, $L_k^{\text{low}}$ | The loss functions on the event $\{x_{\text{query}} = v_k\}$, $\{x_{\text{query}} = v_k\} \cap \{P_{\text{input}} \in \mathcal{E}_{\text{bal}}^*\}$, and the lower bound for $L_k$. |
| $\mathcal{L}_k^*$, $\mathcal{L}_k^{\text{low}}$ (**Imbalanced**) | The minimum value and lower bound of prediction error conditioning on $x_{\text{query}} = v_k$, i.e., $\mathcal{L}_k(\theta)$ (10). |
| $\widetilde{\mathcal{L}}_k(\theta)$ (**Imbalanced**) | The conditional prediction error on the event $\{P_{\text{input}} \in \mathcal{E}_{\text{imbal}}^*\}$. |

## D ANALYSIS FOR THE BALANCED CASE

In this section, we present the analysis for the balanced case, we first discuss the outline of our proof.

### D.1 ROADMAP OF THE PROOF

At the beginning of each phase, we will establish an induction hypothesis, which we expect to remain valid throughout that phase. Subsequently, we will analyze the dynamics under such a hypothesis within the phase, aiming to provide proof of the hypothesis by the phase's end.

The main idea of the proof consists of analyzing the GD dynamics of $A_k^{(t)}$ and $B_{k,n}^{(t)}$. From Definition C.1 and lemma C.2, we have

$$A_k^{(t+1)} = A_k^{(t)} + \eta \alpha_k^{(t)},$$
$$B_{k,n}^{(t+1)} = B_{k,n}^{(t)} + \eta \beta_{k,n}^{(t)},$$

and

$$\alpha_k^{(t)} = \mathbb{E}\left[\mathbf{1}\{x_{\text{query}} = v_k\} \mathbf{Attn}_k^{(t)} \cdot \left(\sum_{m \neq k} \mathbf{Attn}_m^{(t)^2} + (1 - \mathbf{Attn}_k^{(t)})^2\right)\right],$$

$$\beta_{k,n}^{(t)} = \mathbb{E}\left[\mathbf{1}\{x_{\text{query}} = v_k\} \mathbf{Attn}_n^{(t)} \cdot \left(\sum_{m \neq k} \mathbf{Attn}_m^{(t)^2} - \mathbf{Attn}_n^{(t)} - \mathbf{Attn}_k^{(t)}(1 - \mathbf{Attn}_k^{(t)})\right)\right].$$

We divide the learning process of any feature $k$ in the balanced case as follows.

- **Phase I** ($t \in [0, T_{1,k}]$, Appendix D.2): At initialization, $A_k^{(t)}$ keeps growing at a rate at least of $\frac{\eta}{K^2}$, while $B_{k,n}^{(t)}$ oscillates with a smaller rate of $\frac{\eta}{K^3}$. Therefore, the increase in $A_k^{(t)}$ will dominate for a while which defines phase I.

- **Phase II** ($t \in (T_{1,k}, T_{2,k}^\epsilon]$, Appendices D.3 and D.4): After rapid growth of self-attention module parameters in phase I, the query token featuring $v_k$ is aligned with the feature $v_k$ itself effectively and disregard other features. Then the process proceeds to convergence

phase, where $A_k^{(t)}$ monotonically increases and $B_{k,n}^{(t)}$ monotonically decreases, which finally contributes to the convergence of loss. Based on the variation rates of $A_k^{(t)}$ and $B_{k,n}^{(t)}$, the convergence phase further has two sub-stages as follows.

- **Stage I** ($t \in (T_{1,k}, \widetilde{T}_{2,k}^\epsilon]$, Appendix D.3): the increase of $A_k^{(t)}$ remains the rate at least $\Omega(\frac{\epsilon}{K})$ while the decrease of $B_{k,n}^{(t)}$ is slim, and the gap $A_k^{(t)} - \max_{m \neq k} B_{k,m}^{(t)}$ stays within $O(\log(\frac{K}{\epsilon^{\frac{1}{2}}}))$.

- **Stage II** ($t \in (\widetilde{T}_{2,k}^\epsilon, T_{2,k}^\epsilon]$, Appendix D.4): the increase of $A_k^{(t)}$ and the decrease of $B_{k,n}^{(t)}$ both are slim and the attention is nearly focused on the target feature, leading to the convergence of loss.

### D.2 PHASE I: GROWTH OF TARGET FEATURE

In this section, we shall discuss the initial phase of learning the correlation between the query tokens and its corresponding feature. Firstly, we present the induction hypothesis in this phase. For $k$-th feature $v_k$, we define the **Phase I** as all iterations $0 \leq t \leq T_{1,k}$, where

$$T_{1,k} \triangleq \max \left\{ t : A_k^{(t)} \leq \log(K) \right\}.$$

We state the following induction hypothesis, which will hold throughout Phase I. This hypothesis is ultimately proved in

**Induction Hypothesis D.1.** *For each $0 \leq t \leq T_{1,k}$, the following holds:*

a. $A_k^{(t)}$ *is monotonically increasing and* $A_k^{(t)} \in [0, \log(K)]$;

b. $|B_{k,n}^{(t)}| = O(\frac{A_k^{(t)}}{K})$ *for any* $n \neq k$.

#### D.2.1 TECHNICAL LEMMAS

We first introduce several technical lemmas that will be used for the proof of Induction Hypothesis D.1.

**Lemma D.1.** *Suppose Induction Hypothesis D.1 holds at iteration $0 \leq t \leq T_{k,1}$. If $x_{query} = v_k$ and $P_{input} \in \mathcal{E}_{bal}^*$, the following holds*

1. $\mathbf{Attn}_k^{(t)} = \Omega(\frac{1}{K})$;

2. $1 - \mathbf{Attn}_k^{(t)} \geq \Omega(1)$.

*Proof.* Since $x_{\text{query}} = v_k$, then we have

$$
\begin{aligned}
\mathbf{Attn}_k^{(t)} &= \frac{|\mathcal{V}_k| e^{v_k^\top Q^{(t)} v_k}}{\sum_{j \in [N]} e^{E_j^{x \top} Q^{(t)} v_k}} \\
&= \frac{|\mathcal{V}_k| \exp(A_k^{(t)})}{\sum_{m \neq k} |\mathcal{V}_m| \exp(B_{k,m}^{(t)}) + |\mathcal{V}_k| \exp(A_k^{(t)})} \\
&= \frac{1}{\sum_{m \neq k} \frac{|\mathcal{V}_m|}{|\mathcal{V}_k|} \exp(B_{k,m}^{(t)} - A_k^{(t)}) + 1}.
\end{aligned}
$$

By Induction Hypothesis D.1, $e^{-\left(\log(K) + O(\frac{\log(K)}{K})\right)} \leq \exp(B_{k,m}^{(t)} - A_k^{(t)}) \leq e^{O(\frac{\log(K)}{K})}$, thus

$$
\mathbf{Attn}_k^{(t)} \geq \frac{1}{e^{O(\frac{\log(K)}{K})}(\frac{N}{|\mathcal{V}_k|} - 1) + 1} \geq \frac{1}{e^{O(\frac{\log(K)}{K})}(K/L_k^{\text{bal}} - 1) + 1} = \Omega(\frac{1}{K}),
$$

where the second inequality follows since the fact that $P_{\text{input}} \in \mathcal{E}_{\text{bal}}^*$.

On the other hand,

$$\mathbf{Attn}_k^{(t)} \leq \frac{1}{e^{-\left(\log(K)+O(\frac{\log(K)}{K})\right)}\left(\frac{N}{|\mathcal{V}_k|}-1\right)+1} \leq \frac{1}{e^{-1}\left(\frac{1}{U_k^{\mathrm{bal}}}-\frac{1}{K}\right)+1}.$$

Consider $U^{\mathrm{bal}} = \Theta(1)$, we have

$$1 - \mathbf{Attn}_k^{(t)} \geq \frac{\left(\frac{1}{U_k^{\mathrm{bal}}}-\frac{1}{K}\right)}{\left(\frac{1}{U_k^{\mathrm{bal}}}-\frac{1}{K}\right)+e} \geq \Omega(1).$$

$\square$

**Lemma D.2.** *Suppose Induction Hypothesis D.1 holds at iteration $0 \leq t \leq T_{1,k}$. If $x_{query} = v_k$ and $P_{input} \in \mathcal{E}_{bal}^*$, for $n \neq k$, the following holds*

1. $\mathbf{Attn}_n^{(t)} = \Theta\left(\frac{1-\mathbf{Attn}_k^{(t)}}{K}\right)$;

2. *Combining with the Lemma D.1, we immediately have $\mathbf{Attn}_n^{(t)} = \Theta(\frac{1}{K})$.*

*Proof.* Since $x_{\mathrm{query}} = v_k$, then we have

$$\mathbf{Attn}_n^{(t)} = \frac{|\mathcal{V}_n|e^{v_n^\top Q^{(t)}v_k}}{\sum_{j\in[N]}e^{E_j^{x\top}Q^{(t)}v_k}}$$

$$= \frac{|\mathcal{V}_n|\exp(B_{k,n}^{(t)})}{\sum_{m\neq k}|\mathcal{V}_m|\exp(B_{k,m}^{(t)})+|\mathcal{V}_k|\exp(A_k^{(t)})}.$$

By Induction Hypothesis D.1, $e^{-O(\frac{\log(K)}{K})} \leq \exp(B_{k,m}^{(t)}-B_{k,n}^{(t)}) \leq e^{O(\frac{\log(K)}{K})}$, combining with the fact that $\frac{|\mathcal{V}_m|}{|\mathcal{V}_n|} = \Theta(1)$ when $P_{\mathrm{input}} \in \mathcal{E}_{\mathrm{bal}}^*$, thus

$$\frac{\mathbf{Attn}_n^{(t)}}{1-\mathbf{Attn}_k^{(t)}} = \frac{|\mathcal{V}_n|\exp(B_{k,n}^{(t)})}{\sum_{m\neq k}|\mathcal{V}_m|\exp(B_{k,m}^{(t)})} = \frac{1}{\sum_{m\neq k}\frac{|\mathcal{V}_m|}{|\mathcal{V}_n|}\exp(B_{k,m}^{(t)}-B_{k,n}^{(t)})} = \Theta\left(\frac{1}{K}\right).$$

$\square$

### D.2.2 CONTROLLING THE GRADIENT UPDATES IN PHASE I

**Lemma D.3.** *Given any fixed $k \in [K]$, if Induction Hypothesis D.1 holds at iteration $0 \leq t \leq T_{1,k}$, then $\alpha_k^{(t)} \geq 0$ and satisfies*

$$\alpha_k^{(t)} \geq \Omega(\frac{1}{K^2}).$$

*Proof.* By gradient computation from Lemma C.2,

$$\alpha_k^{(t)} = \mathbb{E}\left[\mathbf{1}\{x_{\mathrm{query}}=v_k\}\mathbf{Attn}_k^{(t)}\cdot\left(\sum_{m\neq k}\mathbf{Attn}_m^{(t)\,2}+(1-\mathbf{Attn}_k^{(t)})^2\right)\right]$$

$$= \mathbb{E}\left[\mathbf{1}\{x_{\mathrm{query}}=v_k\cap P_{\mathrm{input}}\in\mathcal{E}_{\mathrm{bal}}^*\}\mathbf{Attn}_k^{(t)}\cdot\left(\sum_{m\neq k}\mathbf{Attn}_m^{(t)\,2}+(1-\mathbf{Attn}_k^{(t)})^2\right)\right]$$

$$+ \mathbb{E}\left[\mathbf{1}\{x_{\mathrm{query}}=v_k\cap P_{\mathrm{input}}\in\mathcal{E}_{\mathrm{bal}}^{*\,c}\}\mathbf{Attn}_k^{(t)}\cdot\left(\sum_{m\neq k}\mathbf{Attn}_m^{(t)\,2}+(1-\mathbf{Attn}_k^{(t)})^2\right)\right]$$

$$\overset{(a)}{\geq} p_k\cdot\mathbb{P}(P_{\mathrm{input}}\in\mathcal{E}_{\mathrm{bal}}^*)\times$$

$$\mathbb{E}\left[\mathbf{Attn}_k^{(t)} \cdot \left(\sum_{m \neq k} \mathbf{Attn}_m^{(t)2} + (1 - \mathbf{Attn}_k^{(t)})^2\right) \mid \{x_{\text{query}} = v_k\} \cap \{P_{\text{input}} \in \mathcal{E}_{\text{bal}}^*\}\right]$$

$$\geq p_k \cdot \mathbb{P}(P_{\text{input}} \in \mathcal{E}_{\text{bal}}^*)\mathbb{E}\left[\mathbf{Attn}_k^{(t)} \cdot (1 - \mathbf{Attn}_k^{(t)})^2 \mid \{x_{\text{query}} = v_k\} \cap \{P_{\text{input}} \in \mathcal{E}_{\text{bal}}^*\}\right] \quad (13)$$

$$\overset{(b)}{\geq} \Omega(\frac{1}{K^2}),$$

where $(a)$ follows from the fact that $x_{\text{query}}$ is independent with $P_{\text{input}}$ and the second term is non-negative, $(b)$ follows from Lemma C.4, Lemma D.1 and the fact that $p_k = \Theta(\frac{1}{K})$ in balanced case and $N \gg K^3$. □

**Lemma D.4.** *Given any fixed $k \in [K]$, if Induction Hypothesis D.1 holds at iteration $0 \leq t \leq T_{1,k}$, then for any $n \neq k$, $\beta_{k,n}^{(t)}$ satisfies*

$$|\beta_{k,n}^{(t)}| \leq O(\frac{\alpha_k^{(t)}}{K}).$$

*Proof.* By gradient computation from Lemma C.2, we have

$$\beta_{k,n}^{(t)} \leq \mathbb{E}\left[\mathbf{1}\{x_{\text{query}} = v_k\}\, \mathbf{Attn}_n^{(t)} \cdot \left(\sum_{m \neq k} \mathbf{Attn}_m^{(t)2}\right)\right] \quad (14)$$

$$-\beta_{k,n}^{(t)} \leq \mathbb{E}\left[\mathbf{1}\{x_{\text{query}} = v_k\}\, \mathbf{Attn}_n^{(t)} \cdot \left(\mathbf{Attn}_n^{(t)} + \mathbf{Attn}_k^{(t)}(1 - \mathbf{Attn}_k^{(t)})\right)\right] \quad (15)$$

For eq. (14),

$$\beta_{k,n}^{(t)} \leq \mathbb{E}\left[\mathbf{1}\{x_{\text{query}} = v_k \cap P_{\text{input}} \in \mathcal{E}_{\text{bal}}^*\}\, \mathbf{Attn}_n^{(t)} \cdot \left(\sum_{m \neq k} \mathbf{Attn}_m^{(t)2}\right)\right]$$

$$+ \mathbb{E}\left[\mathbf{1}\{x_{\text{query}} = v_k \cap P_{\text{input}} \in \mathcal{E}_{\text{bal}}^{*c}\}\, \mathbf{Attn}_n^{(t)} \cdot \left(\sum_{m \neq k} \mathbf{Attn}_m^{(t)2}\right)\right]$$

$$\overset{(a)}{\leq} p_k \cdot \mathbb{P}(P_{\text{input}} \in \mathcal{E}_{\text{bal}}^*) \cdot \mathbb{E}\left[\mathbf{Attn}_n^{(t)} \cdot \left(\max_{m \neq k} \mathbf{Attn}_m^{(t)}\right) \mid \{x_{\text{query}} = v_k\} \cap \{P_{\text{input}} \in \mathcal{E}_{\text{bal}}^*\}\right]$$

$$+ p_k \cdot \mathbb{P}(P_{\text{input}} \in \mathcal{E}_{\text{bal}}^{*c})$$

$$\overset{(b)}{\leq} p_k \mathbb{E}\left[\mathbf{Attn}_n^{(t)} \cdot \left(\max_{m \neq k} \mathbf{Attn}_m^{(t)}\right) \mid \{x_{\text{query}} = v_k\} \cap \{P_{\text{input}} \in \mathcal{E}_{\text{bal}}^*\}\right] + 3p_k \exp\left(-\frac{c_{\text{bal}}^2 N}{25 K^2}\right)$$

$$\overset{(c)}{\leq} O(\frac{1}{K^3}), \quad (16)$$

where $(a)$ follows the fact that $x_{\text{query}}$ is independent with $P_{\text{input}}$, $\mathbf{Attn}_n^{(t)} \leq 1$ and $\sum_{m \neq k} \mathbf{Attn}_m^{(t)2} \leq \max_{m \neq k} \mathbf{Attn}_m^{(t)} \cdot \sum_{m \neq k} \mathbf{Attn}_m^{(t)} \leq \max_{m \neq k} \mathbf{Attn}_m^{(t)}$, $(b)$ holds by Lemma C.4, and $(c)$ follows from Lemma D.2 and the fact that $p_k = \Theta(\frac{1}{K})$ and $N \gg K^3$.

For eq. (15), similar to the derivation above, we have

$$- \beta_{k,n}^{(t)}$$

$$\leq p_k \mathbb{E}\left[\mathbf{Attn}_n^{(t)} \cdot \left(\mathbf{Attn}_n^{(t)} + \mathbf{Attn}_k^{(t)}(1 - \mathbf{Attn}_k^{(t)})\right) \mid \{x_{\text{query}} = v_k\} \cap \mathcal{E}_{\text{bal}}^*\right] + p_k \cdot \mathbb{P}(P_{\text{input}} \in \mathcal{E}_{\text{bal}}^{*c})$$

$$\overset{(a)}{=} p_k \cdot \mathbb{P}(P_{\text{input}} \in \mathcal{E}_{\text{bal}}^{*c}) + p_k \cdot \mathbb{P}(P_{\text{input}} \in \mathcal{E}_{\text{bal}}^*) \times$$

$$\mathbb{E}\left[\Theta(\frac{1 - \mathbf{Attn}_k^{(t)}}{K}) \cdot \left(\Theta(\frac{1 - \mathbf{Attn}_k^{(t)}}{K}) + \mathbf{Attn}_k^{(t)}(1 - \mathbf{Attn}_k^{(t)})\right) \mid \{x_{\text{query}} = v_k\} \cap \mathcal{E}_{\text{bal}}^*\right]$$

$$\overset{(b)}{\leq} p_k \cdot \mathbb{P}(P_{\text{input}} \in \mathcal{E}_{\text{bal}}^*) \mathbb{E}\left[O(\frac{\mathbf{Attn}_k^{(t)}(1-\mathbf{Attn}_k^{(t)})^2}{K}) \mid \{x_{\text{query}} = v_k\} \cap \mathcal{E}_{\text{bal}}^*\right] + 6p_k \exp\left(-\frac{c_{\text{bal}}^2 N}{25K^2}\right)$$

$$\overset{(c)}{\leq} O\left(\frac{\alpha_k^{(t)}}{K} + \frac{1}{K}\exp\left(-\frac{c_{\text{bal}}^2 N}{25K^2}\right)\right) \tag{17}$$

where $(a)$ holds by Lemma D.2 and $(b)$ follows from Lemma D.1 and Lemma C.4, and and $(c)$ follows from Equation (13).

From Lemma D.3 and the choice of $N \gg K^3$, we have

$$\alpha_k^{(t)} \geq \Omega(\frac{1}{K^2}) \gg 6\exp\left(-\frac{c_{\text{bal}}^2 N}{25K^2}\right) \tag{18}$$

Thus, combine Equations (16) to (18), we have

$$|\beta_{n,k}^{(t)}| \leq \max\{O(\frac{\alpha_k^{(t)}}{K}), O(\frac{1}{K^3})\} = O(\frac{\alpha_k^{(t)}}{K}).$$

$\square$

### D.2.3 AT THE END OF PHASE I

**Lemma D.5.** *Given any fixed $k \in [K]$, Induction Hypothesis D.1 holds for all iterations $0 \leq t \leq T_{1,k}$, where $T_{1,k}$ is at most $O(\frac{\log(K)K^2}{\eta})$, and at iteration $t = T_{1,k}+1$, we have*

a. $A_k^{(T_{1,k}+1)} \geq \log(K)$;

b. $\mathbf{Attn}_k^{(T_{1,k}+1)} = \Omega(1)$ *if $x_{query} = v_k$ and $P_{input} \in \mathcal{E}_{bal}^*$.*

*Proof.* If Induction Hypothesis D.1 holds, the existence of $T_{1,k} = O(\frac{\log(K)K^2}{\eta})$ is directly implied by Lemma D.3.

We move to prove Induction Hypothesis D.1. It is easy to verify Induction Hypothesis D.1 holds at $t = 0$. Now we suppose Induction Hypothesis D.1 holds for all iterations $\leq t-1$, and prove it holds at $t$.

By Lemma D.3, we have $\alpha_k^{(t-1)} \geq 0$. Thus $A_k^{(t)} = A_k^{(t-1)} + \eta\alpha_k^{(t-1)} \geq 0$. Moreover, by the definition of $T_{1,k}$, we immediately obtain $A_k^{(t)} \leq \log(K)$.

By Lemma D.4, we have $|\beta_{k,n}^{(t-1)}| \leq O(\frac{\alpha_k^{(t-1)}}{K})$. Thus,

$$|B_{k,n}^{(t)}| \leq |B_{k,n}^{(t-1)}| + \eta O(\frac{\alpha_k^{(t-1)}}{K})$$

$$\leq O(\frac{A_k^{(t-1)}}{K}) + \eta O(\frac{\alpha_k^{(t-1)}}{K})$$

$$\leq O(\frac{A_k^{(t)}}{K}).$$

Moreover, $\mathbf{Attn}_k^{(T_{1,k}+1)} = \Omega(1)$ can be derived from Lemma D.6.

$\square$

After rapid growth of self-attention module parameters in phase I, the query token featuring $v_k$ is aligned with the feature $v_k$ itself effectively and disregards other features. Then the process proceeds to the convergence phase, where $A_k^{(t)}$ monotonically increases and $B_{k,n}^{(t)}$ monotonically decreases, which finally contributes to the convergence of loss. Based on the variation rates of $A_k^{(t)}$ and $B_{k,n}^{(t)}$, the convergence phase further has two sub-stages as follows.

## D.3 PHASE II: CONVERGENCE: STAGE I

Given any $0 < \epsilon < \frac{1}{2}$ for $k \in [K]$, define

$$\widetilde{T}_{2,k}^{\epsilon} := \max \left\{ t > T_{1,k} : A_k^{(t)} - \max_{m \neq k} B_{k,m}^{(t)} \leq \log \left( (\frac{K}{L_k^{\text{bal}}} - 1)((\frac{3}{\epsilon})^{\frac{1}{2}} - 1) \right) \right\}.$$

**Induction Hypothesis D.2.** For $T_{1,k} < t \leq \widetilde{T}_{2,k}^{\epsilon}$, suppose $\text{polylog}(K) \gg \log(\frac{1}{\epsilon})$, the following holds

a. $A_k^{(t)}$ is monotonically increasing and $A_k^{(t)} \in [\log(K), O(\log(K/\epsilon))]$;

b. $B_{k,n}^{(t)}$ is monotonically decreasing and $|B_{k,n}^{(t)}| = O(\frac{A_k^{(t)}}{K})$ for any $n \neq k$.

### D.3.1 TECHNICAL LEMMAS

We first introduce several technical lemmas that will be used for the proof of Induction Hypothesis D.2.

**Lemma D.6.** *Suppose Induction Hypothesis D.2 holds at iteration $T_{1,k} < t \leq \widetilde{T}_{2,k}^{\epsilon}$, if $x_{query} = v_k$ and $P_{input} \in \mathcal{E}_{bal}^*$, the following holds*

*1.* $\textbf{Attn}_k^{(t)} = \Omega(1)$;

*2.* $(1 - \textbf{Attn}_k^{(t)})^2 \geq \Omega(\epsilon) = \Omega(\exp(-\text{polylog}(K)))$.

*Proof.* Since $x_{\text{query}} = v_k$, then we have

$$\textbf{Attn}_k^{(t)} = \frac{|\mathcal{V}_k| \exp(A_k^{(t)})}{\sum_{m \neq k} |\mathcal{V}_m| \exp(B_{k,m}^{(t)}) + |\mathcal{V}_k| \exp(A_k^{(t)})}$$

$$= \frac{1}{\sum_{m \neq k} \frac{|\mathcal{V}_m|}{|\mathcal{V}_k|} \exp(B_{k,m}^{(t)} - A_k^{(t)}) + 1}.$$

By Induction Hypothesis D.2, we obtain

$$\exp(B_{k,m}^{(t)} - A_k^{(t)}) \leq e^{O(\frac{\log(K/\epsilon)}{K}) - \log(K)} \leq e^{O(\frac{\log(K) + \text{polylog}(K)}{K}) - \log(K)} \leq O(\frac{1}{K}).$$

Therefore,

$$\textbf{Attn}_k^{(t)} \geq \frac{1}{O(\frac{1}{K})(\frac{N}{|\mathcal{V}_k|} - 1) + 1} \geq \frac{1}{O(\frac{1}{L_k^{\text{bal}}} - \frac{1}{K}) + 1} \geq \Omega(1).$$

On the other hand, by the definition of $\widetilde{T}_{2,k}^{\epsilon}$, we have

$$1 - \textbf{Attn}_k^{(t)} = \frac{\sum_{m \neq k} \frac{|\mathcal{V}_m|}{|\mathcal{V}_k|} \exp(B_{k,m}^{(t)} - A_k^{(t)})}{\sum_{m \neq k} \frac{|\mathcal{V}_m|}{|\mathcal{V}_k|} \exp(B_{k,m}^{(t)} - A_k^{(t)}) + 1}$$

$$\geq \frac{\exp(\min_{m \neq k} B_{k,m}^{(t)} - A_k^{(t)})(\frac{N}{|\mathcal{V}_k|} - 1)}{\exp(\min_{m \neq k} B_{k,m}^{(t)} - A_k^{(t)})(\frac{N}{|\mathcal{V}_k|} - 1) + 1}$$

$$\geq \frac{\exp(\min_{m \neq k} B_{k,m}^{(t)} - A_k^{(t)})(\frac{K}{U_k^{\text{bal}}} - 1)}{\exp(\min_{m \neq k} B_{k,m}^{(t)} - A_k^{(t)})(\frac{K}{U_k^{\text{bal}}} - 1) + 1}$$

$$= \frac{\exp(\max_{m \neq k} B_{k,m}^{(t)} - A_k^{(t)} - \Delta B_k^{(t)})(\frac{K}{U_k^{\text{bal}}} - 1)}{\exp(\max_{m \neq k} B_{k,m}^{(t)} - A_k^{(t)} - \Delta B_k^{(t)})(\frac{K}{U_k^{\text{bal}}} - 1) + 1}$$

$$\geq \frac{(\frac{K}{L_k^{\text{bal}}} - 1)^{-1}(\epsilon^{-\frac{1}{2}} - 1)^{-1} \cdot e^{-O(\frac{\text{polylog}(K)}{K})}(\frac{K}{U_k^{\text{bal}}} - 1)}{(\frac{K}{L_k^{\text{bal}}} - 1)^{-1}(\epsilon^{-\frac{1}{2}} - 1)^{-1} e^{-O(\frac{\text{polylog}(K)}{K})}(\frac{K}{U_k^{\text{bal}}} - 1) + 1}$$

$$\geq \Omega(\epsilon^{\frac{1}{2}}),$$

where $\Delta B_k^{(t)} = \max_{m\neq k} B_{k,m}^{(t)} - \min_{m\neq k} B_{k,m}^{(t)} = O(\frac{A_k^{(t)}}{K})$, and the first and second inequalities follow from the fact that $\frac{x}{1+x}$ monotonically increases w.r.t. $x \geq 0$, and the third inequality follows from the definition of $\widetilde{T}_{2,k}^{\epsilon}$ and Induction Hypothesis D.2. $\qquad\square$

**Lemma D.7.** *Suppose Induction Hypothesis D.2 holds at iteration $T_{1,k} < t \leq \widetilde{T}_{2,k}^{\epsilon}$, if $x_{query} = v_k$ and $P_{input} \in \mathcal{E}_{bal}^*$, for $n \neq k$, the following holds*

$$\mathbf{Attn}_n^{(t)} = \Theta\left(\frac{1 - \mathbf{Attn}_k^{(t)}}{K}\right).$$

*Proof.* By definition,

$$\mathbf{Attn}_n^{(t)} = \frac{|\mathcal{V}_n| \exp(B_{k,n}^{(t)})}{\sum_{m\neq k} |\mathcal{V}_m| \exp(B_{k,m}^{(t)}) + |\mathcal{V}_k| \exp(A_k^{(t)})}.$$

By Induction Hypothesis D.2, $e^{-O(\frac{\log(K)-\log(\epsilon)}{K})} \leq \exp(B_{k,m}^{(t)} - B_{k,n}^{(t)}) \leq e^{O(\frac{\log(K)-\log(\epsilon)}{K})}$, combining the fact that $-\log(\epsilon) \ll \mathrm{polylog}(K)$ thus

$$\frac{\mathbf{Attn}_n^{(t)}}{1 - \mathbf{Attn}_k^{(t)}} = \frac{|\mathcal{V}_n| \exp(B_{k,n}^{(t)})}{\sum_{m\neq k} |\mathcal{V}_m| \exp(B_{k,m}^{(t)})} = \frac{1}{\sum_{m\neq k} \frac{|\mathcal{V}_m|}{|\mathcal{V}_n|} \exp(B_{k,m}^{(t)} - B_{k,n}^{(t)})} = \Theta(\frac{1}{K}).$$

$\qquad\square$

### D.3.2 CONTROLLING THE GRADIENT UPDATES IN STAGE I OF PHASE II

**Lemma D.8.** *At each iteration $T_{1,k} < t \leq \widetilde{T}_{2,k}^{\epsilon}$, if Induction Hypothesis D.2 holds then $\alpha_k^{(t)} \geq 0$ and satisfies*

$$\alpha_k^{(t)} \geq \Omega(\frac{\epsilon}{K}).$$

*Proof.* The analysis is similar as Lemma D.3, but we need to be more careful about the lower bound of $1 - \mathbf{Attn}_k^{(t)}$. By gradient computation from Lemma C.2,

$$\alpha_k^{(t)} = \mathbb{E}\left[\mathbf{1}\{x_{\mathrm{query}} = v_k\}\, \mathbf{Attn}_k^{(t)} \cdot \left(\sum_{m\neq k} \mathbf{Attn}_m^{(t)2} + (1 - \mathbf{Attn}_k^{(t)})^2\right)\right]$$

$$\geq p_k \cdot \mathbb{P}(P_{\mathrm{input}} \in \mathcal{E}_{\mathrm{bal}}^*)\mathbb{E}\left[\mathbf{Attn}_k^{(t)} \cdot \left(\sum_{m\neq k} \mathbf{Attn}_m^{(t)2} + (1 - \mathbf{Attn}_k^{(t)})^2\right) \mid \{x_{\mathrm{query}} = v_k\} \cap \mathcal{E}_{\mathrm{bal}}^*\right]$$

$$\geq p_k \cdot \mathbb{P}(P_{\mathrm{input}} \in \mathcal{E}_{\mathrm{bal}}^*)\mathbb{E}\left[\mathbf{Attn}_k^{(t)} \cdot (1 - \mathbf{Attn}_k^{(t)})^2 \mid \{x_{\mathrm{query}} = v_k\} \cap \mathcal{E}_{\mathrm{bal}}^*\right]$$

$$\geq \Omega(\frac{\epsilon}{K}),$$

where the last inequality follows from Lemmas C.4 and D.6 and the fact that $p_k = \Theta(\frac{1}{K})$ in balanced case. $\qquad\square$

**Lemma D.9.** *At each iteration $T_{1,k} < t \leq \widetilde{T}_{2,k}^{\epsilon}$, if Induction Hypothesis D.2 holds then given $k \in [K]$, for any $n \neq k$, $\beta_{k,n}^{(t)}$ satisfies*

$$-O(\frac{\alpha_k^{(t)}}{K}) \leq \beta_{k,n}^{(t)} \leq 0.$$

*Proof.* Note that conditioning on the event $\{x_{\text{query}} = v_k\} \cap \{P_{\text{input}} \in \mathcal{E}_{\text{bal}}^*\}$, by Lemmas D.6 and D.7, we have $\mathbf{Attn}_k^{(t)} = \Omega(1)$, $\max_{m \neq k} \mathbf{Attn}_m = O(\frac{1}{K})$, thus

$$\sum_{m \neq k} \mathbf{Attn}^{(t)}{}_m^2 - \mathbf{Attn}_n^{(t)} - \mathbf{Attn}_k^{(t)}(1 - \mathbf{Attn}_k^{(t)}) \leq \max_{m \neq k} \mathbf{Attn}_m^{(t)} \sum_{m \neq k} \mathbf{Attn}_m^{(t)} - \mathbf{Attn}_k^{(t)}(1 - \mathbf{Attn}_k^{(t)})$$

$$= -(1 - \mathbf{Attn}_k^{(t)})(\mathbf{Attn}_k^{(t)} - \max_{m \neq k} \mathbf{Attn}_m^{(t)})$$

$$\leq -\Omega(1 - \mathbf{Attn}_k^{(t)}). \tag{19}$$

Therefore, combine with Lemma C.2

$$\beta_{k,n}^{(t)} \leq \mathbb{E}\left[\mathbf{1}\{x_{\text{query}} = v_k \cap \mathcal{E}_{\text{bal}}^*\} \, \mathbf{Attn}_n^{(t)} \cdot \left(\sum_{m \neq k} \mathbf{Attn}_m^{(t)}{}^2 - \mathbf{Attn}_n^{(t)} - \mathbf{Attn}_k^{(t)}(1 - \mathbf{Attn}_k^{(t)})\right)\right]$$

$$+ \mathbb{E}\left[\mathbf{1}\{x_{\text{query}} = v_k \cap \mathcal{E}_{\text{bal}}^*{}^c\} \, \mathbf{Attn}_n^{(t)} \cdot \left(\sum_{m \neq k} \mathbf{Attn}_m^{(t)2}\right)\right]$$

$$\overset{(a)}{\leq} p_k \cdot \mathbb{P}(P_{\text{input}} \in \mathcal{E}_{\text{bal}}^*) \cdot \mathbb{E}\left[-\Omega(\frac{(1 - \mathbf{Attn}_k^{(t)})^2}{K}) \mid \{x_{\text{query}} = v_k\} \cap \mathcal{E}_{\text{bal}}^*\right] + p_k \cdot \mathbb{P}(\mathcal{E}_{\text{bal}}^*{}^c)$$

$$\overset{(b)}{\leq} p_k \cdot \left(-\Omega(\frac{\epsilon}{K})\right) + 3p_k \exp\left(-\frac{c_{\text{bal}}^2 N}{25K^2}\right)$$

$$\leq 0,$$

where $(a)$ follows from Equation (19), Lemma D.7, $(b)$ follows from Lemmas C.4 and D.6, and the last inequality holds since

$$\frac{\epsilon}{K} \gg \frac{\exp(-\text{polylog}(K))}{K} \gg \exp\left(-\frac{c_{\text{bal}}^2 N}{25K^2}\right).$$

Moreover, following the similar analysis from Lemma D.4, we have

$$-\beta_{k,n}^{(t)} \leq p_k \mathbb{E}\left[\mathbf{Attn}_n^{(t)} \cdot \left(\mathbf{Attn}_n^{(t)} + \mathbf{Attn}_k^{(t)}(1 - \mathbf{Attn}_k^{(t)})\right) \mid \{x_{\text{query}} = v_k\} \cap \mathcal{E}_{\text{bal}}^*\right] + p_k \mathbb{P}(\mathcal{E}_{\text{bal}}^*{}^c)$$

$$\leq p_k \mathbb{E}\left[\Theta(\frac{1 - \mathbf{Attn}_k^{(t)}}{K}) \cdot O\left(\mathbf{Attn}_k^{(t)}(1 - \mathbf{Attn}_k^{(t)})\right) \mid \{x_{\text{query}} = v_k\} \cap \mathcal{E}_{\text{bal}}^*\right]$$

$$+ 6p_k \exp\left(-\frac{c_{\text{bal}}^2 N}{25K^2}\right)$$

$$= p_k \mathbb{E}\left[O(\frac{\mathbf{Attn}_k^{(t)}(1 - \mathbf{Attn}_k^{(t)})^2}{K}) \mid \{x_{\text{query}} = v_k\} \cap \mathcal{E}_{\text{bal}}^*\right] + 6p_k \exp\left(-\frac{c_{\text{bal}}^2 N}{25K^2}\right)$$

$$\leq O(\frac{\alpha_k^{(t)}}{K}).$$

$\square$

### D.3.3 AT THE END OF STAGE I OF PHASE II

**Lemma D.10.** *Given $k \in [K]$, and $0 < \epsilon < 1$, suppose $\text{polylog}(K) \gg \log(\frac{1}{\epsilon})$, then Induction Hypothesis D.2 holds for at least all $T_{1,k} < t \leq \widetilde{T}_{2,k}^\epsilon = T_{1,k} + O(\frac{K \log(K\epsilon^{-\frac{1}{2}})}{\eta\epsilon})$, and at iteration $t = \widetilde{T}_{2,k}^\epsilon + 1$, we have $A_k^{(\widetilde{T}_{2,k}^\epsilon + 1)} \geq \Omega(\log(\frac{K}{\epsilon}))$.*

*Proof.* We first prove the existence of $\widetilde{T}_{2,k}^\epsilon$. Recall that

$$\widetilde{T}_{2,k}^\epsilon := \max\left\{t > T_{1,k} : A_k^{(t)} - \max_{m \neq k} B_{k,m}^{(t)} \leq \log\left((\frac{K}{L_k^{\text{bal}}} - 1)((\frac{3}{\epsilon})^{\frac{1}{2}} - 1)\right)\right\}.$$

When $t \in (T_{1,k}, \widetilde{T}^{\epsilon}_{2,k}]$, consider

$$
\left( A_k^{(t+1)} - \max_{m \neq k} B_{k,m}^{(t+1)} \right) - \left( A_k^{(t)} - \max_{m \neq k} B_{k,m}^{(t)} \right)
$$

$$
= \eta \alpha_k^{(t)} - \left( \max_{m \neq k} B_{k,m}^{(t+1)} - \max_{m \neq k} B_{k,m}^{(t)} \right)
$$

$$
\geq \eta \alpha_k^{(t)} - \max_{m \neq k} \left( B_{k,m}^{(t+1)} - B_{k,m}^{(t)} \right)
$$

$$
= \eta \alpha_k^{(t)} - \max_{m \neq k} \left( \eta \beta_{k,m}^{(t)} \right) \geq \eta \alpha_k^{(t)} = \Omega(\frac{\eta \epsilon}{K})),
$$

where the last inequality follows from Lemma D.9 and the last equation follows from Lemma D.8. Therefore, at most $\widetilde{T}^{\epsilon}_{2,k} - T_{1,k} = O(\frac{K \log\left( (\frac{K}{L_k^{\text{bal}}} - 1)((\frac{3}{\epsilon})^{\frac{1}{2}} - 1) \right)}{\eta \epsilon}) = O(\frac{K \log(K \epsilon^{-\frac{1}{2}})}{\eta \epsilon})$ iterations are needed before $A_k^{(t)} - \max_{m \neq k} B_{k,m}^{(t)}$ exceeds $\log\left( (\frac{K}{L_k^{\text{bal}}} - 1)((\frac{3}{\epsilon})^{\frac{1}{2}} - 1) \right)$.

It is easy to verify Induction Hypothesis D.2 holds at $t = T_{1,k} + 1$. Now we suppose Induction Hypothesis D.2 holds for all iterations in $[T_{1,k} + 1, t - 1]$, and prove it holds at $t$.

By Lemma D.8, we have $\alpha_k^{(t-1)} \geq 0$. Thus $A_k^{(t)} \geq A_k^{(t-1)} \geq \log(K)$. By Lemma D.9, we have $-O(\frac{\alpha_k^{(t-1)}}{K}) \leq \beta_{k,n}^{(t-1)} \leq 0$. Thus,

$$
|B_{k,n}^{(t)}| \leq |B_{k,n}^{(t-1)}| + \eta O(\frac{\alpha_k^{(t-1)}}{K})
$$

$$
\leq O(\frac{A_k^{(t-1)}}{K}) + \eta O(\frac{\alpha_k^{(t-1)}}{K})
$$

$$
\leq O(\frac{A_k^{(t)}}{K}).
$$

Moreover, by the definition of $\widetilde{T}^{\epsilon}_{2,k}$, for any $T_{1,k} < t \leq \widetilde{T}^{\epsilon}_{2,k}$ we immediately have

$$
(1 - O(\frac{1}{K})) A_k^{(t)} \leq A_k^{(t)} - \max_{m \neq k} B_{k,m}^{(t)} \leq \log\left( (\frac{K}{L_k^{\text{bal}}} - 1)((\frac{3}{\epsilon})^{\frac{1}{2}} - 1) \right).
$$

Therefore, $A_k^{(t)} \leq O(\log(\frac{K}{\epsilon}))$ for any $T_{1,k} < t \leq \widetilde{T}^{\epsilon}_{2,k}$.

At iteration $t = \widetilde{T}^{\epsilon}_{2,k} + 1$, we have $A_k^{(\widetilde{T}^{\epsilon}_{2,k}+1)} - \max_{m \neq k} B_{k,m}^{(\widetilde{T}^{\epsilon}_{2,k}+1)} > \log\left( (\frac{K}{L_k^{\text{bal}}} - 1)((\frac{3}{\epsilon})^{\frac{1}{2}} - 1) \right)$, thus $A_k^{(\widetilde{T}^{\epsilon}_{2,k}+1)} \geq \Omega(\log(\frac{K}{\epsilon}))$.

When $\{x_{\text{query}} = v_k\} \cap \{P_{\text{input}} \in \mathcal{E}^*_{\text{bal}}\}$, we obtain

$$
1 - \mathbf{Attn}_k^{(\widetilde{T}^{\epsilon}_{2,k}+1)} = \frac{\sum_{m \neq k} \frac{|\mathcal{V}_m|}{|\mathcal{V}_k|} \exp(B_{k,m}^{(t)} - A_k^{(t)})}{\sum_{m \neq k} \frac{|\mathcal{V}_m|}{|\mathcal{V}_k|} \exp(B_{k,m}^{(t)} - A_k^{(t)}) + 1}
$$

$$
\leq \frac{\exp(\max_{m \neq k} B_{k,m}^{(t)} - A_k^{(t)})(\frac{N}{|\mathcal{V}_k|} - 1)}{\exp(\max_{m \neq k} B_{k,m}^{(t)} - A_k^{(t)})(\frac{N}{|\mathcal{V}_k|} - 1) + 1}
$$

$$
\leq \frac{\exp(\max_{m \neq k} B_{k,m}^{(t)} - A_k^{(t)})(\frac{K}{L_k^{\text{bal}}} - 1)}{\exp(\max_{m \neq k} B_{k,m}^{(t)} - A_k^{(t)})(\frac{K}{L_k^{\text{bal}}} - 1) + 1}
$$

$$\leq \frac{\left(\left(\frac{K}{L_k^{\text{bal}}} - 1\right)\left(\left(\frac{3}{\epsilon}\right)^{\frac{1}{2}} - 1\right)\right)^{-1}\left(\frac{K}{L_k^{\text{bal}}} - 1\right)}{\left(\left(\frac{K}{L_k^{\text{bal}}} - 1\right)\left(\left(\frac{3}{\epsilon}\right)^{\frac{1}{2}} - 1\right)\right)^{-1}\left(\frac{K}{L_k^{\text{bal}}} - 1\right) + 1}$$

$$= (\epsilon/3)^{\frac{1}{2}},$$

where the first inequality follows from the fact that $\frac{x}{1+x}$ monotonically increases w.r.t. $x \geq 0$.

$\square$

### D.4 PHASE II: CONVERGENCE: STAGE II

Given $k \in [K]$, define

$$T_{2,k}^\epsilon := \widetilde{T}_{2,k}^\epsilon + O\left(\frac{K \log\left(K\epsilon^{-\frac{1}{2}}\right)}{\epsilon\eta}\right).$$

**Induction Hypothesis D.3.** Suppose $\text{polylog}(K) \gg \log(\frac{1}{\epsilon})$, for $t \in (\widetilde{T}_{2,k}^\epsilon, T_{2,k}^\epsilon]$, the following holds:

    a. $A_k^{(t)}$ is monotonically increasing but cannot exceed $O(\log(K/\epsilon))$;

    b. $B_{k,m}^{(t)}$ is monotonically decreasing and $|B_{k,m}^{(t)}| = O(\frac{A_k^{(t)}}{K})$ for any $m \neq k$;

#### D.4.1 TECHNICAL LEMMAS

We first introduce several technical lemmas that will be used for the proof of Induction Hypothesis D.3.

**Lemma D.11.** *Suppose Induction Hypothesis D.3 holds at iteration $t \in (\widetilde{T}_{2,k}^\epsilon, T_{2,k}^\epsilon]$, if $x_{query} = v_k$ and $P_{input} \in \mathcal{E}_{bal}^*$, the following holds*

    *1. $\textbf{Attn}_k^{(t)} = \Omega(1)$;*

    *2. $(1 - \textbf{Attn}_k^{(t)})^2 \in [\Omega(\exp(-\text{polylog}(K))), \epsilon)$, where $c > 0$ is some constant.*

*Proof.* Since $x_{\text{query}} = v_k$, then we have

$$\textbf{Attn}_k^{(t)} = \frac{|\mathcal{V}_k| \exp(A_k^{(t)})}{\sum_{m \neq k} |\mathcal{V}_m| \exp(B_{k,m}^{(t)}) + |\mathcal{V}_k| \exp(A_k^{(t)})}$$

$$= \frac{1}{\sum_{m \neq k} \frac{|\mathcal{V}_m|}{|\mathcal{V}_k|} \exp(B_{k,m}^{(t)} - A_k^{(t)}) + 1}.$$

By Induction Hypothesis D.3,

$$\exp(B_{k,m}^{(t)} - A_k^{(t)}) \leq e^{O(\frac{\log(K/\epsilon)}{K}) - \log(K)} \leq e^{O(\frac{\log(K) + \text{polylog}(K)}{K}) - \log(K)} \leq O\left(\frac{1}{K}\right).$$

Therefore,

$$\textbf{Attn}_k^{(t)} \geq \frac{1}{O(\frac{1}{K})(\frac{N}{|\mathcal{V}_k|} - 1) + 1} \geq \frac{1}{O(\frac{1}{L_k^{\text{bal}}} - \frac{1}{K}) + 1} \geq \Omega(1).$$

We first upper bound $1 - \textbf{Attn}_k^{(t)}$,

$$1 - \textbf{Attn}_k^{(t)} = \frac{\sum_{m \neq k} \frac{|\mathcal{V}_m|}{|\mathcal{V}_k|} \exp(B_{k,m}^{(t)} - A_k^{(t)})}{\sum_{m \neq k} \frac{|\mathcal{V}_m|}{|\mathcal{V}_k|} \exp(B_{k,m}^{(t)} - A_k^{(t)}) + 1}$$

$$\leq \frac{\exp(\max_{m \neq k} B_{k,m}^{(t)} - A_k^{(t)})(\frac{N}{|\mathcal{V}_k|} - 1)}{\exp(\max_{m \neq k} B_{k,m}^{(t)} - A_k^{(t)})(\frac{N}{|\mathcal{V}_k|} - 1) + 1}$$

$$\overset{(a)}{\leq} \frac{\exp(\max_{m \neq k} B_{k,m}^{(\widetilde{T}_{2,k}^\epsilon + 1)} - A_k^{(\widetilde{T}_{2,k}^\epsilon + 1)})(\frac{N}{|\mathcal{V}_k|} - 1)}{\exp(\max_{m \neq k} B_{k,m}^{(\widetilde{T}_{2,k}^\epsilon + 1)} - A_k^{(\widetilde{T}_{2,k}^\epsilon + 1)})(\frac{N}{|\mathcal{V}_k|} - 1) + 1}$$

$$\overset{(b)}{<} (\frac{\epsilon}{3})^{\frac{1}{2}},$$

where $(a)$ holds since $\max_{m \neq k} B_{k,m}^{(t)} - A_k^{(t)}$ is non-increasing by Induction Hypothesis D.3, and $(b)$ follows from the definition of $\widetilde{T}_{2,k}^\epsilon$.

Then we lower bound $1 - \mathbf{Attn}_k^{(t)}$ following the similar analysis from Lemma D.6:

$$1 - \mathbf{Attn}_k^{(t)} = \frac{\sum_{m \neq k} \frac{|\mathcal{V}_m|}{|\mathcal{V}_k|} \exp(B_{k,m}^{(t)} - A_k^{(t)})}{\sum_{m \neq k} \frac{|\mathcal{V}_m|}{|\mathcal{V}_k|} \exp(B_{k,m}^{(t)} - A_k^{(t)}) + 1}$$

$$\geq \frac{\exp(\min_{m \neq k} B_{k,m}^{(t)} - A_k^{(t)})(\frac{N}{|\mathcal{V}_k|} - 1)}{\exp(\min_{m \neq k} B_{k,m}^{(t)} - A_k^{(t)})(\frac{N}{|\mathcal{V}_k|} - 1) + 1}$$

$$\geq \frac{\exp(\min_{m \neq k} B_{k,m}^{(t)} - A_k^{(t)})(\frac{K}{U_k^{\text{bal}}} - 1)}{\exp(\min_{m \neq k} B_{k,m}^{(t)} - A_k^{(t)})(\frac{K}{U_k^{\text{bal}}} - 1) + 1}$$

$$\geq \frac{\frac{1}{e^{O(\log(K/\epsilon))}}(\frac{K}{U_k^{\text{bal}}} - 1)}{\frac{1}{e^{O(\log(K/\epsilon))}}(\frac{K}{U_k^{\text{bal}}} - 1) + 1}$$

$$\geq \frac{\frac{1}{e^{O(\text{polylog}(K))}}(\frac{K}{U_k^{\text{bal}}} - 1)}{\frac{1}{e^{O(\text{polylog}(K))}}(\frac{K}{U_k^{\text{bal}}} - 1) + 1}$$

$$\geq \Omega(\exp(-\text{polylog}(K))),$$

where the first three inequalities follow from the fact that $\frac{x}{1+x}$ monotonically increases w.r.t. $x \geq 0$ and $-A_k^{(t)} \geq O(\log(K/\epsilon))$. $\qquad \square$

**Lemma D.12.** *Suppose Induction Hypothesis D.3 holds at iteration* $t \in (\widetilde{T}_{2,k}^\epsilon, T_{2,k}^\epsilon]$, *if* $x_{query} = v_k$ *and* $P_{input} \in \mathcal{E}_{bal}^*$, *for* $n \neq k$, *the following holds*

$$\mathbf{Attn}_n^{(t)} = \Theta\left(\frac{1 - \mathbf{Attn}_k^{(t)}}{K}\right).$$

*Proof.* By definition,

$$\mathbf{Attn}_n^{(t)} = \frac{|\mathcal{V}_n| \exp(B_{k,n}^{(t)})}{\sum_{m \neq k} |\mathcal{V}_m| \exp(B_{k,m}^{(t)}) + |\mathcal{V}_k| \exp(A_k^{(t)})}.$$

By Induction Hypothesis D.3, $e^{-O(\frac{\log(K) - \log(\epsilon)}{K})} \leq \exp(B_{k,m}^{(t)} - B_{k,n}^{(t)}) \leq e^{O(\frac{\log(K) - \log(\epsilon)}{K})}$, combining the fact that $-\log(\epsilon) \ll \text{polylog}(K)$ thus

$$\frac{\mathbf{Attn}_n^{(t)}}{1 - \mathbf{Attn}_k^{(t)}} = \frac{|\mathcal{V}_n| \exp(B_{k,n}^{(t)})}{\sum_{m \neq k} |\mathcal{V}_m| \exp(B_{k,m}^{(t)})} = \frac{1}{\sum_{m \neq k} \frac{|\mathcal{V}_m|}{|\mathcal{V}_n|} \exp(B_{k,m}^{(t)} - B_{k,n}^{(t)})} = \Theta(\frac{1}{K}).$$

$\qquad \square$

### D.4.2 Controlling the Gradient Updates in Stage II of Phase II

**Lemma D.13.** *At each iteration* $t \in (\widetilde{T}_{2,k}^\epsilon, T_{2,k}^\epsilon]$, *if Induction Hypothesis D.3 holds for t, then* $\alpha_k^{(t)} \geq 0$ *and satisfies*

$$\alpha_k^{(t)} \leq O(\frac{\epsilon}{K}).$$

*Proof.* By gradient computation from Lemma C.2,

$$
\alpha_k^{(t)} = \mathbb{E}\left[\mathbf{1}\{x_{\text{query}} = v_k\} \, \mathbf{Attn}_k^{(t)} \cdot \left(\sum_{m\neq k} \mathbf{Attn}_m^{(t)^2} + (1 - \mathbf{Attn}_k^{(t)})^2\right)\right]
$$

$$
\leq p_k \mathbb{E}\left[\mathbf{Attn}_k^{(t)} \cdot \left(\sum_{m\neq k} \mathbf{Attn}_m^{(t)^2} + (1 - \mathbf{Attn}_k^{(t)})^2\right) \mid \{x_{\text{query}} = v_k\} \cap \mathcal{E}_{\text{bal}}^*\right]
$$

$$
+ 6p_k \exp\left(-\frac{c_{\text{bal}}^2 N}{25K^2}\right)
$$

$$
\leq p_k \cdot O(\epsilon) + 6p_k \exp\left(-\frac{c_{\text{bal}}^2 N}{25K^2}\right)
$$

$$
\leq O(\frac{\epsilon}{K}),
$$

where the second inequality follows from Lemmas D.11 and D.12, and the last inequality follows from the fact that $p_k = \Theta(\frac{1}{K})$ and $\epsilon = \Omega(\exp(-\operatorname{polylog}(K))) \gg 6\exp\left(-\frac{c_{\text{bal}}^2 N}{25K^2}\right)$. $\quad\square$

**Lemma D.14.** *At each iteration* $t \in (\widetilde{T}_{2,k}^\epsilon, T_{2,k}^\epsilon]$, *if Induction Hypothesis D.3 holds for t, for any* $n \neq k, \beta_{k,n}^{(t)}$ *satisfies*

$$
-O(\frac{\alpha_k^{(t)}}{K}) \leq \beta_{k,n}^{(t)} \leq 0.
$$

*Proof.* Note that conditioning on the event $\{x_{\text{query}} = v_k\} \cap \{P_{\text{input}} \in \mathcal{E}_{\text{bal}}^*\}, \mathbf{Attn}_k^{(t)} = \Omega(1)$, and $\max_{m\neq k} \mathbf{Attn}_m = O(\frac{\epsilon^{\frac{1}{2}}}{K})$ thus

$$
\sum_{m\neq k} \mathbf{Attn}_m^2 - \mathbf{Attn}_n - \mathbf{Attn}_k(1 - \mathbf{Attn}_k) \leq \max_{m\neq k} \mathbf{Attn}_m \sum_{m\neq k} \mathbf{Attn}_m - \mathbf{Attn}_k(1 - \mathbf{Attn}_k)
$$

$$
= -(1 - \mathbf{Attn}_k)(\mathbf{Attn}_k - \max_{m\neq k} \mathbf{Attn}_m)
$$

$$
\leq -\Omega(1 - \mathbf{Attn}_k) \leq -\Omega(\exp(-\operatorname{polylog}(K))).
$$

Therefore, by gradient computation from Lemma C.2 and $N \gg K^3$,

$$
\beta_{k,n}^{(t)} \leq 6\exp\left(-\frac{c_{\text{bal}}^2 N}{25K^2}\right) - \Omega(\exp(-\operatorname{polylog}(K))) < 0.
$$

Moreover, following the similar analysis from Lemma D.9, we have

$$
-\beta_{k,n}^{(t)} \leq p_k \mathbb{E}\left[\mathbf{Attn}_n^{(t)} \cdot \left(\mathbf{Attn}_n^{(t)} + \mathbf{Attn}_k^{(t)}(1 - \mathbf{Attn}_k^{(t)})\right) \mid \{x_{\text{query}} = v_k\} \cap \mathcal{E}_{\text{bal}}^*\right] + p_k \mathbb{P}(\mathcal{E}_{\text{bal}}^{*\,c})
$$

$$
\leq p_k \mathbb{E}\left[\Theta(\frac{1 - \mathbf{Attn}_k^{(t)}}{K}) \cdot O\left(\mathbf{Attn}_k^{(t)}(1 - \mathbf{Attn}_k^{(t)})\right) \mid \{x_{\text{query}} = v_k\} \cap \mathcal{E}^*\right]
$$

$$
+ 6p_k \exp\left(-\frac{c_{\text{bal}}^2 N}{25K^2}\right)
$$

$$
= p_k \mathbb{E}\left[O(\frac{\mathbf{Attn}_k^{(t)}(1 - \mathbf{Attn}_k^{(t)})^2}{K}) \mid \{x_{\text{query}} = v_k\} \cap \mathcal{E}^*\right] + 6p_k \exp\left(-\frac{c_{\text{bal}}^2 N}{25K^2}\right)
$$

$$
\leq O(\frac{\alpha_k^{(t)}}{K}),
$$

where the last inequality holds since the gradient computation of $\alpha_k^{(t)}$ from Lemma C.2 and $\alpha_k^{(t)} \gg 6\exp\left(-\frac{c_{\text{bal}}^2 N}{25K^2}\right)$. $\quad\square$

### D.4.3 Controlling the Loss in Stage II of Phase II

**Lemma D.15.** *Given $k \in [K]$, and $0 < \epsilon < 1$, suppose $\mathrm{polylog}(K) \gg \log(\frac{1}{\epsilon})$. At each iteration $t \in (\widetilde{T}_{2,k}^{\epsilon}, T_{2,k}^{\epsilon}]$, if Induction Hypothesis D.3 holds for t, then we have $\widetilde{L}_k(\theta^{(t)}) < \frac{p_k \epsilon}{2}$.*

*Proof.* By gradient computation from Lemma C.2,

$$
\begin{aligned}
\widetilde{L}_k(\theta^{(t)}) &= \frac{1}{2}\mathbb{E}\left[\mathbf{1}\{x_{\text{query}} = v_k \cap P_{\text{input}} \in \mathcal{E}_{\text{bal}}^*\}\left(\widehat{y}_{\text{query}} - \langle w, x_{\text{query}}\rangle\right)^2\right] \\
&= \frac{1}{2}\mathbb{E}\left[\mathbf{1}\{x_{\text{query}} = v_k \cap P_{\text{input}} \in \mathcal{E}_{\text{bal}}^*\}\left(\sum_{m \neq k}\mathbf{Attn}_m^{(t)}{}^2 + (1 - \mathbf{Attn}_k^{(t)})^2\right)\right] \\
&\leq \frac{1}{2}p_k\mathbb{P}\left(P_{\text{input}} \in \mathcal{E}_{\text{bal}}^*\right)\cdot\mathbb{E}\left[(O(\frac{1}{K})+1)(1 - \mathbf{Attn}_k^{(t)})^2 \mid x_{\text{query}} = v_k \cap P_{\text{input}} \in \mathcal{E}_{\text{bal}}^*\right] \\
&\leq \frac{1}{2}p_k\cdot(1 + O(\frac{1}{K}))\cdot\epsilon \\
&\leq \frac{2p_k\epsilon}{3},
\end{aligned}
$$

where the first inequality follows from Lemma D.12, and the second inequality follows from Lemma D.11. $\qquad\square$

### D.4.4 At the end of Stage II of Phase II

**Lemma D.16.** *Given $k \in [K]$, and $0 < \epsilon < 1$, suppose $\mathrm{polylog}(K) \gg \log(\frac{1}{\epsilon})$, then Induction Hypothesis D.3 holds for at least all $\widetilde{T}_{2,k}^{\epsilon} < t \leq T_{2,k}^{\epsilon} = \widetilde{T}_{2,k}^{\epsilon} + O(\frac{K\log\left(K\epsilon^{-\frac{1}{2}}\right)}{\epsilon\eta})$.*

*Proof.* It is easy to verify Induction Hypothesis D.3 holds at $t = \widetilde{T}_{2,k}^{\epsilon} + 1$. Now we suppose Induction Hypothesis D.3 holds for all iterations $\widetilde{T}_{2,k}^{\epsilon} \leq t - 1$, and prove it holds at $t$.

For the first claim, we can upper bound the update of $A_k^{(t)}$ by Lemma D.13 as follows:

$$
\begin{aligned}
A_k^{(t)} &\leq A_k^{(t-1)} + \eta\cdot O(\frac{\epsilon}{K}) \\
&\leq A_k^{(\widetilde{T}_{2,k}^{\epsilon}+1)} + \eta(t - \widetilde{T}_{2,k}^{\epsilon} - 1)\cdot O(\frac{\epsilon}{K}) \\
&\leq O(\log(K/\epsilon)) + \eta O(\frac{K\log\left(K\epsilon^{-\frac{1}{2}}\right)}{\epsilon\eta})\cdot O(\frac{\epsilon}{K}) \\
&= O(\log(K/\epsilon)).
\end{aligned}
$$

The second claim is concluded by Lemma D.14 and similar analysis from Lemma D.10.

$\qquad\square$

### D.5 Putting all together: Proof of Main Theorem for Balanced Feature

**Theorem D.1** (Restate of Theorem 3.1 for balanced feature). *Suppose $p_k = \Theta(\frac{1}{K})$ for each $k \in [K]$. For any $0 < \epsilon < 1$, suppose $N \geq \mathrm{poly}(K)$ and $\mathrm{polylog}(K) \gg \log(\frac{1}{\epsilon})$. We apply GD to train the loss function given in eq. (4). Then with at most $T^* = O(\frac{\log(K)K^2}{\eta} + \frac{K\log\left(K\epsilon^{-\frac{1}{2}}\right)}{\epsilon\eta})$ iterations, we have*

1. *The loss converges: $L(\theta^{(T^*)}) - L^* \leq \epsilon$, where $L^* = \Theta(e^{-\mathrm{poly}(K)})$ is the global minimum of the population loss in eq. (4).*

2. *Attention score concentrates: if $x_{query} = v_k$, with probability at least $1 - e^{-\Omega(\text{poly}(K))}$[4], the one-layer transformer nearly "pays all attention" to input tokens featuring $v_k$, i.e., $(1 - \mathbf{Attn}_k^{(T^*)})^2 \leq O(\epsilon)$.*

*Proof.* Denote $T^* = \max_{k \in [K]} \widetilde{T}_{2,k}^\epsilon + 1 = O(\frac{\log(K)K^2}{\eta} + \frac{K \log\left(K\epsilon^{-\frac{1}{2}}\right)}{\epsilon\eta})$. Thus for any $k$, at iteration $T^*$, it is in stage II of the convergence phase, i.e. $T^* \in (\widetilde{T}_{2,k}^\epsilon, T_{2,k}^\epsilon]$. Then by Lemmas D.15 and D.16, for any $k \in [K]$, we obtain:

$$\widetilde{L}_k(\theta^{(T^*)}) \leq \frac{2p_k\epsilon}{3}.$$

Therefore

$$
\begin{aligned}
L(\theta^{(T^*)}) - L^{\text{low}} &= \sum_{k=1}^{K} (L_k(\theta^{(T^*)}) - L_k^{\text{low}}) \\
&\leq \sum_{k=1}^{K} \left( \widetilde{L}_k(\theta^{(T^*)}) + 6p_k \exp\left(-\frac{c_{\text{bal}}^2 N}{25K^2}\right) \right) \\
&\leq \sum_{k=1}^{K} \frac{2p_k\epsilon}{3} + 6 \exp\left(-\frac{c_{\text{bal}}^2 N}{25K^2}\right) \\
&\leq \frac{2\epsilon}{3} + 6 \exp\left(-\frac{c_{\text{bal}}^2 N}{25K^2}\right) \\
&\leq \epsilon,
\end{aligned}
$$

where the first inequality follows from Lemma C.8.

Finally, by Lemma C.7,

$$L(\theta^{(T^*)}) - L^* \leq L(\theta^{(T^*)}) - L^{\text{low}} \leq \epsilon.$$

$\square$

---

[4]The randomness originates from the first $N$ input tokens in the test prompt.

# E  ANALYSIS FOR THE IMBALANCED CASE: UNDER-REPRESENTED FEATURES

In this section, we present the analysis for the under-represented feature $v_k$ with $k > 1$ in the imbalanced case, we first discuss the outline of our proof.

## E.1  ROADMAP OF THE PROOF

At the beginning of each phase, we will establish an induction hypothesis, which we expect to remain valid throughout that phase. Subsequently, we will analyze the dynamics under such a hypothesis within the phase, aiming to provide proof of the hypothesis by the phase's end.

The main idea of the proof consists of analyzing the GD dynamics of $A_k^{(t)}$ and $B_{k,n}^{(t)}$. From Definition C.1 and lemma C.2, we have

$$A_k^{(t+1)} = A_k^{(t)} + \eta \alpha_k^{(t)},$$
$$B_{k,n}^{(t+1)} = B_{k,n}^{(t)} + \eta \beta_{k,n}^{(t)},$$

and

$$\alpha_k^{(t)} = \mathbb{E}\left[\mathbf{1}\{x_{\text{query}} = v_k\} \, \mathbf{Attn}_k^{(t)} \cdot \left(\sum_{m \neq k} \mathbf{Attn}_m^{(t)2} + (1 - \mathbf{Attn}_k^{(t)})^2\right)\right],$$

$$\beta_{k,n}^{(t)} = \mathbb{E}\left[\mathbf{1}\{x_{\text{query}} = v_k\} \, \mathbf{Attn}_n^{(t)} \cdot \left(\sum_{m \neq k} \mathbf{Attn}_m^{(t)2} - \mathbf{Attn}_n^{(t)} - \mathbf{Attn}_k^{(t)}(1 - \mathbf{Attn}_k^{(t)})\right)\right].$$

We divide the under-represented feature $v_k$ with $k > 1$ learning process as follows.

- **Phase I** ($t \in [0, T_{1,k}]$, Appendix E.2): At initialization, $B_{k,1}^{(t)}$ enjoys a much larger reduction rate, i.e. $\beta_{k,1} < 0$ and $|\beta_{k,1}|$ is large. Therefore, the decrease in $B_{k,1}^{(t)}$ will dominate for a while which defines phase I.

- **Phase II** ($t \in (T_{1,k}, T_{2,k}]$, Appendix E.3): At time $T_{2,k} + 1$, the reduction in the value of $B_{k,1}^{(t)}$ causes a corresponding decrease in $|\beta_{k,1}^{(t)}|$, bringing it closer to $\alpha_k^{(t)}$. This marks the inception of phase II. Shortly after entering this phase, the previous assurance of $B_{k,1}^{(t)}$ reduction dominance wanes, as $|\beta_{k,1}^{(t)}|$ approaches a comparable order of magnitude to $\alpha_k^{(t)}$. At this juncture, there is a shift in the leading influence, with the prominence of $A_k^{(t)}$ growth taking over.

- **Phase III** ($t \in (T_{2,k}, T_{3,k}]$, Appendix E.4): Following the transitional phase, $\alpha_k^{(t)}$ grows at approximately $\Theta(\frac{1}{K^{1.5}})$, whereas $|\beta_{k,1}^{(t)}|$ and $|\beta_{k,n}^{(t)}|$ for $n \neq k, 1$ stay at much lower values ($\leq O(\frac{1}{K^{1.98}})$ and $\leq O(\frac{1}{K^3})$ respectively). This consistent gap in magnitude between $\alpha_k^{(t)}$ and $\beta_{k,n}^{(t)}$ leads to the continued rapid growth of $A_k^{(t)}$, while $B_{k,n}^{(t)}$ remains relatively stable.

- **Phase IV** ($t \in (T_{3,k}, T_{4,k}^\epsilon]$, Appendix E.5): At $t = T_{3,k} + 1$, we achieve the desired attention structures for query tokens linked to the under-represented feature $v_k$. We proceed to demonstrate that the dominant growth rate of $A_k^{(t)}$ will gradually slow down and become governed by the loss, leading to the subsequent proof of convergence.

## E.2  PHASE I: DECREASE OF DOMINANT FEATURE

In this section, we will delve into the initial phase of learning dynamics, aimed at mitigating the high occurrence bias of the dominant feature $v_1$. Specifically, for $k > 1$, $B_{k,1}$ will undergo significant reduction during this phase. Let's begin by defining phase I.

For the $k$-th feature $v_k$ with $k > 1$, we define the phase I as all iterations $t \leq T_{1,k}$, where

$$T_{1,k} \triangleq \max\left\{t : B_{k,1}^{(t)} \geq -0.49\log(K)\right\}.$$

We state the following induction hypothesis, which will hold throughout phase I:

**Induction Hypothesis E.1.** Given $k > 1$, for each $0 \leq t \leq T_{1,k}$, the following holds:

a. $A_k^{(t)}$ is monotonically increasing and $A_k^{(t)} \in [0, O(\frac{\log(K)}{K^{0.02}})]$;

b. $B_{k,1}^{(t)}$ is monotonically decreasing and $B_{k,1}^{(t)} \in [-0.49 \log(K), 0]$;

c. $|B_{k,n}^{(t)}| = O(\frac{A_k^{(t)} - B_{k,1}^{(t)}}{K})$ and $B_{k,n}^{(t)} > B_{k,1}^{(t)}$ for any $n \neq k, 1$.

### E.2.1 TECHNICAL LEMMAS

We first introduce several technical lemmas that will be used for the proof of Induction Hypothesis E.1.

**Lemma E.1.** *If Induction Hypothesis E.1 holds at iteration $0 \leq t \leq T_{1,k}$, for the prompt satisfies $x_{query} = v_k$ and $P_{input} \in \mathcal{E}_{imbal}^*$, the following holds*

1. $\mathbf{Attn}_k^{(t)} = \Theta(\frac{1}{K})$;

2. $\mathbf{Attn}_1^{(t)} = \Omega(\frac{1}{K^{0.49}})$;

3. $1 - \mathbf{Attn}_1^{(t)} - \mathbf{Attn}_k^{(t)} \geq \Omega(1)$;

*Proof.* Since $x_{\text{query}} = v_k$, and notice that $|\mathcal{V}_k| > 0$ for $P_{\text{input}} \in \mathcal{E}_{\text{imbal}}^*$, we have

$$
\mathbf{Attn}_k^{(t)} = \frac{|\mathcal{V}_k| e^{v_k^\top Q^{(t)} v_k}}{\sum_{j \in [N]} e^{E_j^{x \top} Q^{(t)} v_k}}
$$

$$
= \frac{|\mathcal{V}_k| \exp(A_k^{(t)})}{\sum_{m \neq k} |\mathcal{V}_m| \exp(B_{k,m}^{(t)}) + |\mathcal{V}_k| \exp(A_k^{(t)})}
$$

$$
= \frac{1}{\sum_{m \neq k} \frac{|\mathcal{V}_m|}{|\mathcal{V}_k|} \exp(B_{k,m}^{(t)} - A_k^{(t)}) + 1}
$$

By Induction Hypothesis E.1, we have

- for $m \neq 1, k$, $e^{-O(\frac{\log(K)}{K^{0.02}})} \leq \exp(B_{k,m}^{(t)} - A_k^{(t)}) \leq e^{O(\frac{\log(K)}{K})}$;

- for $m = 1$, $e^{(-0.49 \log(K) - O(\frac{\log(K)}{K^{0.02}}))} \leq \exp(B_{k,1}^{(t)} - A_k^{(t)}) \leq e^0$.

Combining with the fact that $\sum_{m \neq k} \frac{|\mathcal{V}_m|}{|\mathcal{V}_k|} = \Theta(K)$ for $P_{\text{input}} \in \mathcal{E}_{\text{imbal}}^*$, thus

$$
\mathbf{Attn}_k^{(t)} \geq \Omega(\frac{1}{K}).
$$

On the other hand, since $\frac{N - |\mathcal{V}_1|}{|\mathcal{V}_k|}$ is still $\Theta(K)$, we have

$$
\mathbf{Attn}_k^{(t)} \leq \frac{1}{e^{-O(\frac{\log(K)}{K^{0.02}})} (\frac{N - |\mathcal{V}_1|}{|\mathcal{V}_k|} - 1) + e^{(-0.49 \log(K) - O(\frac{\log(K)}{K^{0.02}}))} \frac{|\mathcal{V}_1|}{|\mathcal{V}_k|} + 1} \leq O(\frac{1}{K}).
$$

Then we turn to $\mathbf{Attn}_1^{(t)}$. Similarly,

$$
\mathbf{Attn}_1^{(t)} = \frac{|\mathcal{V}_1| \exp(B_{k,1}^{(t)})}{\sum_{m \neq k} |\mathcal{V}_m| \exp(B_{k,m}^{(t)}) + |\mathcal{V}_k| \exp(A_k^{(t)})}
$$

$$
= \frac{1}{\sum_{m \neq 1, k} \frac{|\mathcal{V}_m|}{|\mathcal{V}_1|} \exp(B_{k,m}^{(t)} - B_{k,1}^{(t)}) + \frac{|\mathcal{V}_k|}{|\mathcal{V}_1|} \exp(A_k^{(t)} - B_{k,1}^{(t)}) + 1}
$$

By Induction Hypothesis E.1,

- for $m \neq 1, k$, we have $e^0 \leq \exp(B_{k,m}^{(t)} - B_{k,1}^{(t)}) \leq e^{0.49 \log(K) + O(\frac{\log(K)}{K})}$;

- $e^0 \leq \exp(A_k^{(t)} - B_{k,1}^{(t)}) \leq e^{0.49 \log(K) + O(\frac{\log(K)}{K})}$

Hence,

$$\mathbf{Attn}_1^{(t)} \geq \frac{1}{e^{0.49 \log(K) + O(\frac{\log(K)}{K})}(\frac{N}{|\mathcal{V}_1|} - 1) + 1} \geq \Omega(\frac{1}{K^{0.49}}).$$

where the last inequality holds since $\frac{N}{|\mathcal{V}_1|} = \Theta(1)$ for $P_{\text{input}} \in \mathcal{E}_{\text{imbal}}^*$.

For the last statement,

$$1 - \mathbf{Attn}_1^{(t)} \geq \frac{e^0(\frac{N}{|\mathcal{V}_1|} - 1)}{e^0(\frac{N}{|\mathcal{V}_1|} - 1) + 1} \geq \Omega(1).$$

Combining the fact that $\mathbf{Attn}_k^{(t)} = \Theta(\frac{1}{K})$, we have

$$1 - \mathbf{Attn}_k^{(t)} - \mathbf{Attn}_1^{(t)} \geq \Omega(1).$$

$\square$

**Lemma E.2.** *If Induction Hypothesis E.1 holds at iteration $0 \leq t \leq T_{1,k}$, for the prompt satisfies $x_{query} = v_k$ and $P_{input} \in \mathcal{E}_{imbal}^*$, the following holds*

$$\mathbf{Attn}_n^{(t)} = O\left(\frac{1 - \mathbf{Attn}_k^{(t)} - \mathbf{Attn}_1^{(t)}}{K}\right).$$

*Proof.* Since $x_{\text{query}} = v_k$, we have

$$\mathbf{Attn}_n^{(t)} = \frac{|\mathcal{V}_n| e^{v_n^\top Q^{(t)} v_k}}{\sum_{j \in [N]} e^{E_j^{x^\top} Q^{(t)} v_k}}$$

$$= \frac{|\mathcal{V}_n| \exp(B_{k,n}^{(t)})}{\sum_{m \neq k} |\mathcal{V}_m| \exp(B_{k,m}^{(t)}) + |\mathcal{V}_k| \exp(A_k^{(t)})}$$

By Induction Hypothesis E.1, for $m, n \neq 1$, $e^{-O(\frac{\log(K)}{K})} \leq \exp(B_{k,m}^{(t)} - B_{k,n}^{(t)}) \leq e^{O(\frac{\log(K)}{K})}$, combining with the fact that $\frac{|\mathcal{V}_m|}{|\mathcal{V}_n|} = \Theta(1)$ for $P_{\text{input}} \in \mathcal{E}_{\text{imbal}}^*$, thus

$$\frac{\mathbf{Attn}_n^{(t)}}{1 - \mathbf{Attn}_k^{(t)} - \mathbf{Attn}_1^{(t)}} = \frac{|\mathcal{V}_n| \exp(B_{k,n}^{(t)})}{\sum_{m \neq 1,k} |\mathcal{V}_m| \exp(B_{k,m}^{(t)})}$$

$$= \frac{1}{\sum_{m \neq k,1} \frac{|\mathcal{V}_m|}{|\mathcal{V}_n|} \exp(B_{k,m}^{(t)} - B_{k,n}^{(t)})}$$

$$\leq O(\frac{1}{K}).$$

$\square$

### E.2.2 Controlling the Gradient Updates in Phase I

**Lemma E.3.** *Given $k > 1$, if Induction Hypothesis E.1 holds at iteration $0 \leq t \leq T_{1,k}$, then $\alpha_k^{(t)} \geq 0$ and satisfies*

$$\alpha_k^{(t)} = \Theta(\frac{1}{K^2}).$$

*Proof.* By gradient computation from Lemma C.2, we have

$$
\alpha_k^{(t)} = \mathbb{E}\left[\mathbf{1}\{x_{\text{query}} = v_k\}\, \mathbf{Attn}_k^{(t)} \cdot \left(\sum_{m \neq k} \mathbf{Attn}_m^{(t)\,2} + (1 - \mathbf{Attn}_k^{(t)})^2\right)\right]
$$

$$
= \mathbb{E}\left[\mathbf{1}\{x_{\text{query}} = v_k \cap \mathcal{E}_{\text{imbal}}^*\}\, \mathbf{Attn}_k^{(t)} \cdot \left(\sum_{m \neq k} \mathbf{Attn}_m^{(t)\,2} + (1 - \mathbf{Attn}_k^{(t)})^2\right)\right]
$$

$$
+ \mathbb{E}\left[\mathbf{1}\{x_{\text{query}} = v_k \cap {\mathcal{E}_{\text{imbal}}^*}^c\}\, \mathbf{Attn}_k^{(t)} \cdot \left(\sum_{m \neq k} \mathbf{Attn}_m^{(t)\,2} + (1 - \mathbf{Attn}_k^{(t)})^2\right)\right]
$$

$$
\overset{(a)}{\leq} p_k \cdot \mathbb{P}(P_{\text{input}} \in \mathcal{E}_{\text{imbal}}^*)\mathbb{E}\left[\mathbf{Attn}_k^{(t)} \cdot \left(\sum_{m \neq k} \mathbf{Attn}_m^{(t)\,2} + (1 - \mathbf{Attn}_k^{(t)})^2\right) \mid \{x_{\text{query}} = v_k\} \cap \mathcal{E}_{\text{imbal}}^*\right]
$$

$$
+ 2p_k \cdot \mathbb{P}(P_{\text{input}} \in {\mathcal{E}_{\text{imbal}}^*}^c)
$$

$$
\overset{(b)}{\leq} p_k \cdot \mathbb{E}\left[\mathbf{Attn}_k^{(t)} \cdot \left(O(\tfrac{1}{K}) + \mathbf{Attn}_1^{(t)\,2} + (1 - \mathbf{Attn}_k^{(t)})^2\right) \mid \{x_{\text{query}} = v_k\} \cap \mathcal{E}_{\text{imbal}}^*\right]
$$

$$
+ 2p_k \cdot \mathbb{P}(P_{\text{input}} \in {\mathcal{E}_{\text{imbal}}^*}^c)
$$

$$
\overset{(c)}{\leq} O(\tfrac{1}{K^2}).
$$

$(a)$ is derived from the fact that $x_{\text{query}}$ and $P_{\text{input}}$ are independently sampled. Here, we straightforwardly bound $\mathbf{Attn}_k^{(t)} \cdot (\sum_{m \neq k} \mathbf{Attn}_m^{(t)\,2} + (1 - \mathbf{Attn}_k^{(t)})^2)$ by 2 for $P_{\text{input}} \in {\mathcal{E}_{\text{imbal}}^*}^c$. $(b)$ is established by applying Lemma E.2 to $\mathbf{Attn}_m^{(t)}$ for $m \neq 1, k$. $(c)$ is based on Lemma E.1, our selected $p_k$, Lemma C.5, and the evident relation:

$$
3 \exp\left(-\frac{c_{\text{im}}^2 N}{25 K^2}\right) \ll O(\tfrac{1}{K}).
$$

Similarly, we can show that $\alpha_k^{(t)} \geq \Omega(\tfrac{1}{K^2})$. $\qquad\square$

**Lemma E.4.** *Given $k > 1$, if Induction Hypothesis E.1 holds at iteration $0 \leq t \leq T_{k,1}$, then $\beta_{k,1}^{(t)} < 0$ satisfies*

$$
|\beta_{k,1}^{(t)}| \geq \Omega(\frac{1}{K^{1.98}}).
$$

*Proof.* Note that

$$
\sum_{m \neq k} \mathbf{Attn}_m^{(t)\,2} - \mathbf{Attn}_1^{(t)} - \mathbf{Attn}_k^{(t)}(1 - \mathbf{Attn}_k^{(t)})
$$

$$
= \sum_{m \neq 1, k} \mathbf{Attn}_m^{(t)\,2} - \mathbf{Attn}_1^{(t)}(1 - \mathbf{Attn}_1^{(t)}) - \mathbf{Attn}_k^{(t)}(1 - \mathbf{Attn}_k^{(t)})
$$

$$
\leq \max_{m \neq 1, k} \mathbf{Attn}_m^{(t)}(1 - \mathbf{Attn}_1^{(t)} - \mathbf{Attn}_k^{(t)}) - \mathbf{Attn}_1^{(t)}(1 - \mathbf{Attn}_1^{(t)}) - \mathbf{Attn}_k^{(t)}(1 - \mathbf{Attn}_k^{(t)})
$$

$$
\leq -(1 - \mathbf{Attn}_k^{(t)} - \mathbf{Attn}_1^{(t)})(\mathbf{Attn}_1^{(t)} + \mathbf{Attn}_k^{(t)} - \max_{m \neq 1, k} \mathbf{Attn}_m^{(t)}) \tag{20}
$$

Therefore, by gradient computation from Lemma C.2, we have

$$
\beta_{k,1}^{(t)} \leq \mathbb{E}\left[\mathbf{1}\{x_{\text{query}} = v_k \cap P_{\text{input}} \in \mathcal{E}_{\text{imbal}}^*\}\, \mathbf{Attn}_1^{(t)} \cdot \left(\sum_{m \neq k} \mathbf{Attn}_m^{(t)\,2} - \mathbf{Attn}_1^{(t)} - \mathbf{Attn}_k^{(t)}(1 - \mathbf{Attn}_k^{(t)})\right)\right]
$$

$$
+ \mathbb{E}\left[\mathbf{1}\{x_{\text{query}} = v_k \cap P_{\text{input}} \in {\mathcal{E}_{\text{imbal}}^*}^c\}\, \mathbf{Attn}_1^{(t)} \cdot \left(\sum_{m \neq k} \mathbf{Attn}_m^{(t)\,2}\right)\right]
$$

$$\stackrel{(a)}{\leq} p_k \cdot \mathbb{P}(P_{\text{input}} \in \mathcal{E}^{*}_{\text{imbal}}{}^c) + p_k \cdot \mathbb{P}(P_{\text{input}} \in \mathcal{E}^{*}_{\text{imbal}}) \times$$

$$\mathbb{E}\left[-\Omega\left(\frac{(\mathbf{Attn}_1^{(t)} + \mathbf{Attn}_k^{(t)} - \max_{m \neq 1, k} \mathbf{Attn}_m^{(t)})}{K^{0.49}}\right) \mid \{x_{\text{query}} = v_k\} \cap \mathcal{E}^{*}_{\text{imbal}}\right]$$

$$\leq p_k \cdot \left(-\Omega(\frac{1}{K^{0.98}})\right) + 3p_k \exp\left(-\frac{c_{\text{im}}^2 N}{25 K^2}\right)$$

$$= -\Omega(\frac{1}{K^{1.98}}).$$

where $(a)$ follows from eq. (20) and Lemma E.1 The last equality holds since

$$\frac{1}{K^{0.98}} \gg \exp\left(-\frac{c_{\text{im}}^2 N}{25 K^2}\right).$$

$\square$

**Lemma E.5.** *If Induction Hypothesis E.1 holds at iteration $0 \leq t \leq T_{k,1}$, for any $n \neq 1, k$, $\beta_{k,n}^{(t)}$ satisfies*

$$|\beta_{k,n}^{(t)}| \leq O(\frac{\alpha_k^{(t)} - \beta_{k,1}^{(t)}}{K}).$$

*Proof.* By gradient computation from Lemma C.2, we have

$$\beta_{k,n}^{(t)} \leq \mathbb{E}\left[\mathbf{1}\{x_{\text{query}} = v_k\} \mathbf{Attn}_n^{(t)} \cdot \left(\sum_{m \neq k} \mathbf{Attn}_m^{(t)\,2}\right)\right] \tag{21}$$

$$-\beta_{k,n}^{(t)} \leq \mathbb{E}\left[\mathbf{1}\{x_{\text{query}} = v_k\} \mathbf{Attn}_n^{(t)} \cdot \left(\mathbf{Attn}_n^{(t)} + \mathbf{Attn}_k^{(t)}(1 - \mathbf{Attn}_k^{(t)})\right)\right] \tag{22}$$

For eq. (21),

$$\beta_{k,n}^{(t)} \leq \mathbb{E}\left[\mathbf{1}\{x_{\text{query}} = v_k \cap P_{\text{input}} \in \mathcal{E}^{*}_{\text{imbal}}\} \mathbf{Attn}_n^{(t)} \cdot \left(\sum_{m \neq k} \mathbf{Attn}_m^{(t)\,2}\right)\right]$$

$$+ \mathbb{E}\left[\mathbf{1}\{x_{\text{query}} = v_k \cap P_{\text{input}} \in \mathcal{E}^{*}_{\text{imbal}}{}^c\} \mathbf{Attn}_n^{(t)} \cdot \left(\sum_{m \neq k} \mathbf{Attn}_m^{(t)\,2}\right)\right]$$

$$\leq p_k \cdot \mathbb{P}(P_{\text{input}} \in \mathcal{E}^{*}_{\text{imbal}}) \cdot \mathbb{E}\left[\mathbf{Attn}_n^{(t)} \cdot \left(\mathbf{Attn}_1^{(t)\,2} + O(\frac{1}{K})\right) \mid \{x_{\text{query}} = v_k\} \cap \mathcal{E}^{*}_{\text{imbal}}\right]$$

$$+ p_k \cdot \mathbb{P}(P_{\text{input}} \in \mathcal{E}^{*}_{\text{imbal}}{}^c)$$

$$\stackrel{(a)}{\leq} O(\frac{1}{K^3}) + O(\frac{|\beta_{k,1}^{(t)}|}{K}) + 3p_k \exp\left(-\frac{c_{\text{im}}^2 N}{25 K^2}\right)$$

$$\leq O(\frac{1}{K^3}) + O(\frac{|\beta_{k,1}^{(t)}|}{K}).$$

From Lemma E.2, we observe:

$$|\beta_{k,1}^{(t)}| \geq p_k \cdot \mathbb{P}(P_{\text{input}} \in \mathcal{E}^{*}_{\text{imbal}}) \cdot \mathbb{E}\left[\Omega\left(\mathbf{Attn}_1^{(t)\,2}\right) \mid \{x_{\text{query}} = v_k\} \cap \mathcal{E}^{*}_{\text{imbal}}\right].$$

Consequently, $(a)$ is established based on this observation, and it is further noted from Lemma E.2, $\mathbf{Attn}_n^{(t)} \leq O(\frac{1}{K})$.

For eq. (22), we have

$$-\beta_{k,n}^{(t)} \leq p_k \mathbb{E}\left[\mathbf{Attn}_n^{(t)} \cdot \left(\mathbf{Attn}_n^{(t)} + \mathbf{Attn}_k^{(t)}(1 - \mathbf{Attn}_k^{(t)})\right) \mid \{x_{\text{query}} = v_k\} \cap \mathcal{E}^{*}_{\text{imbal}}\right] + p_k \cdot \mathbb{P}(\mathcal{E}^{*}_{\text{imbal}}{}^c)$$

$$\overset{(a)}{\leq} p_k \cdot \mathbb{P}(\mathcal{E}_{\text{imbal}}^*)\mathbb{E}\left[O(\frac{1-\mathbf{Attn}_k^{(t)}}{K}) \cdot \left(O(\frac{1-\mathbf{Attn}_k^{(t)}}{K}) + \mathbf{Attn}_k^{(t)}(1-\mathbf{Attn}_k^{(t)})\right) \mid \{x_{\text{query}} = v_k\} \cap \mathcal{E}_{\text{imbal}}^*\right]$$

$$+ p_k \cdot \mathbb{P}(P_{\text{input}} \in \mathcal{E}_{\text{imbal}}^{*c})$$

$$\leq p_k \cdot \mathbb{P}(\mathcal{E}_{\text{imbal}}^*)\mathbb{E}\left[O(\frac{\mathbf{Attn}_k^{(t)}(1-\mathbf{Attn}_k^{(t)})^2}{K}) \mid \{x_{\text{query}} = v_k\} \cap \mathcal{E}_{\text{imbal}}^*\right] + 3p_k \exp\left(-\frac{c_{\text{im}}^2 N}{25K^2}\right)$$

$$\leq O(\frac{\alpha_k^{(t)}}{K})$$

where $(a)$ holds by Lemma E.2. The last inequality follows the analysis in Lemma E.3, and the fact that

$$\alpha_k^{(t)} \geq \Omega(\frac{1}{K^2}) \gg 3\exp\left(-\frac{c_{\text{im}}^2 N}{25K^2}\right).$$

Thus,

$$|\beta_{k,n}^{(t)}| \leq O(\frac{\alpha_k^{(t)} - \beta_{k,1}^{(t)}}{K}).$$

$\square$

### E.2.3 AT THE END OF PHASE I

**Lemma E.6.** *Given $k \in [K]$, Induction Hypothesis E.1 holds for at least all $t \leq T_{1,k} = O(\frac{\log(K)K^{1.98}}{\eta})$, and at iteration $t = T_{1,k} + 1$, we have*

    a. $B_{k,1}^{(T_{1,k}+1)} \leq -0.49\log(K)$;

    b. $\mathbf{Attn}_1 = O(\frac{1}{K^{0.49}})$ if $x_{query} = v_k$ and $P_{input} \in \mathcal{E}_{imbal}^*$.

*Proof.* The existence of $T_{1,k} = O(\frac{\log(K)K^{1.98}}{\eta})$ is directly implied by Lemma E.3.

It is easy to verify Induction Hypothesis E.1 holds at $t = 0$. Now we suppose Induction Hypothesis E.1 holds for all iterations $\leq t - 1$, and prove it holds at $t$.

By Lemma E.3, we have $\alpha_k^{(t-1)} \geq 0$. Thus $A_k^{(t)} = A_k^{(t-1)} + \eta\alpha_k^{(t-1)} \geq 0$. Morover, combining Lemmas E.3 and E.4, $A_k^{(t)} - A_k^{(0)} \leq O(\frac{|B_{k,1}^{(t)} - B_{k,1}^{(0)}|}{K^{0.02}})$ we immediately obtain $A_k^{(t)} \leq O(\log(K)/K^{0.02})$.

For $m \neq 1, k$, by Lemma E.5, we have $|B_{k,m}^{(t)}| \leq O(\frac{A_k^{(t)} - A_k^{(0)} + |B_{k,1}^{(t)} - B_{k,1}^{(0)}|}{K}) \leq O(\log(K)/K)$.

The proof for the second statement is deferred to the next phase. (Lemma E.7) $\square$

### E.3 PHASE II: SWITCHING OF LEADING INFLUENCE

During phase I, $B_{k,1}^{(t)}$ significantly decreases, resulting in a reduction in $\mathbf{Attn}_1^{(t)}$, while other $\mathbf{Attn}_n^{(t)}$ with $n > 1$ remain approximately at $\Theta(\frac{1}{K})$. By the end of this phase, $(\mathbf{Attn}_1^{(t)})^2$ decreases to $O(\frac{1}{K^{0.98}})$, leading to a decrease in $|\beta_{k,1}^{(t)}|$ as it converges towards $\alpha_k^{(t)}$. At this point, phase II begins. Shortly after entering this phase, the prior guarantee of $B_{k,1}^{(t)}$ reduction dominance diminishes as $|\beta_{k,1}^{(t)}|$ reaches the same order of magnitude as $\alpha_k^{(t)}$.

For $k > 1$, define

$$T_{2,k} \triangleq \max\{t > T_{1,k} : A_k^{(t)} - B_{k,1}^{(t)} \leq 1.01\log(K)\}.$$

We will be based on the following induction hypothesis during phase II.

**Induction Hypothesis E.2.** *For $T_{1,k} < t \leq T_{2,k}$, the following holds*

    a. $A_k^{(t)}$ is monotonically increasing and $A_k^{(t)} \in [0, 0.52\log(K)]$;

b. $B_{k,1}^{(t)}$ is monotonically decreasing and $B_{k,1}^{(t)} \in [-0.51\log(K), -0.49\log(K)]$;

c. $|B_{k,n}^{(t)}| = O(\frac{A_k^{(t)} + |B_{k,1}^{(t)}|}{K})$ for any $n \neq 1, k$.

### E.3.1 TECHNICAL LEMMAS

We first introduce several technical lemmas that will be used for the proof of Induction Hypothesis E.2.

**Lemma E.7.** *If Induction Hypothesis E.2 holds at iteration $T_{1,k} < t \leq T_{2,k}$, if $x_{query} = v_k$ and $P_{input} \in \mathcal{E}_{imbal}^*$, the following holds*

1. $\mathbf{Attn}_k^{(t)} \in [\Omega(\frac{1}{K}), O(\frac{1}{K^{0.48}})]$;

2. $\mathbf{Attn}_1^{(t)} \in [\Omega(\frac{1}{K^{0.51}}), O(\frac{1}{K^{0.49}})]$;

3. $1 - \mathbf{Attn}_1^{(t)} - \mathbf{Attn}_k^{(t)} \geq \Omega(1)$;

*Proof.* Since $x_{query} = v_k$, and notice that $|\mathcal{V}_k| > 0$ for $P_{input} \in \mathcal{E}_{imbal}^*$, we have

$$\mathbf{Attn}_k^{(t)} = \frac{|\mathcal{V}_k| e^{v_k^\top Q^{(t)} v_k}}{\sum_{j \in [N]} e^{E_j^{x\top} Q^{(t)} v_k}}$$

$$= \frac{|\mathcal{V}_k| \exp(A_k^{(t)})}{\sum_{m \neq k} |\mathcal{V}_m| \exp(B_{k,m}^{(t)}) + |\mathcal{V}_k| \exp(A_k^{(t)})}$$

$$= \frac{1}{\sum_{m \neq k} \frac{|\mathcal{V}_m|}{|\mathcal{V}_k|} \exp(B_{k,m}^{(t)} - A_k^{(t)}) + 1}.$$

By Induction Hypothesis E.2

- for $m \neq 1, k$, we have $e^{-O(\frac{\log(K)}{K}) - 0.52\log(K)} \leq \exp(B_{k,m}^{(t)} - A_k^{(t)}) \leq e^{O(\frac{\log(K)}{K})}$;

- for $m = 1$, $e^{-1.01\log(K)} \leq \exp(B_{k,1}^{(t)} - A_k^{(t)}) \leq e^0$

Combining with the fact that $\sum_{m \neq k} \frac{|\mathcal{V}_m|}{|\mathcal{V}_k|} = \Theta(K)$ for $P_{input} \in \mathcal{E}_{imbal}^*$, thus

$$\mathbf{Attn}_k^{(t)} \geq \Omega(\frac{1}{K}).$$

Moreover, since $\frac{N - |\mathcal{V}_1|}{|\mathcal{V}_k|}$ is still $\Theta(K)$, we have

$$\mathbf{Attn}_k^{(t)} \leq \frac{1}{e^{-O(\frac{\log(K)}{K}) - 0.52\log(K)}(\frac{N - |\mathcal{V}_1|}{|\mathcal{V}_k|} - 1) + e^{-1.01\log(K)}\frac{|\mathcal{V}_1|}{|\mathcal{V}_k|} + 1} \leq O(\frac{1}{K^{0.48}}).$$

Then we turn to $\mathbf{Attn}_1^{(t)}$.

$$\mathbf{Attn}_1^{(t)} = \frac{|\mathcal{V}_1| \exp(B_{k,1}^{(t)})}{\sum_{m \neq k} |\mathcal{V}_m| \exp(B_{k,m}^{(t)}) + |\mathcal{V}_k| \exp(A_k^{(t)})}$$

$$= \frac{1}{\sum_{m \neq 1, k} \frac{|\mathcal{V}_m|}{|\mathcal{V}_1|} \exp(B_{k,m}^{(t)} - B_{k,1}^{(t)}) + \frac{|\mathcal{V}_k|}{|\mathcal{V}_1|} \exp(A_k^{(t)} - B_{k,1}^{(t)}) + 1}$$

By Induction Hypothesis E.2,

- for $m \neq 1, k$, we have

$$e^{0.49\log(K) - O(\frac{\log(K)}{K})} \leq \exp(B_{k,m}^{(t)} - B_{k,1}^{(t)}) \leq e^{0.51\log(K) + O(\frac{\log(K)}{K})};$$

- for $m = 1$, $e^{0.49\log(K)} \leq \exp(A_k^{(t)} - B_{k,1}^{(t)}) \leq e^{1.01\log(K)}$.

Thus

$$\mathbf{Attn}_1^{(t)} \geq \frac{1}{e^{0.51 \log(K) + O(\frac{\log(K)}{K})}\left(\frac{N-|\mathcal{V}_k|}{|\mathcal{V}_1|} - 1\right) + e^{1.01 \log(K) + O(\frac{\log(K)}{K})}\frac{|\mathcal{V}_k|}{|\mathcal{V}_1|} + 1} \geq \Omega(\frac{1}{K^{0.51}})$$

For the last statement,

$$1 - \mathbf{Attn}_1^{(t)} \geq \frac{e^{0.49 \log(K) - O(\frac{\log(K)}{K})}\left(\frac{N}{|\mathcal{V}_1|} - 1\right)}{e^{0.49 \log(K) - O(\frac{\log(K)}{K})}\left(\frac{N}{|\mathcal{V}_1|} - 1\right) + 1} \geq \Omega(1).$$

Thus,

$$1 - \mathbf{Attn}_k^{(t)} - \mathbf{Attn}_1^{(t)} \geq \Omega(1).$$

$\square$

**Lemma E.8.** *If Induction Hypothesis E.2 holds at iteration $T_{1,k} < t \leq T_{2,k}$, if $x_{query} = v_k$ and $P_{input} \in \mathcal{E}_{imbal}^*$, for $n \neq 1, k$, the following holds*

$$\mathbf{Attn}_n^{(t)} = O\left(\frac{1 - \mathbf{Attn}_k^{(t)} - \mathbf{Attn}_1^{(t)}}{K}\right).$$

*Proof.* Since $x_{\text{query}} = v_k$, then we have

$$\mathbf{Attn}_n^{(t)} = \frac{|\mathcal{V}_n| e^{v_n^\top Q^{(t)} v_k}}{\sum_{j \in [N]} e^{E_j^{x\top} Q^{(t)} v_k}}$$

$$= \frac{|\mathcal{V}_n| \exp(B_{k,n}^{(t)})}{\sum_{m \neq k} |\mathcal{V}_m| \exp(B_{k,m}^{(t)}) + |\mathcal{V}_k| \exp(A_k^{(t)})}$$

By Induction Hypothesis E.2, for $m, n \neq 1$, $e^{-O(\frac{\log(K)}{K})} \leq \exp(B_{k,m}^{(t)} - B_{k,n}^{(t)}) \leq e^{O(\frac{\log(K)}{K})}$, combining with the fact that $\frac{|\mathcal{V}_m|}{|\mathcal{V}_n|} = \Theta(1)$ for $P_{\text{input}} \in \mathcal{E}_{\text{imbal}}^*$, thus

$$\frac{\mathbf{Attn}_n^{(t)}}{1 - \mathbf{Attn}_k^{(t)} - \mathbf{Attn}_1^{(t)}} = \frac{|\mathcal{V}_n| \exp(B_{k,n}^{(t)})}{\sum_{m \neq 1, k} |\mathcal{V}_m| \exp(B_{k,m}^{(t)})}$$

$$= \frac{1}{\sum_{m \neq k, 1} \frac{|\mathcal{V}_m|}{|\mathcal{V}_n|} \exp(B_{k,m}^{(t)} - B_{k,n}^{(t)})}$$

$$\leq O(\frac{1}{K}).$$

$\square$

### E.3.2 CONTROLLING THE GRADIENT UPDATES IN PHASE II

**Lemma E.9.** *Given $k > 1$, if Induction Hypothesis E.2 holds at iteration $T_{1,k} < t \leq T_{2,k}$, then $\alpha_k^{(t)} \geq 0$ and satisfies*

$$\alpha_k^{(t)} \geq \Omega(\frac{1}{K^2}).$$

*Proof.* By gradient computation in Lemma C.2, we have

$$\alpha_k^{(t)} = \mathbb{E}\left[\mathbf{1}\{x_{\text{query}} = v_k\}\mathbf{Attn}_k^{(t)} \cdot \left(\sum_{m \neq k} \mathbf{Attn}_m^{(t)2} + (1 - \mathbf{Attn}_k^{(t)})^2\right)\right]$$

$$= \mathbb{E}\left[\mathbf{1}\{x_{\text{query}} = v_k \cap \mathcal{E}^*_{\text{imbal}}\} \mathbf{Attn}_k^{(t)} \cdot \left(\sum_{m \neq k} \mathbf{Attn}_m^{(t)2} + (1 - \mathbf{Attn}_k^{(t)})^2\right)\right]$$

$$+ \mathbb{E}\left[\mathbf{1}\{x_{\text{query}} = v_k \cap \mathcal{E}^{*c}_{\text{imbal}}\} \mathbf{Attn}_k^{(t)} \cdot \left(\sum_{m \neq k} \mathbf{Attn}_m^{(t)2} + (1 - \mathbf{Attn}_k^{(t)})^2\right)\right]$$

$$\geq p_k \cdot \mathbb{P}(P \in \mathcal{E}^*_{\text{imbal}})\mathbb{E}\left[\mathbf{Attn}_k^{(t)} \cdot \left(\sum_{m \neq k} \mathbf{Attn}_m^{(t)2} + (1 - \mathbf{Attn}_k^{(t)})^2\right) \mid \{x_{\text{query}} = v_k\} \cap \mathcal{E}^*_{\text{imbal}}\right]$$

$$\geq \Omega(\frac{1}{K^2}).$$

where the last inequality follows from Lemma C.5, Lemma E.7 and our choice of $p_k$. $\qquad\square$

**Lemma E.10.** *Given $k > 1$, if Induction Hypothesis E.2 holds at iteration $T_{k,1} \leq t \leq T_{k,2}$, $\beta_{k,1}^{(t)} < 0$ and satisfies*

$$|\beta_{k,1}^{(t)}| \in [\Omega(\frac{1}{K^{2.02}}), O(\frac{1}{K^{1.97}})].$$

*Proof.* Following the similar computations as Lemma E.4, we have

$$\sum_{m \neq k} \mathbf{Attn}_m^{(t)2} - \mathbf{Attn}_1^{(t)} - \mathbf{Attn}_k^{(t)}(1 - \mathbf{Attn}_k^{(t)})$$

$$\leq -(1 - \mathbf{Attn}_k^{(t)} - \mathbf{Attn}_1^{(t)})(\mathbf{Attn}_1^{(t)} + \mathbf{Attn}_k^{(t)} - \max_{m \neq 1,k} \mathbf{Attn}_m^{(t)})$$

Therefore,

$$\beta_{k,1}^{(t)} \leq \mathbb{E}\left[\mathbf{1}\{x_{\text{query}} = v_k \cap \mathcal{E}^*_{\text{imbal}}\} \mathbf{Attn}_1^{(t)} \cdot \left(\sum_{m \neq k} \mathbf{Attn}_m^{(t)2} - \mathbf{Attn}_1^{(t)} - \mathbf{Attn}_k^{(t)}(1 - \mathbf{Attn}_k^{(t)})\right)\right]$$

$$+ \mathbb{E}\left[\mathbf{1}\{x_{\text{query}} = v_k \cap \mathcal{E}^{*c}_{\text{imbal}}\} \mathbf{Attn}_1^{(t)} \cdot \left(\sum_{m \neq k} \mathbf{Attn}_m^{(t)2}\right)\right]$$

$$\overset{(a)}{\leq} p_k \cdot \mathbb{P}(P_{\text{input}} \in \mathcal{E}^*_{\text{imbal}}) \cdot \mathbb{E}\left[-\Omega(\frac{(\mathbf{Attn}_1^{(t)} + \mathbf{Attn}_k^{(t)} - \max_{m \neq 1,k} \mathbf{Attn}_m^{(t)})}{K^{0.51}}) \mid \{x_{\text{query}} = v_k\} \cap \mathcal{E}^*_{\text{imbal}}\right]$$

$$+ p_k \cdot \mathbb{P}(P_{\text{input}} \in \mathcal{E}^{*c}_{\text{imbal}})$$

$$\overset{(b)}{\leq} p_k \cdot \left(-\Omega(\frac{1}{K^{1.02}})\right) + 3p_k \exp\left(-\frac{c_{\text{im}}^2 N}{25 K^2}\right)$$

$$= -\Omega(\frac{1}{K^{2.02}})$$

where $(a)$ follows from Lemma E.7, $(b)$ is obtained by Lemma E.7 and Lemma C.5. The last inequality holds since

$$\frac{1}{K^{1.02}} \gg \exp\left(-\frac{c_{\text{im}}^2 N}{25 K^2}\right).$$

Moreover,

$$-\beta_{k,1}^{(t)} \leq \mathbb{E}\left[\mathbf{1}\{x_{\text{query}} = v_k \cap \mathcal{E}^*_{\text{imbal}}\} \mathbf{Attn}_1^{(t)} \cdot \left(\mathbf{Attn}_1^{(t)} + \mathbf{Attn}_k^{(t)}(1 - \mathbf{Attn}_k^{(t)})\right)\right]$$

$$+ \mathbb{E}\left[\mathbf{1}\{x_{\text{query}} = v_k \cap \mathcal{E}^{*c}_{\text{imbal}}\} \mathbf{Attn}_1^{(t)} \cdot \left(\mathbf{Attn}_1^{(t)} + \mathbf{Attn}_k^{(t)}(1 - \mathbf{Attn}_k^{(t)})\right)\right]$$

$$\overset{(a)}{\leq} p_k \cdot \mathbb{P}(P_{\text{input}} \in \mathcal{E}^*_{\text{imbal}}) \cdot \mathbb{E}\left[\mathbf{Attn}^{(t)}_1 \cdot O(\mathbf{Attn}^{(t)}_1 + \mathbf{Attn}^{(t)}_k) \mid \{x_{\text{query}} = v_k\} \cap \mathcal{E}^*_{\text{imbal}}\right]$$
$$+ 2p_k \cdot \mathbb{P}(P_{\text{input}} \in \mathcal{E}^{*\,c}_{\text{imbal}})$$
$$\overset{(b)}{\leq} p_k \cdot \left(O(\frac{1}{K^{0.97}})\right) + 6p_k \exp\left(-\frac{c^2_{\text{im}} N}{25 K^2}\right)$$
$$= O(\frac{1}{K^{1.97}}).$$

Here, $(a)$ arises from straightforwardly upper bounding $\mathbf{Attn}^{(t)}_1 + \mathbf{Attn}^{(t)}_k(1 - \mathbf{Attn}^{(t)}_k)$ to $2$ when $P_{\text{input}} \in \mathcal{E}^{*\,c}_{\text{imbal}}$. $(b)$ is established based on Lemma E.7. $\qquad\square$

**Lemma E.11.** *If Induction Hypothesis E.2 holds at iteration $T_{1,k} < t \leq T_{2,k}$, for any $n \neq 1, k$, $\beta^{(t)}_{k,n}$ satisfies*

$$|\beta^{(t)}_{k,n}| \leq O(\frac{\alpha^{(t)}_k - \beta^{(t)}_{k,1}}{K}).$$

*Proof.* By gradient computation from Lemma C.2, we have

$$\beta^{(t)}_{k,n} \leq \mathbb{E}\left[\mathbf{1}\{x_{\text{query}} = v_k\} \mathbf{Attn}^{(t)}_n \cdot \left(\sum_{m \neq k} \mathbf{Attn}^{(t)\,2}_m\right)\right] \tag{23}$$

$$-\beta^{(t)}_{k,n} \leq \mathbb{E}\left[\mathbf{1}\{x_{\text{query}} = v_k\} \mathbf{Attn}^{(t)}_n \cdot \left(\mathbf{Attn}^{(t)}_n + \mathbf{Attn}^{(t)}_k(1 - \mathbf{Attn}^{(t)}_k)\right)\right] \tag{24}$$

For eq. (23),

$$\beta^{(t)}_{k,n} \leq \mathbb{E}\left[\mathbf{1}\{x_{\text{query}} = v_k \cap P_{\text{input}} \in \mathcal{E}^*_{\text{imbal}}\} \mathbf{Attn}^{(t)}_n \cdot \left(\sum_{m \neq k} \mathbf{Attn}^{(t)\,2}_m\right)\right]$$
$$+ \mathbb{E}\left[\mathbf{1}\{x_{\text{query}} = v_k \cap P_{\text{input}} \in \mathcal{E}^{*\,c}_{\text{imbal}}\} \mathbf{Attn}^{(t)}_n \cdot \left(\sum_{m \neq k} \mathbf{Attn}^{(t)\,2}_m\right)\right]$$
$$\leq p_k \cdot \mathbb{P}(P_{\text{input}} \in \mathcal{E}^*_{\text{imbal}}) \cdot \mathbb{E}\left[\mathbf{Attn}^{(t)}_n \cdot \left(\mathbf{Attn}^{(t)\,2}_1 + O(\frac{1}{K})\right) \mid \{x_{\text{query}} = v_k\} \cap \mathcal{E}^*_{\text{imbal}}\right]$$
$$+ p_k \cdot \mathbb{P}(P_{\text{input}} \in \mathcal{E}^{*\,c}_{\text{imbal}})$$
$$\leq O(\frac{1}{K^3}) + O(\frac{|\beta^{(t)}_{k,1}|}{K}) + 3p_k \exp\left(-\frac{c^2_{\text{im}} N}{25 K^2}\right)$$
$$\leq O(\frac{1}{K^3}) + O(\frac{|\beta^{(t)}_{k,1}|}{K}).$$

For eq. (24), we have

$$-\beta^{(t)}_{k,n} \leq p_k \mathbb{E}\left[\mathbf{Attn}^{(t)}_n \cdot \left(\mathbf{Attn}^{(t)}_n + \mathbf{Attn}^{(t)}_k(1 - \mathbf{Attn}^{(t)}_k)\right) \mid \{x_{\text{query}} = v_k\} \cap \mathcal{E}^*_{\text{imbal}}\right] + p_k \cdot \mathbb{P}(\mathcal{E}^{*\,c}_{\text{imbal}})$$
$$= p_k \cdot \mathbb{P}(\mathcal{E}^*_{\text{imbal}}) \mathbb{E}\left[O(\frac{1 - \mathbf{Attn}^{(t)}_k}{K}) \cdot \left(O(\frac{1 - \mathbf{Attn}^{(t)}_k}{K}) + \mathbf{Attn}^{(t)}_k(1 - \mathbf{Attn}^{(t)}_k)\right) \mid \{x_{\text{query}} = v_k\} \cap \mathcal{E}^*_{\text{imbal}}\right]$$
$$+ 2p_k \cdot \mathbb{P}(\mathcal{E}^{*\,c}_{\text{imbal}})$$
$$\leq p_k \cdot \mathbb{P}(\mathcal{E}^*_{\text{imbal}}) \mathbb{E}\left[O(\frac{\mathbf{Attn}^{(t)}_k(1 - \mathbf{Attn}^{(t)}_k)^2}{K}) \mid \{x_{\text{query}} = v_k\} \cap \mathcal{E}^*_{\text{imbal}}\right] + 6p_k \exp\left(-\frac{c^2_{\text{im}} N}{25 K^2}\right)$$
$$\leq O(\frac{\alpha^{(t)}_k}{K})$$

From the analysis in Lemma E.9, we have

$$\alpha_k^{(t)} \geq \Omega(\frac{1}{K^2}) \gg 6 \exp\left(-\frac{c_{\text{im}}^2 N}{25K^2}\right)$$

Thus,

$$|\beta_{k,n}^{(t)}| \leq O(\frac{\alpha_k^{(t)} - \beta_{k,1}^{(t)}}{K}).$$

$\square$

### E.3.3 AT THE END OF PHASE II

**Lemma E.12.** *Given $k \in [K]$, Induction Hypothesis E.2 holds for at least all $T_{1,k} < t \leq T_{2,k} = T_{1,k} + O(\frac{\log(K)K^2}{\eta})$, and at iteration $t = T_{2,k} + 1$, we have*

 *a.* $A_k^{(T_{2,k}+1)} \geq 0.5\log(K)$;

 *b.* $B_k^{(T_{2,k}+1)} \geq -0.51\log(K)$.

*Proof.* The existence of $T_{2,k} = T_{1,k} + O(\frac{\log(K)K^2}{\eta})$ is directly implied by Lemmas E.9 and E.10.

It is easy to verify Induction Hypothesis E.2 holds at $T_{1,k} + 1$. Now we suppose Induction Hypothesis E.2 holds for all iterations $\leq t - 1$, and prove it holds at $t$.

For $m \neq 1, k$, by Lemma E.11, we have $|B_{k,m}^{(t)}| \leq |B_{k,m}^{(T_{1,k}+1)}| + O(\frac{A_k^{(T_{2,k})} - A_k^{(T_{1,k}+1)} + |B_{k,1}^{(T_{2,k})} - B_{k,1}^{(T_{1,k}+1)}|}{K}) \leq O(\log(K)/K)$.

Now suppose $A_k^{(T_{2,k}+1)} < 0.5\log(K)$, then $B_{k,1}^{(T_{2,k}+1)} < -0.51\log(K)$. Consider the first time that $B_{k,1}^{(t)}$ reaches $-0.501\log(K)$, denoted as $\widetilde{T}$. Note that $\widetilde{T} < T_{2,k}^{(t)}$ since the change of $B_{k,1}^{(t)}$, i.e. $|\beta_{k,1}^{(t)}| \ll \log(K)$. Then for $t \geq \widetilde{T}$, we have the following fact: if $x_{\text{query}} = v_k$ and $P_{\text{input}} \in \mathcal{E}_{\text{imbal}}^*$, the following holds:

 1. $\mathbf{Attn}_k^{(t)} \in [\Omega(\frac{1}{K}), O(\frac{1}{K^{0.5}})]$;

 2. $\mathbf{Attn}_1^{(t)} \leq O(\frac{1}{K^{0.501}})$;

Therefore, following the similar analysis as Lemma E.10, we have

$$\begin{aligned}
|\beta_{k,1}^{(t)}| &\leq \mathbb{E}\left[\mathbf{1}\{x_{\text{query}} = v_k \cap \mathcal{E}_{\text{imbal}}^*\} \mathbf{Attn}_1^{(t)} \cdot \left(\mathbf{Attn}_1^{(t)} + \mathbf{Attn}_k^{(t)}(1 - \mathbf{Attn}_k^{(t)})\right)\right] \\
&\quad + \mathbb{E}\left[\mathbf{1}\{x_{\text{query}} = v_k \cap \mathcal{E}_{\text{imbal}}^*{}^c\} \mathbf{Attn}_1^{(t)} \cdot \left(\mathbf{Attn}_1^{(t)} + \mathbf{Attn}_k^{(t)}(1 - \mathbf{Attn}_k^{(t)})\right)\right] \\
&\leq p_k \cdot \mathbb{P}(P \in \mathcal{E}_{\text{imbal}}^*) \cdot \mathbb{E}\left[\mathbf{Attn}_1^{(t)} \cdot O(\mathbf{Attn}_1^{(t)} + \mathbf{Attn}_k^{(t)}) \mid \{x_{\text{query}} = v_k\} \cap \mathcal{E}_{\text{imbal}}^*\right] \\
&\quad + 2p_k \cdot \mathbb{P}(\mathcal{E}_{\text{imbal}}^*{}^c) \\
&\leq p_k \cdot \left(O(\frac{1}{K^{1.02}})\right) + O(\frac{\alpha_k^{(t)}}{K^{0.501}}) + 6p_k \exp\left(-\frac{c_{\text{im}}^2 N}{25K^2}\right) \\
&\leq O(\frac{\alpha_k^{(t)}}{K^{0.01}}).
\end{aligned}$$

where the last inequality follows Lemma E.9.

Since $|B_{k,1}^{(T_{2,k}+1)} - B_{k,1}^{(\widetilde{T})}| \geq \Omega(\log(K))$, we have $A_k^{(T_{2,k}+1)} \geq |B_{k,1}^{(T_{2,k}+1)} - B_{k,1}^{(\widetilde{T})}| \cdot \Omega(K^{0.01}) + A_k^{(\widetilde{T})} \gg \Omega(K^{0.01}\log(K))$, which contradicts the assumption that $A_k^{(T_{2,k}+1)} < 0.5\log(K)$. Therefore, $A_k^{(T_{2,k}+1)} \geq 0.5\log(K)$. Noting that once $B_{k,1}^{(t)}$ passes $-0.501\log(K)$, it will changes will be largely smaller compared to the increase of $A_k^{(t)}$. Thus, $B_{k,1}^{(T_{2,k}+1)} \geq -0.51\log(K)$.

$\square$

### E.4 PHASE III: GROWTH OF TARGET FEATURE

After the transition phase, $A_k^{(t)}$ will experience a larger gradient, with the growth of $A_k^{(t)}$ becoming the dominant effect in this phase. For $k$-th feature $v_k$, we define the phase III as all iterations $T_{2,k} < t \leq T_{3,k}$, where

$$T_{3,k} \triangleq \max\left\{t > T_{2,k} : A_k^{(t)} \leq \log(K)\right\}.$$

We state the following induction hypothesis, which will hold throughout phase III:

**Induction Hypothesis E.3.** For each $T_{2,k} < t \leq T_{3,k}$, the following holds:

    a. $A_k^{(t)}$ is monotonically increasing and $A_k^{(t)} \in [0.5\log(K), \log(K)]$;

    b. $B_{k,1}^{(t)}$ is monotonically decreasing and $B_{k,1}^{(t)} \in [-0.51\log(K) - O(\frac{\log(K)}{K^{0.48}}), -0.49\log(K)]$

    c. $|B_{k,n}^{(t)}| = O(\frac{A_k^{(t)} + |B_{k,1}^{(t)}|}{K})$ for any $n \neq 1, k$.

#### E.4.1 TECHNICAL LEMMAS

We first introduce several technical lemmas that will be used for the proof of Induction Hypothesis E.3.

**Lemma E.13.** *If Induction Hypothesis E.3 holds at iteration $T_{k,2} < t \leq T_{k,3}$, if $x_{query} = v_k$ and $P_{input} \in \mathcal{E}_{imbal}^*$, the following holds*

    *1. $\mathbf{Attn}_k^{(t)} = \Omega(\frac{1}{K^{0.5}})$;*

    *2. $\mathbf{Attn}_1^{(t)} \in [\Omega(\frac{1}{K^{0.51}}), O(\frac{1}{K^{0.49}})]$;*

    *3. $1 - \mathbf{Attn}_k^{(t)} \geq \Omega(1)$.*

*Proof.* Since $x_{query} = v_k$, and notice that $|\mathcal{V}_k| > 0$ for $P_{input} \in \mathcal{E}_{imbal}^*$, we have

$$\mathbf{Attn}_k^{(t)} = \frac{|\mathcal{V}_k| e^{v_k^\top Q^{(t)} v_k}}{\sum_{j \in [N]} e^{E_j^{x\top} Q^{(t)} v_k}}$$

$$= \frac{|\mathcal{V}_k| \exp(A_k^{(t)})}{\sum_{m \neq k} |\mathcal{V}_m| \exp(B_{k,m}^{(t)}) + |\mathcal{V}_k| \exp(A_k^{(t)})}$$

$$= \frac{1}{\sum_{m \neq k} \frac{|\mathcal{V}_m|}{|\mathcal{V}_k|} \exp(B_{k,m}^{(t)} - A_k^{(t)}) + 1}$$

By Induction Hypothesis E.3,

    • for $m \neq 1$, $e^{-\left(\log(K) + O(\frac{\log(K)}{K})\right)} \leq \exp(B_{k,m}^{(t)} - A_k^{(t)}) \leq e^{O(\frac{\log(K)}{K}) - 0.5\log(K)}$;

    • $e^{-\left(1.51\log(K) + O(\frac{\log(K)}{K})\right)} \leq \exp(B_{k,1}^{(t)} - A_k^{(t)}) \leq e^{-1.01\log(K)}$

thus

$$\mathbf{Attn}_k^{(t)} \geq \frac{1}{e^{O(\frac{\log(K)}{K}) - 0.5\log(K)}\left(\frac{N - |\mathcal{V}_1|}{|\mathcal{V}_k|} - 1\right) + e^{-1.01\log(K)} \frac{|\mathcal{V}_1|}{|\mathcal{V}_k|} + 1} \geq \Omega(\frac{1}{K^{0.5}})$$

where the second inequality follows from the fact that $P_{input} \in \mathcal{E}_{imbal}^*$.

On the other hand,

$$\mathbf{Attn}_k^{(t)} \leq \frac{1}{e^{-\left(\log(K) + O(\frac{\log(K)}{K})\right)}\left(\frac{N - |\mathcal{V}_1|}{|\mathcal{V}_k|} - 1\right) + 1} \leq \frac{1}{e^{-1}\left(\frac{1}{U_k^{im}} - \frac{1}{K}\right) + 1}.$$

Thus,

$$1 - \mathbf{Attn}_k^{(t)} \geq \frac{e^{-\left(\log(K) + O(\frac{\log(K)}{K})\right)}\left(\frac{N - |\mathcal{V}_1|}{|\mathcal{V}_k|} - 1\right) + e^{-1.01\log(K)} \frac{|\mathcal{V}_1|}{|\mathcal{V}_k|}}{e^{-\left(\log(K) + O(\frac{\log(K)}{K})\right)}\left(\frac{N - |\mathcal{V}_1|}{|\mathcal{V}_k|} - 1\right) + e^{-1.01\log(K)} \frac{|\mathcal{V}_1|}{|\mathcal{V}_k|} + 1} \geq \Omega(1).$$

Then we turn to $\mathbf{Attn}_1^{(t)}$.

$$\mathbf{Attn}_1^{(t)} = \frac{|\mathcal{V}_1| \exp(B_{k,1}^{(t)})}{\sum_{m \neq k} |\mathcal{V}_m| \exp(B_{k,m}^{(t)}) + |\mathcal{V}_k| \exp(A_k^{(t)})}$$

$$= \frac{1}{\sum_{m \neq 1,k} \frac{|\mathcal{V}_m|}{|\mathcal{V}_k|} \exp(B_{k,m}^{(t)} - B_{k,1}^{(t)}) + \frac{|\mathcal{V}_k|}{|\mathcal{V}_1|} \exp(A_k^{(t)} - B_{k,1}^{(t)}) + 1}$$

By Induction Hypothesis E.3,

- for $m \neq 1, k$, we have

$$e^{0.49 \log(K) - O(\frac{\log(K)}{K})} \leq \exp(B_{k,m}^{(t)} - B_{k,1}^{(t)}) \leq e^{0.51 \log(K) + O(\frac{\log(K)}{K})};$$

- for $m = 1$, $e^{1.01 \log(K)} \leq \exp(A_k^{(t)} - B_{k,1}^{(t)}) \leq e^{1.51 \log(K) + O(\frac{\log(K)}{K})}$

thus

$$\mathbf{Attn}_1^{(t)} \leq \frac{1}{e^{0.49 \log(K) - O(\frac{\log(K)}{K})}(\frac{N - |\mathcal{V}_k|}{|\mathcal{V}_1|} - 1) + e^{1.01 \log(K)} \frac{|\mathcal{V}_k|}{|\mathcal{V}_1|} + 1} \leq O(\frac{1}{K^{0.49}}).$$

$$\mathbf{Attn}_1^{(t)} \geq \frac{1}{e^0 (\frac{N - |\mathcal{V}_k|}{|\mathcal{V}_1|} - 1) + e^{1.51 \log(K) + O(\frac{\log(K)}{K})} \frac{|\mathcal{V}_k|}{|\mathcal{V}_1|} + 1} \geq \Omega(\frac{1}{K^{0.51}}).$$

$\square$

**Lemma E.14.** *If Induction Hypothesis E.3 holds at iteration $T_{2,k} < t \leq T_{3,k}$, if $x_{query} = v_k$ and $P_{input} \in \mathcal{E}_{imbal}^*$, for $n \neq 1, k$, the following holds*

$$\mathbf{Attn}_n^{(t)} = O\left(\frac{1 - \mathbf{Attn}_k^{(t)} - \mathbf{Attn}_1^{(t)}}{K}\right).$$

*Proof.* By Induction Hypothesis E.3, $e^{-O(\frac{\log(K)}{K})} \leq \exp(B_{k,m}^{(t)} - B_{k,n}^{(t)}) \leq e^{O(\frac{\log(K)}{K})}$, combining with the fact that $\frac{|\mathcal{V}_m|}{|\mathcal{V}_n|} = \Theta(1)$ when $P_{input} \in \mathcal{E}_{imbal}^*$, thus

$$\frac{\mathbf{Attn}_n^{(t)}}{1 - \mathbf{Attn}_k^{(t)} - \mathbf{Attn}_1^{(t)}} = \frac{|\mathcal{V}_n| \exp(B_{k,n}^{(t)})}{\sum_{m \neq 1,k} |\mathcal{V}_m| \exp(B_{k,m}^{(t)})} = \frac{1}{\sum_{m \neq 1,k} \frac{|\mathcal{V}_m|}{|\mathcal{V}_n|} \exp(B_{k,m}^{(t)} - B_{k,n}^{(t)})} \leq O(\frac{1}{K}).$$

$\square$

### E.4.2 CONTROLLING THE GRADIENT UPDATES IN PHASE III

**Lemma E.15.** *At each iteration $T_{2,k} < t \leq T_{3,k}$, if Induction Hypothesis E.3 holds then $\alpha_k^{(t)} \geq 0$ and satisfies*

$$\alpha_k^{(t)} \geq \Omega(\frac{1}{K^{1.5}}).$$

*Proof.* By gradient computation from Lemma C.2, we have

$$\alpha_k^{(t)} = \mathbb{E}\left[\mathbf{1}\{x_{query} = v_k\} \mathbf{Attn}_k^{(t)} \cdot \left(\sum_{m \neq k} \mathbf{Attn}_m^{(t)2} + (1 - \mathbf{Attn}_k^{(t)})^2\right)\right]$$

$$= \mathbb{E}\left[\mathbf{1}\{x_{query} = v_k \cap \mathcal{E}_{imbal}^*\} \mathbf{Attn}_k^{(t)} \cdot \left(\sum_{m \neq k} \mathbf{Attn}_m^{(t)2} + (1 - \mathbf{Attn}_k^{(t)})^2\right)\right]$$

$$+ \mathbb{E}\left[\mathbf{1}\{x_{query} = v_k \cap {\mathcal{E}_{imbal}^*}^c\} \mathbf{Attn}_k^{(t)} \cdot \left(\sum_{m \neq k} \mathbf{Attn}_m^{(t)2} + (1 - \mathbf{Attn}_k^{(t)})^2\right)\right]$$

$$\geq p_k \cdot \mathbb{P}(P_{\text{input}} \in \mathcal{E}^*_{\text{imbal}}) \mathbb{E}\left[\mathbf{Attn}^{(t)}_k \cdot \left(\sum_{m \neq k} \mathbf{Attn}^{(t)}_m{}^2 + (1 - \mathbf{Attn}^{(t)}_k)^2\right) \mid \{x_{\text{query}} = v_k\} \cap \mathcal{E}^*_{\text{imbal}}\right]$$

$$\geq p_k \cdot \mathbb{P}(P_{\text{input}} \in \mathcal{E}^*_{\text{imbal}}) \mathbb{E}\left[\mathbf{Attn}^{(t)}_k \cdot (1 - \mathbf{Attn}^{(t)}_k)^2 \mid \{x_{\text{query}} = v_k\} \cap \mathcal{E}^*_{\text{imbal}}\right]$$

$$\geq \Omega(\frac{1}{K^{1.5}})$$

where the last inequality follows from Lemma C.5, Lemma E.13 and our choice of $p_k$. $\qquad\square$

**Lemma E.16.** *Given $k > 1$, if Induction Hypothesis E.3 holds at iteration $T_{k,2} \leq t \leq T_{k,3}$, $\beta^{(t)}_{k,1} < 0$ satisfies*

$$|\beta^{(t)}_{k,1}| \leq [\Omega(\frac{1}{K^{2.01}}), O(\frac{\alpha^{(t)}_k}{K^{0.48}})].$$

*Proof.* Following the similar computations as Lemma E.4, we have

$$\sum_{m \neq k} \mathbf{Attn}^{(t)}_m{}^2 - \mathbf{Attn}^{(t)}_1 - \mathbf{Attn}^{(t)}_k(1 - \mathbf{Attn}^{(t)}_k)$$

$$\leq -(1 - \mathbf{Attn}^{(t)}_k - \mathbf{Attn}^{(t)}_1)(\mathbf{Attn}^{(t)}_1 + \mathbf{Attn}^{(t)}_k - \max_{m \neq 1,k} \mathbf{Attn}^{(t)}_m)$$

Therefore,

$$\beta^{(t)}_{k,1} \leq \mathbb{E}\left[\mathbf{1}\{x_{\text{query}} = v_k \cap \mathcal{E}^*_{\text{imbal}}\} \mathbf{Attn}^{(t)}_1 \cdot \left(\sum_{m \neq k} \mathbf{Attn}^{(t)}_m{}^2 - \mathbf{Attn}^{(t)}_1 - \mathbf{Attn}^{(t)}_k(1 - \mathbf{Attn}^{(t)}_k)\right)\right]$$

$$+ \mathbb{E}\left[\mathbf{1}\{x_{\text{query}} = v_k \cap \mathcal{E}^*_{\text{imbal}}{}^c\} \mathbf{Attn}^{(t)}_1 \cdot \left(\sum_{m \neq k} \mathbf{Attn}^{(t)}_m{}^2\right)\right]$$

$$\overset{(a)}{\leq} p_k \cdot \mathbb{P}(P_{\text{input}} \in \mathcal{E}^*_{\text{imbal}}) \cdot \mathbb{E}\left[-\Omega(\frac{(\mathbf{Attn}^{(t)}_1 + \mathbf{Attn}^{(t)}_k - \max_{m \neq 1,k} \mathbf{Attn}^{(t)}_m)}{K^{0.51}}) \mid \{x_{\text{query}} = v_k\} \cap \mathcal{E}^*_{\text{imbal}}\right]$$

$$+ p_k \cdot \mathbb{P}(P_{\text{input}} \in \mathcal{E}^*_{\text{imbal}}{}^c)$$

$$\overset{(b)}{\leq} p_k \cdot \left(-\Omega(\frac{1}{K^{1.01}})\right) + 3p_k \exp\left(-\frac{c^2_{\text{im}}N}{25K^2}\right)$$

$$= -\Omega(\frac{1}{K^{2.01}}).$$

where $(a)$ and $(b)$ both follow from Lemma E.13. The last inequality holds since

$$\frac{1}{K^{1.01}} \gg \exp\left(-\frac{c^2_{\text{im}}N}{25K^2}\right).$$

Moreover,

$$-\beta^{(t)}_{k,1} \leq \mathbb{E}\left[\mathbf{1}\{x_{\text{query}} = v_1 \cap P_{\text{input}} \in \mathcal{E}^*_{\text{imbal}}\} \mathbf{Attn}^{(t)}_1 \cdot (\mathbf{Attn}_1 + \mathbf{Attn}_k(1 - \mathbf{Attn}_k))\right]$$

$$+ \mathbb{E}\left[\mathbf{1}\{x_{\text{query}} = v_k \cap P_{\text{input}} \in \mathcal{E}^*_{\text{imbal}}{}^c\} \mathbf{Attn}^{(t)}_1 \cdot (\mathbf{Attn}_1 + \mathbf{Attn}_k(1 - \mathbf{Attn}_k))\right]$$

$$\leq p_k \cdot \mathbb{P}(P_{\text{input}} \in \mathcal{E}^*_{\text{imbal}}) \cdot \mathbb{E}\left[\mathbf{Attn}_1 \cdot O(\mathbf{Attn}_1 + \mathbf{Attn}_k) \mid \{x_{\text{query}} = v_k\} \cap \mathcal{E}^*_{\text{imbal}}\right]$$

$$+ 2p_k \cdot \mathbb{P}(P_{\text{input}} \in \mathcal{E}^*_{\text{imbal}}{}^c)$$

$$\leq p_k \cdot \left(O(\frac{1}{K^{0.98}})\right) + O(\frac{\alpha^{(t)}_k}{K^{0.49}}) + 6p_k \exp\left(-\frac{c^2_{\text{im}}N}{25K^2}\right)$$

$$\leq O(\frac{\alpha_k^{(t)}}{K^{0.48}})$$

where the last inequality follows from Lemma E.15. $\qquad\square$

**Lemma E.17.** *If Induction Hypothesis E.3 holds at iteration $T_{2,k} < t \leq T_{3,k}$, for any $n \neq 1, k$,* $\beta_{k,n}^{(t)}$ *satisfies*

$$|\beta_{k,n}^{(t)}| \leq O(\frac{\alpha_k^{(t)} - \beta_{k,1}^{(t)}}{K}).$$

*Proof.* By gradient computation from Lemma C.2, we have

$$\beta_{k,n}^{(t)} \leq \mathbb{E}\left[\mathbf{1}\{x_{\text{query}} = v_k\} \, \mathbf{Attn}_n^{(t)} \cdot \left(\sum_{m \neq k} \mathbf{Attn}_m^{(t)\,2}\right)\right] \tag{25}$$

$$-\beta_{k,n}^{(t)} \leq \mathbb{E}\left[\mathbf{1}\{x_{\text{query}} = v_k\} \, \mathbf{Attn}_n^{(t)} \cdot \left(\mathbf{Attn}_n^{(t)} + \mathbf{Attn}_k^{(t)}(1 - \mathbf{Attn}_k^{(t)})\right)\right] \tag{26}$$

For eq. (25),

$$\beta_{k,n}^{(t)} \leq \mathbb{E}\left[\mathbf{1}\{x_{\text{query}} = v_k \cap P_{\text{input}} \in \mathcal{E}_{\text{imbal}}^*\} \, \mathbf{Attn}_n^{(t)} \cdot \left(\sum_{m \neq k} \mathbf{Attn}_m^{(t)\,2}\right)\right]$$

$$+ \mathbb{E}\left[\mathbf{1}\{x_{\text{query}} = v_k \cap P_{\text{input}} \in {\mathcal{E}_{\text{imbal}}^*}^c\} \, \mathbf{Attn}_n^{(t)} \cdot \left(\sum_{m \neq k} \mathbf{Attn}_m^{(t)\,2}\right)\right]$$

$$\leq p_k \cdot \mathbb{P}(P_{\text{input}} \in \mathcal{E}_{\text{imbal}}^*) \cdot \mathbb{E}\left[\mathbf{Attn}_n^{(t)} \cdot \left(\mathbf{Attn}_1^{(t)\,2} + O(\frac{1}{K})\right) \mid \{x_{\text{query}} = v_k\} \cap P_{\text{input}} \in \mathcal{E}_{\text{imbal}}^*\right]$$

$$+ p_k \cdot \mathbb{P}(P_{\text{input}} \in {\mathcal{E}_{\text{imbal}}^*}^c)$$

$$\leq O(\frac{1}{K^3}) + O(\frac{|\beta_{k,1}^{(t)}|}{K}) + 6p_k \exp\left(-\frac{c_{\text{im}}^2 N}{25K^2}\right)$$

$$\leq O(\frac{1}{K^3}) + O(\frac{|\beta_{k,1}^{(t)}|}{K}).$$

For eq. (26), we have

$$- \beta_{k,n}^{(t)} \leq p_k \mathbb{E}\left[\mathbf{Attn}_n^{(t)} \cdot \left(\mathbf{Attn}_n^{(t)} + \mathbf{Attn}_k^{(t)}(1 - \mathbf{Attn}_k^{(t)})\right) \mid \{x_{\text{query}} = v_k\} \cap P_{\text{input}} \in \mathcal{E}_{\text{imbal}}^*\right]$$

$$+ p_k \cdot \mathbb{P}(P_{\text{input}} \in {\mathcal{E}_{\text{imbal}}^*}^c)$$

$$= p_k \cdot \mathbb{P}(P_{\text{input}} \in \mathcal{E}_{\text{imbal}}^*)\mathbb{E}\left[O(\frac{1 - \mathbf{Attn}_k^{(t)}}{K}) \cdot \left(O(\frac{1 - \mathbf{Attn}_k^{(t)}}{K}) + \mathbf{Attn}_k^{(t)}(1 - \mathbf{Attn}_k^{(t)})\right) \mid \{x_{\text{query}} = v_k\} \cap \mathcal{E}_{\text{imbal}}^*\right]$$

$$+ p_k \cdot \mathbb{P}(P_{\text{input}} \in {\mathcal{E}_{\text{imbal}}^*}^c)$$

$$\leq p_k \cdot \mathbb{P}(\mathcal{E}_{\text{imbal}}^*)\mathbb{E}\left[O(\frac{\mathbf{Attn}_k^{(t)}(1 - \mathbf{Attn}_k^{(t)})^2}{K}) \mid \{x_{\text{query}} = v_k\} \cap \mathcal{E}_{\text{imbal}}^*\right] + 6p_k \exp\left(-\frac{c_{\text{im}}^2 N}{25K^2}\right)$$

$$\leq O(\frac{\alpha_k^{(t)}}{K})$$

From the analysis in Lemma E.15, we have

$$\alpha_k^{(t)} \geq \Omega(\frac{1}{K^{1.5}})$$

Thus,

$$|\beta_{k,n}^{(t)}| \leq O(\frac{\alpha_k^{(t)} - \beta_{k,1}^{(t)}}{K}).$$

$\qquad\square$

### E.4.3 AT THE END OF PHASE III

**Lemma E.18.** *Given $k > 1$, Induction Hypothesis E.3 holds for at least all $T_{2,k} < t \leq T_{3,k} = T_{2,k} + O(\frac{\log(K)K^{1.5}}{\eta})$, and at iteration $t = T_{3,k} + 1$, we have*

a. $A_k^{(T_{3,k}+1)} \geq \log(K)$;

b. $\mathbf{Attn}_k = \Omega(1)$ if $x_{query} = v_k$ and $P_{input} \in \mathcal{E}_{imbal}^*$.

*Proof.* The existence of $T_{3,k} = T_{2,k} + O(\frac{\log(K)K^{1.5}}{\eta})$ is directly implied by Lemma E.15.

It is easy to verify Induction Hypothesis E.3 holds at $t = T_{2,k} + 1$. Now we suppose Induction Hypothesis E.3 holds for all iterations $\leq t - 1$, and prove it holds at $t$.

By Lemma E.15, we have $\alpha_k^{(t-1)} \geq 0$. Thus $A_k^{(t)} = A_k^{(t-1)} + \eta \alpha_k^{(t-1)} \geq 0.5 \log(K)$. Morover, by Lemma E.16, we have $|B_{k,1}^{(t)} - B_{k,1}^{(T_{2,k}+1)}| \leq O(\frac{A_k^{(t)} - A_k^{(T_{2,k}+1)}}{K^{0.48}})$ we immediately obtain $B_{k,1}^{(t)} \geq -O(\log(K)/K^{0.48}) - 0.51 \log(K)$.

For $m \neq 1, k$, by Lemma E.17, we have $|B_{k,m}^{(t)}| \leq O(\frac{A_k^{(t)} - A_k^{(T_{2,k}+1)} + |B_{k,1}^{(t)} - B_{k,1}^{(T_{2,k}+1)}|}{K}) \leq O(\log(K)/K)$.

The proof for the second statement is deferred to the next phase. (Lemma E.19)  □

### E.5 PHASE IV: CONVERGENCE

When we reach $t = T_{3,k} + 1$, we have already achieved the desired attention structure for the query token associated with feature $v_k$. In this final phase, we establish that these structures, encompassing each under-represented feature, represent the solutions toward which the algorithm converges. This provides a more robust guarantee than merely stumbling upon these solutions at intermediate steps.

Given any $0 < \epsilon < 1$ for $k \in [K]$, define

$$T_{4,k}^\epsilon \triangleq \max\{t > T_{3,k} : A_k^{(t)} \leq \log\left((\frac{e(1 - L_1^{im})K + U_1^{im}K^{0.51}}{L_k^{im}} - 1)((\frac{3}{\epsilon})^{\frac{1}{2}} - 1)\right)\}.$$

**Induction Hypothesis E.4.** For $T_{3,k} < t \leq T_{4,k}^\epsilon$, suppose $\text{polylog}(K) \gg \log(\frac{1}{\epsilon})$, the following holds

a. $A_k^{(t)}$ is monotonically increasing and $A_k^{(t)} \in [\log(K), O(\log(K/\epsilon))]$;

b. $B_{k,1}^{(t)}$ is monotonically decreasing and

$$B_{k,1}^{(t)} \in [-0.51 \log(K) - O(\frac{\log(K)}{K^{0.48}}), -0.49 \log(K)]$$

c. $B_{k,n}^{(t)}$ is monotonically decreasing and $|B_{k,n}^{(t)}| = O(\frac{\log(K/\epsilon)}{K})$ for any $n \neq 1, k$.

### E.5.1 TECHNICAL LEMMAS

We first introduce several technical lemmas that will be used for the proof of Induction Hypothesis E.4.

**Lemma E.19.** *If Induction Hypothesis E.4 holds at iteration $T_{3,k} < t \leq T_{4,k}^\epsilon$, if $x_{query} = v_k$ and $P_{input} \in \mathcal{E}_{imbal}^*$, the following holds*

1. $\mathbf{Attn}_k^{(t)} = \Omega(1)$;

2. $(1 - \mathbf{Attn}_k^{(t)})^2 \geq \Omega(\epsilon) = \Omega(\exp(-\text{polylog}(K)))$.

*Proof.* Since $x_{\text{query}} = v_k$, then we have

$$\mathbf{Attn}_k^{(t)} = \frac{|\mathcal{V}_k| \exp(A_k^{(t)})}{\sum_{m \neq k} |\mathcal{V}_m| \exp(B_{k,m}^{(t)}) + |\mathcal{V}_k| \exp(A_k^{(t)})}$$

$$= \frac{1}{\sum_{m \neq k} \frac{|\mathcal{V}_m|}{|\mathcal{V}_k|} \exp(B_{k,m}^{(t)} - A_k^{(t)}) + 1}$$

By Induction Hypothesis E.4, we have

- for $m \neq 1, k$:

$$\exp(B_{k,m}^{(t)} - A_k^{(t)}) \le e^{O(\frac{\log(K/\epsilon)}{K}) - \log(K)} \le e^{O(\frac{\log(K) + \text{polylog}(K)}{K}) - \log(K)} \le O(\frac{1}{K}).$$

- for $m = 1$, $\exp(B_{k,1}^{(t)} - A_k^{(t)}) \le O(\frac{1}{K^{1.49}})$.

Therefore,

$$\mathbf{Attn}_k^{(t)} \ge \frac{1}{O(\frac{1}{K})(\frac{N - |\mathcal{V}_1|}{|\mathcal{V}_k|} - 1) + O(\frac{1}{K^{1.49}})\frac{|\mathcal{V}_1|}{|\mathcal{V}_k|} + 1} \ge \Omega(1).$$

On the other hand, we have

$$1 - \mathbf{Attn}_k^{(t)} = \frac{\sum_{m \neq k} \frac{|\mathcal{V}_m|}{|\mathcal{V}_k|} \exp(B_{k,m}^{(t)} - A_k^{(t)})}{\sum_{m \neq k} \frac{|\mathcal{V}_m|}{|\mathcal{V}_k|} \exp(B_{k,m}^{(t)} - A_k^{(t)}) + 1}$$

$$\overset{(a)}{\ge} \frac{\exp(\min_{m \neq 1,k} B_{k,m}^{(t)} - A_k^{(t)})(\frac{N - |\mathcal{V}_1|}{|\mathcal{V}_k|} - 1) + \exp(B_{k,1}^{(t)} - A_k^{(t)})\frac{|\mathcal{V}_1|}{|\mathcal{V}_k|}}{\exp(\min_{m \neq 1,k} B_{k,m}^{(t)} - A_k^{(t)})(\frac{N - |\mathcal{V}_1|}{|\mathcal{V}_k|} - 1) + \exp(B_{k,1}^{(t)} - A_k^{(t)})\frac{|\mathcal{V}_1|}{|\mathcal{V}_k|} + 1}$$

$$\ge \frac{\exp(\min_{m \neq k} B_{k,m}^{(t)} - A_k^{(t)})(\frac{(1 - U_1^{\text{im}})K}{U_k^{\text{im}}} - 1) + \exp(B_{k,1}^{(t)} - A_k^{(t)}) \cdot \frac{L_1^{\text{im}}}{U_k^{\text{im}}}}{\exp(\min_{m \neq k} B_{k,m}^{(t)} - A_k^{(t)})(\frac{(1 - U_1^{\text{im}})K}{U_k^{\text{im}}} - 1) + \exp(B_{k,1}^{(t)} - A_k^{(t)}) \cdot \frac{L_1^{\text{im}}}{U_k^{\text{im}}} + 1}$$

$$= \frac{(\frac{(1 - U_1^{\text{im}})K}{U_k^{\text{im}}} - 1 + \frac{L_1^{\text{im}} K^{0.49}}{U_k^{\text{im}}}) \exp(-A_k^{(t)})}{(\frac{(1 - U_1^{\text{im}})K}{U_k^{\text{im}}} - 1 + \frac{L_1^{\text{im}} K^{0.49}}{U_k^{\text{im}}}) \exp(-A_k^{(t)}) + 1}$$

$$\ge \Omega(\epsilon^{\frac{1}{2}}).$$

$(a)$ follows from the fact that $\frac{x}{1+x}$ increases w.r.t. $x$. $\qquad\square$

**Lemma E.20.** *If Induction Hypothesis E.4 holds at iteration $T_{3,k} < t \le T_{4,k}^\epsilon$, if $x_{query} = v_k$ and $P_{input} \in \mathcal{E}_{imbal}^*$, the following holds*

1. $\mathbf{Attn}_n^{(t)} = \Theta\left(\frac{1 - \mathbf{Attn}_k^{(t)}}{K}\right)$ *for $n \neq 1, k$,*

2. $\mathbf{Attn}_1^{(t)} \in [\Omega(\frac{1 - \mathbf{Attn}_k^{(t)}}{K^{0.51}}), O\left(\frac{1 - \mathbf{Attn}_k^{(t)}}{K^{0.49}}\right)].$

*Proof.*

$$\frac{\mathbf{Attn}_n^{(t)}}{1 - \mathbf{Attn}_k^{(t)}} = \frac{|\mathcal{V}_n| \exp(B_{k,n}^{(t)})}{\sum_{m \neq k} |\mathcal{V}_m| \exp(B_{k,m}^{(t)})}$$

If $n \neq 1$, by Induction Hypothesis E.4,

- for $m \neq 1, k$, $e^{-O(\frac{\log(K) - \log(\epsilon)}{K})} \le \exp(B_{k,m}^{(t)} - B_{k,n}^{(t)}) \le e^{O(\frac{\log(K) - \log(\epsilon)}{K})}$,

- for $m = 1$, $e^{-0.51 \log(K) - O(\frac{\log(K/\epsilon)}{K})} \le \exp(B_{k,1}^{(t)} - B_{k,n}^{(t)}) \le 0.$

combining with the fact that when $P_{\text{input}} \in \mathcal{E}_{\text{imbal}}^*$, $\frac{|\mathcal{V}_m|}{|\mathcal{V}_n|} = \Theta(1)$, $\frac{|\mathcal{V}_1|}{|\mathcal{V}_n|} = \Theta(K)$ , and $-\log(\epsilon) \ll \text{polylog}(K)$, thus

$$\frac{\mathbf{Attn}_n^{(t)}}{1 - \mathbf{Attn}_k^{(t)}} = \frac{|\mathcal{V}_n| \exp(B_{k,n}^{(t)})}{\sum_{m \neq k} |\mathcal{V}_m| \exp(B_{k,m}^{(t)})} = \frac{1}{\sum_{m \neq k} \frac{|\mathcal{V}_m|}{|\mathcal{V}_n|} \exp(B_{k,m}^{(t)} - B_{k,n}^{(t)})} = \Theta(\frac{1}{K}).$$

For $n = 1$, by Induction Hypothesis E.4, $e^{0.49 \log(K) - O(\frac{\log(K/\epsilon)}{K})} \leq \exp(B_{k,m}^{(t)} - B_{k,1}^{(t)}) \leq^{0.51 \log(K) + O(\frac{\log(K/\epsilon)}{K})}$, for $m \neq 1$, combining with the fact that $\frac{|\mathcal{V}_m|}{|\mathcal{V}_1|} = \Theta(\frac{1}{K})$ when $P_{\text{input}} \in \mathcal{E}_{\text{imbal}}^*$, and $-\log(\epsilon) \ll \text{polylog}(K)$, we have

$$\frac{\mathbf{Attn}_1^{(t)}}{1 - \mathbf{Attn}_k^{(t)}} = \frac{1}{\sum_{m \neq k} \frac{|\mathcal{V}_m|}{|\mathcal{V}_n|} \exp(B_{k,m}^{(t)} - B_{k,n}^{(t)})} \leq O(\frac{1}{K \cdot \frac{1}{K} \cdot e^{0.49 \log(K) - O(\frac{\log(K/\epsilon)}{K})} + 1}) = O(\frac{1}{K^{0.49}}).$$

$$\frac{\mathbf{Attn}_1^{(t)}}{1 - \mathbf{Attn}_k^{(t)}} = \frac{1}{\sum_{m \neq k} \frac{|\mathcal{V}_m|}{|\mathcal{V}_n|} \exp(B_{k,m}^{(t)} - B_{k,n}^{(t)})} \geq O(\frac{1}{K \cdot \frac{1}{K} \cdot e^{0.51 \log(K) + O(\frac{\log(K/\epsilon)}{K})} + 1}) \geq \Omega(\frac{1}{K^{0.51}}).$$

$\square$

### E.5.2 Controlling the Gradient Updates in Phase IV

**Lemma E.21.** *At each iteration $T_{3,k} < t \leq T_{4,k}^\epsilon$, if Induction Hypothesis E.4 holds then $\alpha_k^{(t)} \geq 0$ and satisfies*

$$\alpha_k^{(t)} \geq \Omega(\frac{\epsilon}{K}).$$

*Proof.* The analysis is similar to Lemma E.15, but we need to be more careful about the lower bound of $1 - \mathbf{Attn}_k^{(t)}$. By gradient computation

$$\alpha_k^{(t)} = \mathbb{E}\left[\mathbf{1}\{x_{\text{query}} = v_k\} \mathbf{Attn}_k^{(t)} \cdot \left(\sum_{m \neq k} \mathbf{Attn}_m^{(t)^2} + (1 - \mathbf{Attn}_k^{(t)})^2\right)\right]$$

$$\geq p_k \cdot \mathbb{P}(P \in \mathcal{E}^*) \mathbb{E}\left[\mathbf{Attn}_k^{(t)} \cdot \left(\sum_{m \neq k} \mathbf{Attn}_m^{(t)^2} + (1 - \mathbf{Attn}_k^{(t)})^2\right) \mid \{x_{\text{query}} = v_k\} \cap \mathcal{E}^*\right]$$

$$\geq p_k \cdot \mathbb{P}(P \in \mathcal{E}^*) \mathbb{E}\left[\mathbf{Attn}_k^{(t)} \cdot (1 - \mathbf{Attn}_k^{(t)})^2 \mid \{x_{\text{query}} = v_k\} \cap \mathcal{E}^*\right]$$

$$\geq \Omega(\frac{\epsilon}{K})$$

where the last inequality follows from Lemma E.19 and our choice of $p_k$. $\square$

**Lemma E.22.** *At each iteration $T_{3,k} < t \leq T_{4,k}^\epsilon$, if Induction Hypothesis E.4 holds then given $k \in [K]$, $\beta_{k,1}^{(t)}$ satisfies*

$$-O(\frac{\alpha_k^{(t)}}{K^{0.49}}) \leq \beta_{k,n}^{(t)} \leq 0.$$

*Proof.* Following the similar computations as Lemma E.4, we have

$$\sum_{m \neq k} \mathbf{Attn}_m^{(t)^2} - \mathbf{Attn}_1^{(t)} - \mathbf{Attn}_k^{(t)}(1 - \mathbf{Attn}_k^{(t)})$$

$$\leq -(1 - \mathbf{Attn}_k^{(t)} - \mathbf{Attn}_1^{(t)})(\mathbf{Attn}_1^{(t)} + \mathbf{Attn}_k^{(t)} - \max_{m \neq 1,k} \mathbf{Attn}_m^{(t)}).$$

Therefore,

$$
\begin{aligned}
\beta_{k,1}^{(t)} \leq & \, \mathbb{E}\left[\mathbf{1}\{x_{\text{query}} = v_k \cap \mathcal{E}_{\text{imbal}}^*\}\, \mathbf{Attn}_1^{(t)} \cdot \left(\sum_{m \neq k} \mathbf{Attn}_m^{(t)\,2} - \mathbf{Attn}_1^{(t)} - \mathbf{Attn}_k^{(t)}(1 - \mathbf{Attn}_k^{(t)})\right)\right] \\
& + \mathbb{E}\left[\mathbf{1}\{x_{\text{query}} = v_k \cap \mathcal{E}_{\text{imbal}}^{*\,c}\}\, \mathbf{Attn}_1^{(t)} \cdot \left(\sum_{m \neq k} \mathbf{Attn}_m^{(t)\,2}\right)\right] \\
\overset{(a)}{\leq} & \, p_k \cdot \mathbb{P}(P_{\text{input}} \in \mathcal{E}_{\text{imbal}}^*) \cdot \mathbb{E}\left[-\Omega(\frac{(1 - \mathbf{Attn}_k^{(t)})^2}{K^{0.51}}) \mid \{x_{\text{query}} = v_k\} \cap \mathcal{E}_{\text{imbal}}^*\right] \\
& + p_k \cdot \mathbb{P}(\mathcal{E}_{\text{imbal}}^{*\,c}) \\
\leq & \, p_k \cdot \left(-\Omega(\frac{\epsilon}{K^{0.51}})\right) + 3 p_k \exp\left(-\frac{c_{\text{im}}^2 N}{25 K^2}\right) \\
< & \, 0.
\end{aligned}
$$

where $(a)$ follows from Lemma E.20. The last inequality holds since

$$
\frac{\epsilon}{K^{0.51}} \geq \frac{\exp(-\operatorname{polylog}(K))}{K^{0.51}} \gg \exp\left(-\frac{c_{\text{im}}^2 N}{25 K^2}\right).
$$

Moreover,

$$
\begin{aligned}
-\beta_{k,1}^{(t)} \leq & \, \mathbb{E}\left[\mathbf{1}\{x_{\text{query}} = v_k \cap \mathcal{E}_{\text{imbal}}^*\}\, \mathbf{Attn}_1^{(t)} \cdot \left(\mathbf{Attn}_1^{(t)} + \mathbf{Attn}_k^{(t)}(1 - \mathbf{Attn}_k^{(t)})\right)\right] \\
& + \mathbb{E}\left[\mathbf{1}\{x_{\text{query}} = v_k \cap \mathcal{E}_{\text{imbal}}^{*\,c}\}\, \mathbf{Attn}_1^{(t)} \cdot \left(\mathbf{Attn}_1^{(t)} + \mathbf{Attn}_k^{(t)}(1 - \mathbf{Attn}_k^{(t)})\right)\right] \\
\leq & \, p_k \cdot \mathbb{P}(P_{\text{input}} \in \mathcal{E}_{\text{imbal}}^*) \cdot \mathbb{E}\left[\cdot O(\mathbf{Attn}_1^{(t)}(1 - \mathbf{Attn}_k^{(t)})) \mid \{x_{\text{query}} = v_k\} \cap \mathcal{E}_{\text{imbal}}^*\right] \\
& + 2 p_k \cdot \mathbb{P}(\mathcal{E}_{\text{imbal}}^{*\,c}) \\
\leq & \, O(\frac{\alpha_k^{(t)}}{K^{0.49}}) + 6 p_k \exp\left(-\frac{c_{\text{im}}^2 N}{25 K^2}\right) \\
= & \, O(\frac{\alpha_k^{(t)}}{K^{0.49}}).
\end{aligned}
$$

$\square$

**Lemma E.23.** *At each iteration $T_{3,k} < t \leq T_{4,k}^\epsilon$, if Induction Hypothesis E.4 holds then given $k \in [K]$, for any $n \neq 1, k$, $\beta_{k,n}^{(t)}$ satisfies*

$$
-O(\frac{\alpha_k^{(t)}}{K}) \leq \beta_{k,n}^{(t)} \leq 0.
$$

*Proof.* Note that conditioning on the event $\{x_{\text{query}} = v_k\} \cap \mathcal{E}_{\text{imbal}}^*$, by Lemmas E.19 and E.20, we have $\mathbf{Attn}_k^{(t)} = \Omega(1)$, $\max_{m \neq k} \mathbf{Attn}_m = O(\frac{1}{K^{0.49}})$, thus

$$
\begin{aligned}
\sum_{m \neq k} \mathbf{Attn}_m^{(t)\,2} - \mathbf{Attn}_n^{(t)} - \mathbf{Attn}_k^{(t)}(1 - \mathbf{Attn}_k^{(t)}) &\leq \max_{m \neq k} \mathbf{Attn}_m^{(t)} \sum_{m \neq k} \mathbf{Attn}_m^{(t)} - \mathbf{Attn}_k^{(t)}(1 - \mathbf{Attn}_k^{(t)}) \\
&= -(1 - \mathbf{Attn}_k^{(t)})(\mathbf{Attn}_k^{(t)} - \max_{m \neq k} \mathbf{Attn}_m^{(t)}) \\
&\leq -\Omega(1 - \mathbf{Attn}_k^{(t)}). \quad (27)
\end{aligned}
$$

Therefore,

$$
\beta_{k,n}^{(t)} \leq \mathbb{E}\left[\mathbf{1}\{x_{\text{query}} = v_k \cap \mathcal{E}^*\}\, \mathbf{Attn}_n^{(t)} \cdot \left(\sum_{m \neq k} \mathbf{Attn}_m^{(t)\,2} - \mathbf{Attn}_n^{(t)} - \mathbf{Attn}_k^{(t)}(1 - \mathbf{Attn}_k^{(t)})\right)\right]
$$

$$+ \mathbb{E}\left[ \mathbf{1}\{x_{\text{query}} = v_k \cap \mathcal{E}^*_{\text{imbal}}{}^c\} \mathbf{Attn}_n^{(t)} \cdot \left( \sum_{m \neq k} \mathbf{Attn}_m^{(t)}{}^2 \right) \right]$$

$$\leq p_k \cdot \mathbb{P}(P_{\text{input}} \in \mathcal{E}^*_{\text{imbal}}) \cdot \mathbb{E}\left[ -\Omega(\frac{(1 - \mathbf{Attn}_k)^2}{K}) \mid \{x_{\text{query}} = v_k\} \cap \mathcal{E}^* \right] + p_k \cdot \mathbb{P}(P_{\text{input}} \in \mathcal{E}^*_{\text{imbal}}{}^c)$$

$$\leq p_k \cdot \left( -\Omega(\frac{\epsilon}{K}) \right) + 3 p_k \exp\left( -\frac{c_{\text{im}}^2 N}{25 K^2} \right)$$

$$\leq 0.$$

The last inequality holds since

$$\frac{\epsilon}{K} \gg \frac{\exp(-\text{polylog}(K))}{K} \gg \exp\left( -\frac{c_{\text{im}}^2 N}{25 K^2} \right).$$

Moreover, we have

$$-\beta_{k,n}^{(t)} \leq p_k \mathbb{E}\left[ \mathbf{Attn}_n^{(t)} \cdot \left( \mathbf{Attn}_n^{(t)} + \mathbf{Attn}_k^{(t)}(1 - \mathbf{Attn}_k^{(t)}) \right) \mid \{x_{\text{query}} = v_k\} \cap \mathcal{E}^*_{\text{imbal}} \right] + 2 p_k \mathbb{P}(\mathcal{E}^*_{\text{imbal}}{}^c)$$

$$\leq p_k \mathbb{E}\left[ \Theta(\frac{1 - \mathbf{Attn}_k^{(t)}}{K}) \cdot O\left( \mathbf{Attn}_k^{(t)}(1 - \mathbf{Attn}_k^{(t)}) \right) \mid \{x_{\text{query}} = v_k\} \cap \mathcal{E}^*_{\text{imbal}} \right] + 6 p_k \exp\left( -\frac{c_{\text{im}}^2 N}{25 K^2} \right)$$

$$= p_k \mathbb{E}\left[ O(\frac{\mathbf{Attn}_k^{(t)}(1 - \mathbf{Attn}_k^{(t)})^2}{K}) \mid \{x_{\text{query}} = v_k\} \cap \mathcal{E}^*_{\text{imbal}} \right] + 6 p_k \exp\left( -\frac{c_{\text{im}}^2 N}{25 K^2} \right)$$

$$\leq O(\frac{\alpha_k^{(t)}}{K}).$$

$\square$

### E.5.3 AT THE END OF PHASE IV

**Lemma E.24.** *Given $k > 1$, and $0 < \epsilon < 1$, suppose $\text{polylog}(K) \gg \log(\frac{1}{\epsilon})$, then Induction Hypothesis E.4 holds for at least all $T_{3,k} < t \leq T_{4,k}^{\epsilon} = T_{3,k} + O(\frac{K \log(K \epsilon^{-\frac{1}{2}})}{\eta \epsilon})$, and at iteration $t = T_{4,k}^{\epsilon} + 1$, we have*

1. $\widetilde{\mathcal{L}}_k(\theta^{T_{4,k}^{\epsilon}+1}) < \frac{\epsilon}{2}$

2. *If $x_{query} = v_k$ and $P_{input} \in \mathcal{E}^*_{imbal}$, we have $(1 - \mathbf{Attn}_k^{(T_{4,k}^{\epsilon}+1)})^2 \leq O(\epsilon)$.*

*Proof.* The existence of $T_{4,k}^{\epsilon} = T_{3,k} + O(\frac{K \log(K \epsilon^{-\frac{1}{2}})}{\eta \epsilon})$ is directly implied by Lemma E.21.

It is easy to verify Induction Hypothesis E.4 holds at $t = T_{3,k} + 1$. Now we suppose Induction Hypothesis E.4 holds for all iterations $\leq t - 1$, and prove it holds at $t$.

By Lemma E.21, we have $\alpha_k^{(t-1)} \geq 0$. Thus $A_k^{(t)} = A_k^{(t-1)} + \eta \alpha_k^{(t-1)} \geq \log(K)$. Moreover, by Lemma E.22, we have $|B_{k,1}^{(t)} - B_{k,1}^{(T_{3,k}+1)}| \leq O(\frac{A_k^{(t)} - A_k^{(T_{3,k}+1)}}{K^{0.49}})$ we immediately obtain $B_{k,1}^{(t)} \geq -O(A_k^{(t)}/K^{0.49}) - O(\log(K)/K^{0.48}) - 0.51 \log(K)$.

For $m \neq 1, k$, by Lemma E.23, we have $|B_{k,m}^{(t)} - B_{k,m}^{(T_{3,k}+1)}| \leq O(\frac{A_k^{(t)} - A_k^{(T_{3,k}+1)}}{K}) \leq O(\log(K/\epsilon)/K)$, thus $|B_{k,m}^{(t)}| \leq O(\log(K/\epsilon)/K) + O(\log(K)/K) = O(\log(K/\epsilon)/K)$.

At iteration $t = T_{4,k}^{\epsilon} + 1$, we have $A_k^{(t)} \geq \log\left( (\frac{e(1 - L_1^{\text{im}})K + U_1^{\text{im}} K^{0.51}}{L_k^{\text{im}}} - e)((\frac{3}{\epsilon})^{\frac{1}{2}} - 1) \right)$, thus when $\{x_{\text{query}} = v_k\} \cap \{P_{\text{input}} \in \mathcal{E}^*_{\text{imbal}}\}$, we obtain

$$1 - \mathbf{Attn}_k^{(t)} = \frac{\sum_{m \neq k} \frac{|\mathcal{V}_m|}{|\mathcal{V}_k|} \exp(B_{k,m}^{(t)} - A_k^{(t)})}{\sum_{m \neq k} \frac{|\mathcal{V}_m|}{|\mathcal{V}_k|} \exp(B_{k,m}^{(t)} - A_k^{(t)}) + 1}$$

$$\leq \frac{\exp(\max_{m\neq 1,k} B_{k,m}^{(t)} - A_k^{(t)})(\frac{N-|\mathcal{V}_1|}{|\mathcal{V}_k|} - 1) + \exp(B_{k,1}^{(t)} - A_k^{(t)})\frac{|\mathcal{V}_1|}{|\mathcal{V}_k|}}{\exp(\max_{m\neq 1,k} B_{k,m}^{(t)} - A_k^{(t)})(\frac{N-|\mathcal{V}_1|}{|\mathcal{V}_k|} - 1) + \exp(B_{k,1}^{(t)} - A_k^{(t)})\frac{|\mathcal{V}_1|}{|\mathcal{V}_k|} + 1}$$

$$\leq \frac{\exp(1 - A_k^{(t)})(\frac{(1-L_1^{\mathrm{im}})K}{L_k^{\mathrm{im}}} - 1) + \exp(-0.49\log(K) - A_k^{(t)})\frac{U_1^{\mathrm{im}}K}{L_k^{\mathrm{im}}}}{\exp(1 - A_k^{(t)})(\frac{(1-L_1^{\mathrm{im}})K}{L_k^{\mathrm{im}}} - 1) + \exp(-0.49\log(K) - A_k^{(t)})\frac{U_1^{\mathrm{im}}K}{L_k^{\mathrm{im}}} + 1}$$

$$= \frac{\left(\left(\frac{e(1-L_1^{\mathrm{im}})K + U_1^{\mathrm{im}}K^{0.51}}{L_k} - e\right)\right)\exp(-A_k^{(t)})}{\left(\left(\frac{e(1-L_1^{\mathrm{im}})K + U_1^{\mathrm{im}}K^{0.51}}{L_k^{\mathrm{im}}} - e\right)\right)\exp(-A_k^{(t)}) + 1}$$

$$\leq \frac{((\frac{3}{\epsilon})^{\frac{1}{2}} - 1)^{-1}}{((\frac{3}{\epsilon})^{\frac{1}{2}} - 1)^{-1} + 1}$$

$$= (\epsilon/3)^{\frac{1}{2}}.$$

$$\widetilde{\mathcal{L}}_k(\theta^{(t)}) = \frac{1}{2}\mathbb{E}\left[\mathbf{1}\{P_{\mathrm{input}} \in \mathcal{E}_{\mathrm{imbal}}^*\}\left(\sum_{m\neq k}\mathbf{Attn}_m^{(t)\,2} + (1 - \mathbf{Attn}_k^{(t)})^2\right) \mid x_{\mathrm{query}} = v_k\right]$$

$$\leq \frac{1}{2}\mathbb{P}\left(P_{\mathrm{input}} \in \mathcal{E}_{\mathrm{imbal}}^*\right) \cdot \mathbb{E}\left[(O(\frac{1}{K^{0.49}}) + 1)(1 - \mathbf{Attn}_k^{(t)})^2 \mid x_{\mathrm{query}} = v_k \cap P_{\mathrm{input}} \in \mathcal{E}_{\mathrm{imbal}}^*\right]$$

$$\leq \frac{1}{2}(1 + O(\frac{1}{K^{0.49}})) \cdot \frac{\epsilon}{3}$$

$$\leq \frac{\epsilon}{2}.$$

$\square$

## E.6 PUTTING ALL TOGETHER: PROOF OF MAIN THEOREM FOR UNDER-REPRESENTED FEATURE

**Theorem E.1** (Restate of Theorem 3.2 for Under-represented Feature). *Suppose $p_1 = \Theta(1)$ and $p_k = \Theta(\frac{1}{K})$ for $2 \leq k \leq K$. For any $0 < \epsilon < 1$, suppose $N \geq \mathrm{poly}(K)$, and $\mathrm{polylog}(K) \gg \log(\frac{1}{\epsilon})$. We apply GD to train the loss function given in eq. (4). Then the following results hold.*

1. *The prediction error for **under-represented** feature converges: for $v_k$ with $2 \leq k \leq K$, with at most $T_k = O(\frac{\log(K)K^2}{\eta} + \frac{K\log\left(K\epsilon^{-\frac{1}{2}}\right)}{\epsilon\eta})$ GD iterations, $\mathcal{L}_k(\theta^{(T_k)}) \leq \mathcal{L}_k^* + \epsilon$, where $\mathcal{L}_k^* = \Theta(e^{-\mathrm{poly}(K)})$ is the global minimum of eq. (6);*

2. *Attention score concentrates: for each $2 \leq k < K$, if the query token is $v_k$, then after $T_k$ iterations, with probability at least $1 - e^{-\Omega(\mathrm{poly}(K))}$, the one-layer transformer nearly "pays all attention" to input tokens featuring $v_k$: $(1 - \mathbf{Attn}_k^{(T_k)})^2 \leq O(\epsilon)$.*

*Proof.* The first statement is obtained by letting $T_k = T_{4,k}^\epsilon + 1$, and combining Lemma E.24, Lemma C.9 and Lemma C.10:

$$\mathcal{L}_k(\theta^{(T_k)}) - \mathcal{L}_k^* \leq \mathcal{L}_k(\theta^{(T_k)}) - \mathcal{L}_k^{\mathrm{low}} \leq \widetilde{\mathcal{L}}_k(\theta^{(T_k)}) + 3\exp\left(-\frac{c_{\mathrm{im}}^2 N}{25K^2}\right) < \epsilon.$$

The second statement directly follows Lemma E.24. $\square$

# F   ANALYSIS FOR THE IMBALANCED CASE: DOMINANT FEATURE

In this section, we delve into the analysis of the dominant feature. The training dynamics for the dominant feature $v_1$ are relatively straightforward, comprising only a single phase.

Noticing that at the beginning $t = 0$, we already have the following lemma:

**Lemma F.1.** *For $x_{query} = v_1$ and $P_{input} \in \mathcal{E}^*_{imbal}$, at $t = 0$, we have $\mathbf{Attn}_1^{(0)} = \Omega(1)$, $\mathbf{Attn}_k^{(0)} = O(\frac{1}{K})$ for $k > 1$.*

Thus, we can directly enter the convergence phase, which is defined as follows. Given any $0 < \epsilon < 1$ define

$$T_{1,*}^\epsilon \triangleq \max\{t \geq 0 : A_1^{(t)} - \max_{m \neq 1} B_{1,m}^{(t)} \leq \log\left((\frac{1}{L_1^{im}} - 1)((\frac{2}{\epsilon})^{\frac{1}{2}} - 1)\right)\}.$$

**Induction Hypothesis F.1.** *For $0 \leq t \leq T_{1,*}^\epsilon$, suppose $\text{polylog}(K) \gg \log(\frac{1}{\epsilon})$, the following holds*

a. $A_1^{(t)}$ *is monotonically increasing and* $A_1^{(t)} \in [0, O(\log(1/\epsilon))]$;

b. $B_{k,n}^{(t)}$ *is monotonically decreasing and* $-O(\frac{A_1^{(t)}}{K}) \leq B_{1,n}^{(t)} \leq 0$ *for any $n \neq 1$.*

## F.1   TECHNICAL LEMMAS

We first introduce several technical lemmas that will be used for the proof of Induction Hypothesis F.1.

**Lemma F.2.** *If Induction Hypothesis F.1 holds at iteration $0 < t \leq T_{1,*}^\epsilon$, if $x_{query} = v_1$ and $\mathcal{E}^*_{imbal} \in P_{input}$, the following holds*

1. $\mathbf{Attn}_1^{(t)} = \Omega(1)$;

2. $(1 - \mathbf{Attn}_1^{(t)})^2 \geq \Omega(\epsilon) = \Omega(\exp(-\text{polylog}(K)))$.

*Proof.* Since $x_{query} = v_1$, then we have

$$\mathbf{Attn}_1^{(t)} = \frac{|\mathcal{V}_1|\exp(A_k^{(t)})}{\sum_{m \neq 1}|\mathcal{V}_m|\exp(B_{1,m}^{(t)}) + |\mathcal{V}_1|\exp(A_k^{(t)})}$$

$$= \frac{1}{\sum_{m \neq k}\frac{|\mathcal{V}_m|}{|\mathcal{V}_k|}\exp(B_{k,m}^{(t)} - A_k^{(t)}) + 1}$$

By Induction Hypothesis F.1, we have

$$\mathbf{Attn}_1^{(t)} \geq \frac{1}{\sum_{m \neq k}\frac{|\mathcal{V}_m|}{|\mathcal{V}_k|}\exp(B_{k,m}^{(0)} - A_k^{(0)}) + 1}$$

$$\geq \frac{1}{(\frac{N}{L_1^{im}N} - 1) + 1} \geq \Omega(1).$$

On the other hand, by the definition of $T_{1,*}^\epsilon$, we have

$$1 - \mathbf{Attn}_1^{(t)} = \frac{\sum_{m \neq 1}\frac{|\mathcal{V}_m|}{|\mathcal{V}_1|}\exp(B_{1,m}^{(t)} - A_k^{(t)})}{\sum_{m \neq 1}\frac{|\mathcal{V}_m|}{|\mathcal{V}_1|}\exp(B_{1,m}^{(t)} - A_1^{(t)}) + 1}$$

$$\overset{(a)}{\geq} \frac{\exp(\min_{m \neq 1}B_{1,m}^{(t)} - A_1^{(t)})(\frac{N}{|\mathcal{V}_1|} - 1)}{\exp(\min_{m \neq 1}B_{1,m}^{(t)} - A_1^{(t)})(\frac{N}{|\mathcal{V}_1|} - 1) + 1}$$

$$\geq \frac{\exp(\min_{m \neq 1}B_{1,m}^{(t)} - A_1^{(t)})(\frac{1}{U_1^{im}} - 1)}{\exp(\min_{m \neq 1}B_{1,m}^{(t)} - A_1^{(t)})(\frac{1}{U_1^{im}} - 1) + 1}$$

$$
= \frac{\exp(\max_{m\neq 1} B_{1,m}^{(t)} - A_1^{(t)} - \Delta B_1^{(t)})(\frac{1}{U_1^{\mathrm{im}}} - 1)}{\exp(\max_{m\neq 1} B_{1,m}^{(t)} - A_1^{(t)} - \Delta B_1^{(t)})(\frac{1}{U_1^{\mathrm{im}}} - 1) + 1}
$$

$$
\geq \frac{(\frac{1}{p_1 L_1} - 1)^{-1}((\frac{2}{\epsilon})^{\frac{1}{2}} - 1)^{-1} \cdot e^{-O(\frac{\mathrm{polylog}(K)}{K})}(\frac{1}{U_1^{\mathrm{im}}} - 1)}{(\frac{1}{L_1^{\mathrm{im}}} - 1)^{-1}((\frac{2}{\epsilon})^{\frac{1}{2}} - 1)^{-1}(\frac{1}{U_1^{\mathrm{im}}} - 1)e^{-O(\frac{\mathrm{polylog}(K)}{K})} + 1}
$$

$$
\geq \Omega(\epsilon^{\frac{1}{2}}).
$$

where $\Delta B_k^{(t)} = \max_{m\neq k} B_{k,m}^{(t)} - \min_{m\neq k} B_{k,m}^{(t)} = O(\frac{A_k^{(t)}}{K})$, $(a)$ follows from the fact that $\frac{x}{1+x}$ increases w.r.t. $x$. $\qquad\square$

**Lemma F.3.** *If Induction Hypothesis F.1 holds at iteration* $0 \leq t \leq T_{1,*}^{\epsilon}$, *if* $x_{\tau,query} = v_1$ *and* $P \in \mathcal{E}^*$, *for* $n \neq 1$, *the following holds*

$$
\mathbf{Attn}_n^{(t)} = \Theta\left(\frac{(1 - \mathbf{Attn}_1^{(t)})}{K}\right).
$$

*Proof.*

$$
\mathbf{Attn}_n^{(t)} = \frac{|\mathcal{V}_n| \exp(B_{1,n}^{(t)})}{\sum_{m\neq k} |\mathcal{V}_m| \exp(B_{1,m}^{(t)}) + |\mathcal{V}_1| \exp(A_1^{(t)})}
$$

By Induction Hypothesis F.1, $e^{-O(\frac{p\log(\frac{1}{\epsilon})}{K})} \leq \exp(B_{k,m}^{(t)} - B_{k,n}^{(t)}) \leq e^{O(\frac{p\log(\frac{1}{\epsilon})}{K})}$, combining the fact that $-\log(\epsilon) \ll \mathrm{polylog}(K)$ thus

$$
\frac{\mathbf{Attn}_n^{(t)}}{1 - \mathbf{Attn}_k^{(t)}} = \frac{|\mathcal{V}_n| \exp(B_{k,n}^{(t)})}{\sum_{m\neq k} |\mathcal{V}_m| \exp(B_{k,m}^{(t)})} = \frac{1}{\sum_{m\neq k} \frac{|\mathcal{V}_m|}{|\mathcal{V}_n|} \exp(B_{k,m}^{(t)} - B_{k,n}^{(t)})} = \Theta(\frac{1}{K}).
$$

$\qquad\square$

### F.2 CONTROLLING THE GRADIENT UPDATES

**Lemma F.4.** *At each iteration* $0 \leq t \leq T_{1,*}^{\epsilon}$, *if Induction Hypothesis F.1 holds then* $\alpha_1^{(t)} \geq 0$ *and satisfies*

$$
\alpha_k^{(t)} \geq \Omega(\epsilon).
$$

*Proof.* By gradient computation

$$
\alpha_1^{(t)} = \mathbb{E}\left[\mathbf{1}\{x_{\mathrm{query}} = v_1\} \mathbf{Attn}_1^{(t)} \cdot \left(\sum_{m\neq 1} \mathbf{Attn}_m^{(t)\,2} + (1 - \mathbf{Attn}_1^{(t)})^2\right)\right]
$$

$$
\geq p_1 \cdot \mathbb{P}(P_{\mathrm{input}} \in \mathcal{E}_{\mathrm{imbal}}^*)\mathbb{E}\left[\mathbf{Attn}_k^{(t)} \cdot \left(\sum_{m\neq k} \mathbf{Attn}_m^{(t)\,2} + (1 - \mathbf{Attn}_k^{(t)})^2\right) \mid \{x_{\mathrm{query}} = v_k\} \cap \mathcal{E}_{\mathrm{imbal}}^*\right]
$$

$$
\geq p_1 \cdot \mathbb{P}(P_{\mathrm{input}} \in \mathcal{E}_{\mathrm{imbal}}^*)\mathbb{E}\left[\mathbf{Attn}_k^{(t)} \cdot (1 - \mathbf{Attn}_k^{(t)})^2 \mid \{x_{\mathrm{query}} = v_k\} \cap \mathcal{E}^*\right]
$$

$$
\geq \Omega(\epsilon)
$$

where the last inequality follows from Lemma F.2 and our choice of $p_1$. $\qquad\square$

**Lemma F.5.** *At each iteration* $0 \leq t \leq T_{1,*}^{\epsilon}$, *if Induction Hypothesis F.1 holds then , for any* $n \neq 1$, $\beta_{1,n}^{(t)}$ *satisfies*

$$
-O(\frac{\alpha_1^{(t)}}{K}) \leq \beta_{1,n}^{(t)} \leq 0.
$$

*Proof.* Note that conditioning on the event $\{x_{\text{query}} = v_1\} \cap \mathcal{E}^*_{\text{imbal}}$, by Lemmas F.2 and F.3, we have $\mathbf{Attn}_1^{(t)} = \Omega(1)$, $\max_{m \neq 1} \mathbf{Attn}_m^{(t)} = O(\frac{1}{K})$, thus

$$
\sum_{m \neq 1} \mathbf{Attn}_m^{(t)^2} - \mathbf{Attn}_n^{(t)} - \mathbf{Attn}_1^{(t)}(1 - \mathbf{Attn}_1^{(t)})
$$

$$
\leq \max_{m \neq 1} \mathbf{Attn}_m^{(t)} \sum_{m \neq 1} \mathbf{Attn}_m^{(t)} - \mathbf{Attn}_1^{(t)}(1 - \mathbf{Attn}_1^{(t)})
$$

$$
= -(1 - \mathbf{Attn}_1^{(t)})(\mathbf{Attn}_1^{(t)} - \max_{m \neq 1} \mathbf{Attn}_m^{(t)})
$$

$$
\leq -\Omega(1 - \mathbf{Attn}_1^{(t)}). \tag{28}
$$

Therefore,

$$
\beta_{1,n}^{(t)} \leq \mathbb{E}\left[ \mathbf{1}\{x_{\text{query}} = v_1 \cap \mathcal{E}^*_{\text{imbal}}\} \, \mathbf{Attn}_n^{(t)} \cdot \left( \sum_{m \neq 1} \mathbf{Attn}_m^{(t)^2} - \mathbf{Attn}_n^{(t)} - \mathbf{Attn}_1^{(t)}(1 - \mathbf{Attn}_1^{(t)}) \right) \right]
$$

$$
+ \mathbb{E}\left[ \mathbf{1}\{x_{\text{query}} = v_1 \cap \mathcal{E}^{*c}_{\text{imbal}}\} \, \mathbf{Attn}_n^{(t)} \cdot \left( \sum_{m \neq 1} \mathbf{Attn}_m^{(t)^2} \right) \right]
$$

$$
\overset{(a)}{\leq} p_1 \cdot \mathbb{P}(P_{\text{input}} \in \mathcal{E}^*_{\text{imbal}}) \cdot \mathbb{E}\left[ -\Omega(\frac{(1 - \mathbf{Attn}_1^{(t)})^2}{K}) \mid \{x_{\text{query}} = v_1\} \cap \mathcal{E}^*_{\text{imbal}} \right] + p_1 \cdot \mathbb{P}(\mathcal{E}^{*c}_{\text{imbal}})
$$

$$
\overset{(b)}{\leq} p_1 \cdot \left( -\Omega(\frac{\epsilon}{K}) \right) + 3p_1 \exp\left( -\frac{c_{\text{im}}^2 N}{25K^2} \right)
$$

$$
\leq 0
$$

where $(a)$ follows from eq. (28) and Lemma F.3, $(b)$ follows from Lemma F.2. The last inequality holds since

$$
\frac{\epsilon}{K} \gg \frac{\exp(-\text{polylog}(K))}{K} \gg \exp\left( -\frac{c_{\text{im}}^2 p_2 N}{25K^2} \right).
$$

Moreover, we have

$$
-\beta_{1,n}^{(t)} \leq p_1 \mathbb{E}\left[ \mathbf{Attn}_n^{(t)} \cdot \left( \mathbf{Attn}_n^{(t)} + \mathbf{Attn}_1^{(t)}(1 - \mathbf{Attn}_1^{(t)}) \right) \mid \{x_{\text{query}} = v_1\} \cap \mathcal{E}^*_{\text{imbal}} \right] + 2p_1 \mathbb{P}(\mathcal{E}^{*c}_{\text{imbal}})
$$

$$
\leq p_1 \mathbb{E}\left[ \Theta(\frac{1 - \mathbf{Attn}_1^{(t)}}{K}) \cdot O\left( \mathbf{Attn}_k^{(t)}(1 - \mathbf{Attn}_1^{(t)}) \right) \mid \{x_{\text{query}} = v_1\} \cap \mathcal{E}^*_{\text{imbal}} \right] + 6p_1 \exp\left( -\frac{c_{\text{im}}^2 N}{25K^2} \right)
$$

$$
= p_1 \mathbb{E}\left[ O(\frac{\mathbf{Attn}_1^{(t)}(1 - \mathbf{Attn}_1^{(t)})^2}{K}) \mid \{x_{\text{query}} = v_1\} \cap \mathcal{E}^*_{\text{imbal}} \right] + 6p_1 \exp\left( -\frac{c_{\text{im}}^2 p^2 N}{25K^2} \right)
$$

$$
\leq O(\frac{\alpha_1^{(t)}}{K}).
$$

$\square$

### F.3 AT THE END OF THE PHASE

**Lemma F.6.** *Given $0 < \epsilon < \frac{1}{2}$, suppose $\text{polylog}(K) \gg \log(\frac{1}{\epsilon})$, then Induction Hypothesis F.1 holds for at least all $0 \leq t \leq T_{1,*}^\epsilon = O(\frac{\log(\epsilon^{-\frac{1}{2}})}{\eta\epsilon})$, and at iteration $t = T_{1,*}^\epsilon + 1$, we have*

1. *$\widetilde{\mathcal{L}}_1(\theta^{T_{1,*}^\epsilon + 1}) < \epsilon/2$;*

2. *If $x_{query} = 1$ and $P_{input} \in \mathcal{E}^*_{imbal}$, we have $(1 - \mathbf{Attn}_1^{(T_{1,*}^\epsilon + 1)})^2 \leq O(\epsilon)$.*

*Proof.* We first prove the existence of $T_{1,*}^\epsilon$. Recall that

$$T_{1,*}^\epsilon = \max\{t \geq 0 : A_1^{(t)} - \max_{m \neq 1} B_{1,m}^{(t)} \leq \log\left(\left(\frac{1}{L_1^{\text{im}}} - 1\right)\left(\left(\frac{2}{\epsilon}\right)^{\frac{1}{2}} - 1\right)\right)\}.$$

When $t \in [0, T_{1,*}^\epsilon]$, we can simply lower bound the update of $A_k^{(t)} - \max_{m \neq k} B_{k,m}^{(t)}$ as

$$A_k^{(t+1)} - \max_{m \neq k} B_{k,m}^{(t+1)} \geq A_k^{(t+1)} \geq (A_k^{(t)} + \Omega(\frac{\eta\epsilon}{K}))$$

Therefore, at most $T_{1,*}^\epsilon = O(\frac{\log\left(\left(\frac{1}{L_1^{\text{im}}} - 1\right)\left(\left(\frac{2}{\epsilon}\right)^{\frac{1}{2}} - 1\right)\right)}{\eta\epsilon}) = O(\frac{\log(\epsilon^{-\frac{1}{2}})}{\eta\epsilon})$ iterations are needed before $A_k^{(t)} - \max_{m \neq k} B_{k,m}^{(t)}$ exceeds $\log\left(\left(\frac{1}{L_1^{\text{im}}} - 1\right)\left(\left(\frac{2}{\epsilon}\right)^{\frac{1}{2}} - 1\right)\right)$.

It is easy to verify Induction Hypothesis F.1 holds at $t = 0$. Now we suppose Induction Hypothesis F.1 holds for all iterations $0 \leq t - 1$, and prove it holds at $t$.

By Lemma F.4, we have $\alpha_1^{(t-1)} \geq 0$. Thus $A_1^{(t)} \geq A_1^{(t-1)} \geq 0$. By Lemma F.5, we have $-O(\frac{\alpha_1^{(t-1)}}{K}) \leq \beta_{1,n}^{(t-1)} \leq 0$. Thus,

$$-B_{1,n}^{(t)} \leq -B_{1,n}^{(t-1)} + \eta O(\frac{\alpha_k^{(t-1)}}{K})$$

$$\leq O(\frac{A_1^{(t-1)}}{K}) + \eta O(\frac{\alpha_1^{(t-1)}}{K})$$

$$\leq O(\frac{A_1^{(t)}}{K}).$$

Moreover, by the definition of $T_{1,*}^\epsilon$, for any $t \leq T_{1,*}^\epsilon$ we immediately have

$$A_1^{(t)} \leq A_1^{(t)} - \max_{m \neq 1} B_{1,m}^{(t)} \leq \log\left(\left(\frac{1}{L_1^{\text{im}}} - 1\right)\left(\left(\frac{2}{\epsilon}\right)^{\frac{1}{2}} - 1\right)\right)$$

Therefore, $A_1^{(t)} \leq O(\log(\frac{1}{\epsilon}))$.

At iteration $t = T_{1,*}^\epsilon + 1$, we have $A_1^{(t)} - \max_{m \neq 1} B_{1,m}^{(t)} > \log\left(\left(\frac{1}{L_1^{\text{im}}} - 1\right)\left(\left(\frac{2}{\epsilon}\right)^{\frac{1}{2}} - 1\right)\right)$, thus when $\{x_{\text{query}} = v_1\} \cap \{P_{\text{input}} \in \mathcal{E}_{\text{imbal}}^*\}$, we obtain

$$1 - \mathbf{Attn}_1^{(t)} = \frac{\sum_{m \neq 1} \frac{|\mathcal{V}_m|}{|\mathcal{V}_1|} \exp(B_{1,m}^{(t)} - A_1^{(t)})}{\sum_{m \neq 1} \frac{|\mathcal{V}_m|}{|\mathcal{V}_1|} \exp(B_{1,m}^{(t)} - A_1^{(t)}) + 1}$$

$$\leq \frac{\exp(\max_{m \neq 1} B_{1,m}^{(t)} - A_1^{(t)})(\frac{N}{|\mathcal{V}_1|} - 1)}{\exp(\max_{m \neq 1} B_{1,m}^{(t)} - A_1^{(t)})(\frac{N}{|\mathcal{V}_1|} - 1) + 1}$$

$$\leq \frac{\exp(\max_{m \neq 1} B_{1,m}^{(t)} - A_1^{(t)})(\frac{1}{L_1^{\text{im}}} - 1)}{\exp(\max_{m \neq 1} B_{1,m}^{(t)} - A_1^{(t)})(\frac{1}{L_1^{\text{im}}} - 1) + 1}$$

$$\leq \frac{\left(\left(\frac{1}{L_1^{\text{im}}} - 1\right)\left(\left(\frac{2}{\epsilon}\right)^{\frac{1}{2}} - 1\right)\right)^{-1}\left(\frac{1}{L_1^{\text{im}}} - 1\right)}{\left(\left(\frac{1}{L_1^{\text{im}}} - 1\right)\left(\left(\frac{2}{\epsilon}\right)^{\frac{1}{2}} - 1\right)\right)^{-1}\left(\frac{1}{L_1^{\text{im}}} - 1\right) + 1}$$

$$= (\epsilon/2)^{\frac{1}{2}}.$$

Similarly,

$$\widetilde{\mathcal{L}}_1(\theta^{(t)}) = \frac{1}{2}\mathbb{E}\left[\mathbf{1}\{P_{\text{input}} \in \mathcal{E}_{\text{imbal}}^*\}\left(\sum_{m \neq k} \mathbf{Attn}_m^{(t)^2} + (1 - \mathbf{Attn}_k^{(t)})^2\right) \mid x_{\text{query}} = v_k\right]$$

$$\leq \frac{1}{2} \mathbb{P}\left(P_{\text{input}} \in \mathcal{E}^*_{\text{imbal}}\right) \cdot \mathbb{E}\left[(O(\frac{1}{K}) + 1)(1 - \mathbf{Attn}_k^{(t)})^2 \mid x_{\text{query}} = v_k \cap P_{\text{input}} \in \mathcal{E}^*_{\text{imbal}}\right]$$

$$\leq \frac{1}{2} \cdot (1 + O(\frac{1}{K})) \cdot \frac{\epsilon}{2}$$

$$\leq \epsilon/2.$$

$\square$

### F.4 PROOF OF MAIN THEOREM FOR DOMINANT FEATURE

**Theorem F.1** (Restate of Theorem 3.2). *Suppose $p_1 = \Theta(1)$ and $p_k = \Theta(\frac{1}{K})$ for $2 \leq k \leq K$. For any $0 < \epsilon < 1$, suppose $N \geq \text{poly}(K)$, and $\text{polylog}(K) \gg \log(\frac{1}{\epsilon})$. We apply GD to train the loss function given in eq. (4). Then the following results hold.*

1. *The prediction error for **dominant** feature converges: for $v_1$, with at most $T_1 = O(\frac{\log(\epsilon^{-\frac{1}{2}})}{\eta\epsilon})$ GD iterations, $\mathcal{L}_1(\theta^{(T_1)}) \leq \mathcal{L}_1^* + \epsilon$, where $\mathcal{L}_1^* = \Theta(e^{-\text{poly}(K)})$ is the global minimum of eq. (6);*

2. *Attention score concentrates: $k = 1$, if the query token is $v_k$, then after $T_k$ iterations, with probability at least $1 - e^{-\Omega(\text{poly}(K))}$, the one-layer transformer nearly "pays all attention" to input tokens featuring $v_k$: $(1 - \mathbf{Attn}_k^{(T_k)})^2 \leq O(\epsilon)$.*

*Proof.* The first statement is obtained by letting $T_1 = T_{1,*}^\epsilon + 1$, and combining Lemma F.6, Lemma C.9 and Lemma C.10:

$$\mathcal{L}_1(\theta^{(T_1)}) - \mathcal{L}_1^* \leq \mathcal{L}_1(\theta^{(T_1)}) - \mathcal{L}_1^{\text{low}} \leq \widetilde{\mathcal{L}}_1(\theta^{(T_1)}) + 3\exp\left(-\frac{c_{\text{im}}^2 N}{25K^2}\right) < \epsilon.$$

The second statement directly follows Lemma F.6. $\square$

