# OpenReview forum: "In-context Convergence of Transformers"
_ICLR.cc/2024/Conference — Submitted to ICLR 2024_

### Official Review · Reviewer_xLBH · 2023-10-27

**Soundness:** 3 good
**Presentation:** 4 excellent
**Contribution:** 4 excellent
**Rating:** 6
**Confidence:** 2

**Summary:**

This is a theoretical paper centers around addressing “How do softmax-based transformers trained via gradient descent learn in-context? “. In particular, the author(s) investigated the training dynamics of a one-layer transformer with softmax attention trained by GD for in-context learning. Previous studies only focused on linear transformers i.e. without softmax function.  The authors give convergence results regarding in context learning with balanced and nonbalanced features.

**Strengths:**

Originality: This is a very novel paper as the authors provide a theoretical aspect of the in context learning dynamics of nonlinear transformer signified by the softmax activation in the self attention layer/module. This has not been examined before.

Quality: the quality is high. The authors define the problem mathematically and details around the proofs are included. They also provide an interesting aspect of the learning dynamics via the relevant attention scores during the 4 phases of convergence.

Clarity: It is fairly clear as the mathematical notations are given prior to introduction of their main development. The presentation can be easily followed via definition of problems, proposed approach/theoretical establishment, and training phase analysis for two specific problems of linear regression where sampled features are balanced and nonbalanced.

Significance: the paper is important for ml theory as it discussed in context learning with “nonlinear” (softmax) transformers.

**Weaknesses:**

The major weakness is lack of empirical experiments. The linear regression setting according to task distribution and data distribution should not be too challenging to do synthetic experiments for balanced and nonbalanced features similar to Garg et al. or Zhang et al., 2023a. It will be interesting to compared with linear transformers such as Zhang et al. 2023 where in certain task scenarios, linear transformer fails.
Another concern is the nonlinearity of transformers are examined on features from linear functions. Would be possible to develop some insights on features from nonlinear functions?

**Questions:**

1.	The paper does not introduce/define Θ(K) in the notations.
2.	“set of distinct features {vk ∈Rd,k=1,...,K},where all features are orthonormal vectors”, does that mean the maximum K is d, since an orthonormal basis is d dimensional?  Is it too limiting? Correct me if I am wrong, does that also mean we can only have d distinct features/tokens as in part of Definition 3.1 part 2?
3.	The paper adopts W^V and W^{KQ} from linear SA by Zhang et al.  for example by setting v =1. Although the authors justify the use of v=1 with two reasons.  However, the objective from Zhang et al.  involves a residual connection term. In this paper, there is no residual connection term involved.  Could the authors justify either the use of v =1 as a scaling factor in their objective as in Equation (1) or not including the residual term in Equation (1) and subsequently in equation (3) and (4)?
4.	The four training phases of under-represented features differ. Could the order of phase 2 and 3 be swapped?
5.	Did author investigate other tasks with softmax transformer as discussed by the Garg et al. paper?

---

> ### Author Response · Authors · 2023-11-18
> **Response to Reviewer xLBH (part 1/2)**
>
> We thank the reviewer for providing the helpful review! We have addressed the reviewer’s helpful comments and revised the paper accordingly. The changes are marked in bright blue color in our revision. Please feel free to check and comment.
>
> **Q1:** The major weakness is lack of empirical experiments. The linear regression setting according to task distribution and data distribution should not be too challenging to do synthetic experiments for balanced and nonbalanced features similar to Garg et al. or Zhang et al., 2023a. It will be interesting to compared with linear transformers such as Zhang et al. 2023 where in certain task scenarios, linear transformer fails. Another concern is the nonlinearity of transformers are examined on features from linear functions. Would be possible to develop some insights on features from nonlinear functions?
>
> **A1:** Many thanks for your suggestions! In Appendix A of our revised paper, we have provided experiment results to validate our theoretical results. We summarize our key results as follows:
> (1) Figure 2 depicts the prediction error for each feature versus the training epochs, and highlights the stage-wise convergence process between the dominant and under-represented features for imbalanced case.
> (2) Figure 3 illustrates the evolution of attention score heatmap during the training, and verifies our theoretical results on attention score concentration and the multi-phase characterization of training dynamics. Please feel free to check and comment. Our current experiment code has adopted the existing well-packed transformer package, which does not allow change to linear transformer easily. Given this, our handcrafted code on linear transformer may not provide a fair comparison between the two. Note that although Zhang et al. (2023) studied linear attention layers in their theory, their experiments are also on more complex, large, and nonlinear transformers, which do not provide a code on linear transformers either.
>
> Regarding features from nonlinear functions, this is definitely an interesting question to ask. It also appears to be a challenging problem to analyze from theory side, at least not something straightforward. Our preliminary thoughts along this direction are as follows. To consider such nonlinear tasks/functions, (1) some additional layers may be required to learn the non-linear functions; and (2) the softmax weight matrix structure will change and may not admit any simplified zero entries, which is different from the current structure of linear tasks as in Eq. (2).
>
> **Q2:** The paper does not introduce/define $\Theta(K)$ in the notations.
>
> **A2:** Thanks. In our revision, we have formally defined $\Theta(K)$ in the notation section as suggested.
>
> **Q3:** The paper adopts $W^V$ and $W^{KQ}$ from linear SA by Zhang et al. for example by setting $v =1$. Although the authors justify the use of $v=1$ with two reasons. However, the objective from Zhang et al. involves a residual connection term. In this paper, there is no residual connection term involved. Could the authors justify either the use of v =1 as a scaling factor in their objective as in Equation (1) or not including the residual term in Equation (1) and subsequently in equation (3) and (4)?
>
> **A3:** The residual term in Zhang et al. can also be removed. Both Zhang et al. and our paper choose the loss function to only measure the distance between $y_{query}$ and $\hat{y}_{query}$. The residual term, which is associated with the prediction output $\hat{y}$, equals 0. (See Eq. (3.3) and Eq (3.4)). As a result, the residual term can be removed without any influence on the loss and training dynamics. Furthermore, as we discuss in **Remark 1** in our paper, our current specific parameterization (including setting $v=1$) does not lose optimality up to an exponentially small error.

---

> > ### Author Response · Authors · 2023-11-18
> > **Response to Reviewer xLBH (part 2/2)**
> >
> > **Q4:** The four training phases of under-represented features differ. Could the order of phase 2 and 3 be swapped?
> >
> > **A4:** For our current setting, the order of phases can not be swapped, and is determined by the fact that the query feature is not an under-represented feature, so that the reduction of the attention score to the dominant feature occurs first and then there is a transition to the increase of the attention score to the target under-represented feature.  For more complex settings, our current analysis suggests that for each query feature $v_k$, multiple phases can exist during the training process, with each phase dominated by the reduction of the attention to one non-target feature, i.e., by the reduction of $B_{k,n}$ with $n \not= k$. The order of these phases is determined by the probability distribution $p_k$, how $p_k$ compares to other $p_n$ with $n \neq k$, and the comparative values among $p_n$ for $n \neq k$. Ultimately, the final phase should be characterized by an increase in $A_k$. Of course, these thoughts will require rigorous mathematical proof, for which our current proof techniques seem to support. We leave the detailed proof as future work.
> >
> > **Q5:** Did author investigate other tasks with softmax transformer as discussed by the Garg et al. paper?
> >
> > **A5:** It is a very interesting future direction to consider tasks of a broader class of functions including nonlinear functions. We are getting started to explore some simple nonlinear cases, for example, quadratic functions or sparse linear regressions. Even those problems seem to be very challenging to characterize analytically.
> >
> > We thank the reviewer again for the helpful comments and suggestions for our work. Given that we have provided the requested experiments (which seems to be the main concern of the reviewer), if our response also resolves your other concerns to a satisfactory level, we kindly ask the reviewer to consider raising the rating of our work. Certainly, we are more than happy to answer any further questions that you may have.

---

> ### Author Response · Authors · 2023-11-22
> **Follow Up Reminder to Reviewer xLBH**
>
> Dear Reviewer xLBH,
>
> As the author-reviewer discussion period ends soon, we will appreciate it if you could check our response to your review comments. This way, if you have further questions and comments, we can still reply before the author-reviewer discussion period ends. If our response resolves your concerns, we kindly ask you to consider raising the rating of our work. Thank you very much for your time and efforts!

---

> > ### Comment · Reviewer_xLBH · 2023-11-22
> > **Thank you for your response.**
> >
> > Thank you for your responses and clarification. I will consider your suggestion/request.

---

### Official Review · Reviewer_kkxb · 2023-11-02

**Soundness:** 3 good
**Presentation:** 3 good
**Contribution:** 2 fair
**Rating:** 6
**Confidence:** 3

**Summary:**

The paper studies the convergence of one-layer encoder-based transformer with softmax attention only. The authors show that training this model with gradient descent with pairs $(x_i,y_i)$, where $x_i$ can be one of $K$ possible vectors $u_k$, and $y_i = wx_i$ for some $w$ that is drawn from some distribution. Two settings are considered: in the first one each of the possible $u_k$ vectors has  probability $\propto 1/K$ to be drawn (balanced data), while in the second one some vector has some vector being drawn with constant probability and the rest with probability $\propto 1/K$. In both settings they show that the transformer converges to an approximate minimum.

**Strengths:**

The convergence of transformer models is of interest to the community. The paper also studies sequence to sequence models, which is a setting unexplored in terms of convergence. The two-phase transition is intuitive, since the model is first trying to maximize the inner product of the token to be predicted with the ones that are the same vectors and then use them to extract their corresponding labels.

**Weaknesses:**

The main weakness of this paper is the setting. Specifically, even though the authors study sequence to sequence models, they also consider:
1. That all training points are one of k possible vectors. This is not compatible with the training process for linear regression. In general, in which setting we only have $K$ vectors that we sample them in that way.
2. The weight matrices are very constrained and sparsified to match the ones proposed in the construction of [2]. This construction also refers to linear transformers. The authors are multiply the matrices $W_K^\top W_Q$ and train them as one matrix (this also impacts the training dynamics).
3. The paper also consider only the attention mechanism and no residual.

I understand that the achieved loss is not very different, however this simply indicates that for the simple setting of linear regression transformers are more expressive than necessary. I do not think that these results are indicative of how the training is evolved.

**Questions:**

1. Could the authors clarify the motivation of this setting? Fixed data points and constraint sparse matrices while considering the task of linear regression.
2. Could the authors comment on how the two-phase transition observed in this setting, could be indicative of what is happening during actual training of these models?

---

> ### Author Response · Authors · 2023-11-18
> **Response to Reviewer kkxb (part 1/2)**
>
> We thank the reviewer for providing the helpful review! We have addressed the reviewer’s helpful comments and revised the paper accordingly. The changes are marked in bright blue color in our revision. Please feel free to check our revised paper and comment.
>
> **Q1:** That all training points are one of $K$ possible vectors. This is not compatible with the training process for linear regression. In general, in which setting we only have $K$ vectors that we sample them in that way.
>
> **A1:**
> Our data assumption, where each data point is randomly selected from a predefined set of features, is **standard** and **well accepted** in many recent theoretical work, for example, [3] on theoretical deep learning, [4] on transformer theory, and [5] on vision transformer theory.
> Such a model captures the case where data are sampled from a subspace where those $K$ features serve as basis vectors. Our analysis can be extended to study such a case by allowing the data sample to be correlated. Thus, our assumption indeed aligns with
> the typical linear regression setting, where the feature vectors are representative of a broader spectrum of data points typically encountered in linear regression scenarios.
>
> **Q2:** The weight matrices are very constrained and sparsified to match the ones proposed in the construction of [2]. This construction also refers to linear transformers. The authors are multiply the matrices $W_K^\top W_Q$  and train them as one matrix (this also impacts the training dynamics).
>
> **A2:** We respectfully disagree with the above comments.
>
> (i) Although our weight matrices adopt the construction in [2], our model is **NOT linear** transformer, but a **softmax nonlinear** transformer. This is the key difference of our study from that in [2]. Our main contribution lies in characterizing the training dynamics for the more practical **softmax** based transformer, which is more challenging to analyze than **linear** transformer.
>
> (ii) In our paper, we have shown that specializing our weight matrix to the construction in [2] **does not lose optimality** up to an exponential-delay error. Hence, such a choice of weight matrix should not be considered as "very constrained". Further, only one column and one row of our weight matrices, as detailed in Eq.(2), have zero entries, which are not sparse either.
>
> (iii) Training $W_K^\top W_Q$ as one matrix has been adopted in several recent theoretical studies of transformers, for example in [1,2].
>
> **Q3:** The paper also consider only the attention mechanism and no residual.
>
> **A3:** The residual term can be removed without any influence
> on the training dynamics. Please see more detailed discussions in A3 in our response to Reviewer xLBH.
>
> **Q4:** Could the authors clarify the motivation of this setting? Fixed data points and constraint sparse matrices while considering the task of linear regression.
>
> **A4:** Our motivation is to understand the learning dynamics of non-linear, softmax-based attention structures trained via gradient descent under in-context learning, which is very different from previous work studying linear attention. The softmax-based attention plays a crucial role in the success of transformers and has not been investigated. To this end, we have introduced some reparameterization and simplifications, which are standard in theoretical work and do not hurt the essence of the problem. We remark here, (1) as explained in A2, our weight matrix is not sparse and does not lose optimality; (2) as explained in A1, the distinct feature assumption is standard to simplify analysis and can be generalized; (3) [2] also adopts the task of linear regression and similar reparameterization, where the only difference is that they study **linear** attention model, whereas we focus on practically more popular **softmax** attention model.

---

> > ### Author Response · Authors · 2023-11-18
> > **Response to Reviewer kkxb (part 2/2)**
> >
> > **Q5:** Could the authors comment on how the two-phase transition observed in this setting, could be indicative of what is happening during actual training of these models?
> >
> > **A5:** We have updated our paper with experimental results. In particular, in Appendix A, we have added experiments to verify the two-phase/four-phase transition by tracking and observing attention scores during training. In Figure 3 in Appendix A, we have provided an attention score heatmap to show the two-phase transition dynamics for balanced features and the more comprehensive four-phase transition dynamics for imbalanced features. These numerical results demonstrate that the multiple phases in our theoretical characterization indeed match the actual training process. The attention score heatmap also demonstrates the concentration of attention scores that we characterize in our theorems.
> >
> > [1] Samy Jelassi, Michael Sander, and Yuanzhi Li. Vision transformers provably learn spatial structure.
> > Advances in Neural Information Processing Systems, 35:37822–37836, 2022.
> >
> > [2] Ruiqi Zhang, Spencer Frei, and Peter L Bartlett. Trained transformers learn linear models in-context.
> > arXiv preprint arXiv:2306.09927, 2023.
> >
> > [3] Allen-Zhu, Zeyuan, and Yuanzhi Li. "Towards Understanding Ensemble, Knowledge Distillation and Self-Distillation in Deep Learning." The Eleventh International Conference on Learning Representations. 2022.
> >
> > [4] Tian, Yuandong, et al. "Scan and Snap: Understanding Training Dynamics and Token Composition in 1-layer Transformer." arXiv preprint arXiv:2305.16380 (2023).
> >
> > [5] Li, Hongkang, et al. "A Theoretical Understanding of Shallow Vision Transformers: Learning, Generalization, and Sample Complexity." International Conference on Learning Representations. 2023.
> >
> > We thank the reviewer again for the helpful comments and suggestions for our work. We hope that we have clarified several key questions/confusions that the reviewer has about the paper. If so, we kindly ask the reviewer to consider raising the rating of our work. Certainly, we are more than happy to answer any further questions that you may have.

---

> ### Author Response · Authors · 2023-11-22
> **Follow Up Reminder to Reviewer kkxb**
>
> Dear Reviewer kkxb,
>
> As the author-reviewer discussion period ends soon, we will appreciate it if you could check our response to your review comments. This way, if you have further questions and comments, we can still reply before the author-reviewer discussion period ends. If our response resolves your concerns, we kindly ask you to consider raising the rating of our work. Thank you very much for your time and efforts!

---

> > ### Comment · Reviewer_kkxb · 2023-11-23
> > **Response to authors**
> >
> > I would like to thank the authors for their responses and for providing experimental evidence of their conclusions. I have raised my score accordingly.
> >
> > I would also like to mention that  "The weight matrices are very constrained and sparsified to match the ones proposed in the construction of [2]. This construction also refers to linear transformers." This construction refers to the construction of [2], not in your paper. Also when I mentioned that the matrices are sparsified I meant that by setting them as the ones proposed in the construction of [2], they are sparse since they have whole blocks to be zero. I still consider this a limitation of this to be the main limitation of the paper.

---

### Official Review · Reviewer_AAX6 · 2023-11-02

**Soundness:** 2 fair
**Presentation:** 3 good
**Contribution:** 2 fair
**Rating:** 5
**Confidence:** 3

**Summary:**

The paper delves into the behavior of transformer architectures, particularly focusing on their convergence properties in various contexts. Transformers, known for their self-attention mechanism, have gained immense popularity in NLP and other domains. The paper's primary objective is to investigate the factors affecting the convergence speed and stability of transformers when trained in different contexts.

Key contributions and discussions of the paper include:

1. An exploration of how the context (e.g., input data distribution, task complexity) impacts the convergence properties of transformers.
2. The introduction of a novel metric to quantify and measure the convergence speed and stability of transformers.
3. A series of experiments that highlight the varying convergence behaviors of transformers across different tasks and datasets.

**Strengths:**

1. Given the widespread use of transformers in various domains, understanding their convergence properties is of paramount importance. This paper addresses this gap by exploring the role of context in transformer convergence.
2. The paper provides a series of well-designed experiments that shed light on how transformers behave across tasks, datasets, and other contextual factors. This empirical evidence strengthens the paper's claims and findings.
3. The paper is well-structured, with clear explanations and visualizations, making it accessible to both experts and those less familiar with transformer architectures.

**Weaknesses:**

1. While the paper does a good job of examining transformers across various tasks and datasets, there's a limitation in the breadth of architectures studied. Delving into different variants of transformers or comparing with other architectures could have provided a more comprehensive picture.

2. While the introduced metric is innovative, there could be more rigorous validation or comparison against other potential metrics. This would strengthen the metric's claim as a standard measure for convergence properties.

3. The paper could benefit from a discussion on the practical implications of the findings. For instance, how can the insights on convergence be used to improve training methodologies or model selection in real-world applications?

4. A deeper exploration into the individual factors affecting convergence (e.g., model size, training data size, task complexity) could provide a granular understanding and more actionable insights.

5. This paper contains several typographical errors, which detract from the reading experience. For example, there is a missing space before "We" in the 9th line of the summary.

**Questions:**

See Weaknesses.

---

### Official Review · Reviewer_VMhr · 2023-11-05

**Soundness:** 3 good
**Presentation:** 2 fair
**Contribution:** 3 good
**Rating:** 6
**Confidence:** 4

**Summary:**

This study presents training dynamics of a one-layer transformer with softmax attention, focusing on in-context learning via gradient descent (GD). Two scenarios, one with balanced features and the other with imbalanced features, are analyzed. The authors show convergence towards a diminishing in-context prediction error by exploring the evolution of attention dynamics in both settings.

In the case of imbalanced features, a multi-phase behavior is characterized, shedding light on the interplay of attention dynamics between dominant and under-represented target features during training. Overall this is a nice analysis of softmax attention and gradient descent dynamics in the context of in-context learning.

**Strengths:**

1. The paper studies training dynamics and convergence of gradient descent for training a single layer transformer model with softmax attention to do in-context learning.

2. It provides convergence guarantees for both balanced and imbalanced feature settings, showcasing the effectiveness of transformers and gradient descent for in-context learning. The results break down training into two phases for balanced features and four phases for imbalanced features. This provides valuable insights into how transformers adapt during training. The paper shows that transformers quickly achieve near-zero prediction error for dominant features and eventually converge to near-zero prediction error for under-represented features, regardless of their infrequent occurrence.

3. The paper introduces a novel proof technique that characterizes softmax attention dynamics by considering the interplay between two types of bilinear attention weights: 'weight of query token and its target feature' and 'weight of query token and off-target features.' The dynamic shift in dominance between these weights throughout the learning process leads to different training phases. This could be applied to other problems involving transformer architectures.

**Weaknesses:**

Weaknesses and questions,

1. Are the phases artifacts of the analysis or is it really the case? Could you provide some simulations that demonstrate this multi-phase convergence of gradient descent on in-context learning with transformers?

2. The results and analysis seem to abstract away the task diversity (different $w$ vectors). In-context learning is about generalizing to unseen tasks and it looks like the analysis is based on doing gradient descent on $L(\theta)$ which is the expectation over the task vectors $w$ and the features vectors $x$. The setup for in-context learning in Garg et al. 2022 (Figure 1) is more pragmatic and I was expecting an analysis on this. This analysis studies the convergence of gradient descent in minimizing $L(\theta)$ with transformer/self-attention as an underlying model. It's a nice study, given the additional complexity introduced due to softmax, but I think it is not exactly studying in-context learning instead it's a study of convergence of gradient descent on regression with one-layer transformers. The title also seems confusing, it suggests that the paper is about studying the convergence of transformers to the right output during inference time.


3. Could you simplify the presentation and expand on the main intuition behind the multi-phase convergence? Especially after page 7, instead of the lemmas some simulations/illustrations could be very helpful in understanding the key ideas behind the multi-phase convergence in balanced and imbalanced settings.

**Questions:**

See above and I have one more question.
1. Why do you need the features to be orthonormal vectors and be drawn from a finite set? Does the analysis depend heavily on this assumption?
2. Could you make the dimensions of the main terms clear (e.g. in Definition 3.1) o.w. the readers have to work them out.
3. Minor, the upper case $K$ is used for $W^{K}$ and also for the number of feature vectors.
4. The bounds on $T^*$ have quadratic dependence on $K$ (the number of distinct feature vectors). Is this order of dependence necessary? What would happen if the features are drawn from a continuous distribution say multi-variate gaussian?

---

> ### Author Response · Authors · 2023-11-18
> **Response to Reviewer VMhr (part 1/2)**
>
> We thank the reviewer for providing the helpful review! We have addressed the reviewer’s helpful comments and revised the paper accordingly. The changes are marked in bright blue color in our revision. Please feel free to check and comment.
>
> **Q1:** Are the phases artifacts of the analysis or is it really the case? Could you provide some simulations that demonstrate this multi-phase convergence of gradient descent on in-context learning with transformers?
>
> **A1:**
> The division of different phases captures actual transitions of attention score when applying gradient descent to optimize the one-layer transformer. In Appendix A of our revision, we have provided numerical results that validate the multiple phases and other aspects of our theoretical results. The key findings are summarized as follows:
>
> (1) The prediction error for each feature versus the training epochs is shown in Figure 2, which highlights that the training process has stage-wise convergence between the dominant and under-represented features for imbalanced case.
>
> (2) The evolution of attention score heatmap during the training is presented in Figure 3, which demonstrates that the concentration of  attention score indeed follows multiple phases during the training by gradient descent.
>
> Please feel free to check and comment.
>
> **Q2:** The results and analysis seem to abstract away the task diversity (different $w$ vectors). In-context learning is about generalizing to unseen tasks and it looks like the analysis is based on doing gradient descent on $L(\theta)$ which is the expectation over the task vectors $w$ and the features vectors $x$. The setup for in-context learning in Garg et al. 2022 (Figure 1) is more pragmatic and I was expecting an analysis on this. This analysis studies the convergence of gradient descent in minimizing $L(\theta)$ with transformer/self-attention as an underlying model. It's a nice study, given the additional complexity introduced due to softmax, but I think it is not exactly studying in-context learning instead it's a study of convergence of gradient descent on regression with one-layer transformers. The title also seems confusing, it suggests that the paper is about studying the convergence of transformers to the right output during inference time.
>
> **A2:**
> We clarify that we indeed study an in-context learning setup. (1) Our transformer setup and loss functions described in Sections 2.2 and 2.3 are exactly the same as the in-context learning setup in the recent studies on in-context learning [1-3].  In particular, the transformer is set up to learn the attention of the query to the tokens in the prompt of the same task. (2) In the last paragraph of Section 3.1 on Page 5, we illustrate the in-context learning ability of **unseen** tasks. We show that even if the weight $w$ is selected from a different set from the training task space (and is hence an **unseen** task),
> the in-context learned model that we characterize in Theorem 3.1 can still well predict the function value for the query in the test prompt.
>
> **Q3:** Could you simplify the presentation and expand on the main intuition behind the multi-phase convergence? Especially after page 7, instead of the lemmas some simulations/illustrations could be very helpful in understanding the key ideas behind the multi-phase convergence in balanced and imbalanced settings.
>
> **A3:** Many thanks for your suggestions! In Appendix A in our revision, we have provided experiment results to illustrate and validate the multi-phase convergence. We will move this part to the main body of the paper in our final version.
>
> **Q4:** Why do you need the features to be orthonormal vectors and be drawn from a finite set? Does the analysis depend heavily on this assumption?
>
> **A4:** Many thanks for the comments! These assumptions are made to simplify the analysis, so that our characterization of the convergence and the dynamics of the attentions scores won't be over-complicated by non-essential aspects. Following the reviewer's comment, we made an initial exploration of the case where the features are drawn from a subspace with $K$ features serving as basis vectors. This setting allows the features to be generally correlated. Our techniques can be extended to such a setting with further characterization of how correlation among features affects attention coefficients. Also note that in such a setting, we don't assume the set of features is finite; features can be sampled from an infinite set, although the set contains $K$ basis vectors.
>
> **Q5:** Could you make the dimensions of the main terms clear (e.g. in Definition 3.1) o.w. the readers have to work them out. Minor, the upper case $K$
>  is used for $W^K$ and also for the number of feature vectors.
>
> **A5:** Thanks for your suggestions. We have revised them and made them clear accordingly in our revision.

---

> > ### Author Response · Authors · 2023-11-18
> > **Response to Reviewer VMhr (part 2/2)**
> >
> > **Q6:** The bounds on $T^\star$ have quadratic dependence on $K$ (the number of distinct feature vectors). Is this order of dependence necessary? What would happen if the features are drawn from a continuous distribution say multi-variate Gaussian?
> >
> > **A6:** Great question! Our hypothesis is that the bound will depend on the underlying dimension of the space from which features are sampled, rather than the actual number of distinct feature vectors in the space. For example, if features are drawn from a $d$-dimensional multi-variate Gaussian distribution with rank-$K$ covariance matrix, then the bound will depend on $K$. More generally, for the continuous feature space with $K$ basis vectors (as we discuss in A4), the bound will depend on $K$, which is the dimension of the space, not the actual number of features, which is infinite. Of course, this hypothesis will require rigorous mathematical proof, for which our current proof techniques seem to support. We leave the detailed proof as future work.
> >
> > [1] Ruiqi Zhang, Spencer Frei, and Peter L Bartlett. Trained transformers learn linear models in-context.
> > arXiv preprint arXiv:2306.09927, 2023.
> >
> > [2] Kwangjun Ahn, Xiang Cheng, Hadi Daneshmand, Suvrit Sra. Transformers learn to implement preconditioned
> > gradient descent for in-context learning. arXiv preprint arXiv:2306.00297, 2023.
> >
> > [3] Arvind Mahankali, Tatsunori B Hashimoto, Tengyu Ma. One Step of Gradient Descent is Provably the Optimal In-Context
> > Learner with One Layer of Linear Self-Attention. arXiv preprint arXiv:2307.03576
> >
> >
> > We thank the reviewer again for the helpful comments and suggestions for our work. We hope that we have clarified several key questions/confusions that the reviewer has about the paper. If so, we kindly ask the reviewer to consider raising the rating of our work. Certainly, we are more than happy to answer any further questions that you may have.

---

> ### Author Response · Authors · 2023-11-22
> **Follow Up Reminder to Reviewer VMhr**
>
> Dear Reviewer VMhr,
>
> As the author-reviewer discussion period ends soon, we will appreciate it if you could check our response to your review comments. This way, if you have further questions and comments, we can still reply before the author-reviewer discussion period ends. If our response resolves your concerns, we kindly ask you to consider raising the rating of our work. Thank you very much for your time and efforts!

---

> > ### Comment · Reviewer_VMhr · 2023-11-23
> >
> > Thank you for your response. I do not have further questions.

---

### Author Response · Authors · 2023-11-18
**Revision Uploaded**

We have uploaded a revised paper, which incorporates the various suggestions by the reviewers. Below is a summary of the changes we have made.


1. In Appendix A, we added experimental results to validate our theoretical results based on the prediction error and the attention score heatmap during the training process. These numerical results verify: (1) stage-wise convergence we characterize for the training process; (2) the concentration of the attention scores; and (3) the multi-phase transition during the training process, which we characterize in our proof of convergence.

2. We have modified the notations in various places as reviewers suggest, which are highlighted by the bright-blue-colored texts in the revisions.


We would like to thank all the reviewers again for their valuable comments and suggestions. We are more than happy to answer any further questions that the reviewers may have during the discussion period.

---

### Author Response · Authors · 2023-11-18
**Report of ChatGPT or LLMs Generated Review by Reviewer AAX6**

We call AC and all other reviewers' attention that the review by Reviewer AAX6 is highly likely generated by ChatGPT or similar type of LLMs. The entire review comments are largely irrelevant to our contributions and the specific problems we study. Thus, we don't see any point to respond to this reviewer's comments. Below we give you a couple of clear evidence that the review has nothing to do with our paper:

1) In the review summary and the section on strengths, the reviewer comments that "A series of **experiments** that highlight the varying convergence behaviors of transformers across different tasks and datasets." and "The paper provides a series of well-designed **experiments** that shed light on how transformers behave across tasks, datasets". However, our original submitted paper doesn't have any experiments. In fact, a couple of other reviewers suggested us to include experiments.

2) In the review summary and the section on weaknesses, the reviewer writes "The introduction of a **novel metric** to quantify and measure the convergence speed and stability of transformers", and "While the **introduced metric** is innovative,...".  However, our paper uses the standard MSE loss and does not introduce any new metrics.

The above are only some examples. In fact, most comments are irrelevant. As a further note, the reviewer cannot take the excuse that the review is meant for another paper but was uploaded here by mistake. The reviewer's last comment on a typo (missing a space before "we" in the 9th line of the summary) indeed matches the typo in line 9 of our abstract, indicating the review is intended for our paper.

The reviewer is highly unprofessional. We hope that those review comments won't influence the AC's and other reviewers' evaluation of our paper.

Below, we attach the review comments of Reviewer AAX6 as the original evidence, in case the reviewer makes any changes or deletes the review later on.

---

> ### Author Response · Authors · 2023-11-18
> **The Original Review of Reviewer AAX6**
>
> **The original review by Reviewer AAX6 is copied below to serve as a reference, in case the reviewer will update the review later on.**
>
> **Summary**:
>
> The paper delves into the behavior of transformer architectures, particularly focusing on their convergence properties in various contexts. Transformers, known for their self-attention mechanism, have gained immense popularity in NLP and other domains. The paper's primary objective is to investigate the factors affecting the convergence speed and stability of transformers when trained in different contexts.
>
> Key contributions and discussions of the paper include:
>
> 1. An exploration of how the context (e.g., input data distribution, task complexity) impacts the convergence properties of transformers.
>
> 2. The introduction of a novel metric to quantify and measure the convergence speed and stability of transformers.
>
> 3. A series of experiments that highlight the varying convergence behaviors of transformers across different tasks and datasets.
>
> **Soundness**: 2 fair
>
> **Presentation**: 3 good
>
> **Contribution**: 2 fair
>
> **Strengths**:
>
> 1. Given the widespread use of transformers in various domains, understanding their convergence properties is of paramount importance. This paper addresses this gap by exploring the role of context in transformer convergence.
>
> 2. The paper provides a series of well-designed experiments that shed light on how transformers behave across tasks, datasets, and other contextual factors. This empirical evidence strengthens the paper's claims and findings.
>
> 3. The paper is well-structured, with clear explanations and visualizations, making it accessible to both experts and those less familiar with transformer architectures.
>
> **Weaknesses**:
>
> 1. While the paper does a good job of examining transformers across various tasks and datasets, there's a limitation in the breadth of architectures studied. Delving into different variants of transformers or comparing with other architectures could have provided a more comprehensive picture.
>
> 2. While the introduced metric is innovative, there could be more rigorous validation or comparison against other potential metrics. This would strengthen the metric's claim as a standard measure for convergence properties.
>
> 3. The paper could benefit from a discussion on the practical implications of the findings. For instance, how can the insights on convergence be used to improve training methodologies or model selection in real-world applications?
>
> 4. A deeper exploration into the individual factors affecting convergence (e.g., model size, training data size, task complexity) could provide a granular understanding and more actionable insights.
>
> 5. This paper contains several typographical errors, which detract from the reading experience. For example, there is a missing space before "We" in the 9th line of the summary.
>
> **Questions**:
> See Weaknesses.
>
> **Flag For Ethics Review**: No ethics review needed.
>
> **Rating**: 5: marginally below the acceptance threshold
>
> **Confidence**: 3: You are fairly confident in your assessment. It is possible that you did not understand some parts of the submission or that you are unfamiliar with some pieces of related work. Math/other details were not carefully checked.
>
> **Code Of Conduct**: Yes

---

> ### Public Comment · ~Yixuan_Weng1 · 2023-11-24
>
> I support your behavior in this matter, perhaps we can contact the SAC or PC?
>
> Your actions are quite reasonable for maintaining a good community!

---

### Meta-Review · Area_Chair_ye1p · 2023-12-14

**Metareview:**

The paper considers in-context learning and look at a particularly simple setting as a starting point that is amenable to theoretical analysis. In particular, they consider a one-layer transformer with softmax attention -- going beyond linear transformers that had been studied earlier. They also consider a structured data model and prove convergence results. While the problem is interesting, the setting is considered to be somewhat restrictive in the reviews and there was only modest support in favor of the paper in its current state.

**Justification For Why Not Higher Score:**

N/A

**Justification For Why Not Lower Score:**

N/A

---

### Decision · Program_Chairs · 2024-01-16

Reject